# IL-1 protects from fatal systemic candidiasis in mice by inhibiting oxidative phosphorylation and hypoxia

Sofia Horn [1], Mareike Schmid[1], Ivan Berest [1], Federica Piattini[1], Jing Zhang[2], Katrien de Bock [2], Olivier Devuyst [3], Stellor Nlandu Khodo [3], Jan Kisielow [1] & Manfred Kopf [1] ✉

Invasive *C. albicans* infections result in high mortality rates. While IL-1 is important to combat *C. albicans* infections, the underlying mechanisms remain unclear. Using global and conditional *Il1r1* knockouts in mice, here we show that IL-1R signaling in non-hematopoietic cells in the kidney and brain is crucial for a protective response. In the kidney, endothelial IL-1R contributes to fungal clearance independent of neutrophil recruitment, while IL-1R in hematopoietic cells is dispensable. IL-1R signaling indirectly recruits neutrophils and monocytes in the brain by regulating chemokines and adhesion molecules. Single-nucleus-RNA-sequencing data implicates excessive metabolic activity and oxidative phosphorylation across all cell types in the kidney of *Il1r1*-deficient mice within a few hours upon infection, with associated, localized hypoxia at infection foci. Lastly, we find that hypoxia promotes fungal growth and pathogenicity. In summary, our results show that IL-1R-signaling in non-hematopoietic cells is required to prevent fatal candidiasis by inhibiting a metabolic shift, including excessive oxidative phosphorylation and hypoxia.

*Candida albicans* is a fungal opportunistic pathogen, residing as a part of the commensal microbiota in mucosal tissues of most humans. In healthy humans, if *Candida* yeast enter the bloodstream circulation, they are removed by circulating granulocytes[1]. Systemic candidiasis is a condition arising in immune-compromised individuals or hospitalized patients with catheters, in which *Candida* species leak into the bloodstream and invade various internal organs. It is challenging to treat such infections, as evident by the high mortality rates of approximately 46–75%[2,3]. To establish infection, the *Candida* yeast invade organs such as the kidney, the brain, the liver, and the spleen, where they change their morphology into hyphae[4], a filamentous morphology that is highly pathogenic. Despite the availability of many anti-fungal agents, treating candidiasis in hospitals is often ineffective, illustrating the need for further research. A better understanding of anti-fungal immunity would allow more

effective treatment strategies. The basic principles of sensing *C. albicans* infection and innate immune responses involved in host protection have been well established, with neutrophils, inflammatory monocytes, and tissue-resident macrophages playing key roles[5–11]. Neutrophils can recognize and block *Candida* hyphae by producing reactive oxygen species (ROS) and anti-microbial peptides and forming NETs. Infiltrating inflammatory monocytes differentiate at the site of infection into monocyte-derived macrophages, which acquire the ability to produce reactive nitrogen intermediates and thereby enable fungicidal activity[9,12]. Several members of the IL-1 cytokine family[13], IL-17A/F[14], IL-12, and IL-23 have all been recognized to contribute to efficient control of fungal infections[15]. Indeed, both IL-1α and IL-1β have previously been implicated in immune protection from mucosal and systemic candidiasis[16–20]. However, detailed cellular and molecular mechanisms by which IL-1R signaling

[1]Department of Biology, Institute of Molecular Health Sciences, ETH Zurich, Zurich, Switzerland. [2]Department of Health Sciences and Technology, Laboratory of Exercise and Health, ETH Zurich, Zurich, Switzerland. [3]Institute of Physiology, University of Zurich, Zurich, Switzerland. ✉e-mail: manfred.kopf@ethz.ch

contributes to the control of systemic candidiasis are insufficiently established.

Here we show that IL-1R signaling in immune cells is dispensable for protection against fungal growth and for survival during systemic candidiasis in mice. In addition, IL-1R is not required for a protective pro-inflammatory response. An extensive analysis of kidney- and brain-infiltrating cells in global and conditional IL-1R-deficient mice revealed that IL-1R-signaling in non-hematopoietic cell types is responsible for protection. A delayed neutrophil and monocyte infiltration explains excessive *Candida* growth in the brain but not in the kidney of *Il1r1-/-* mice. In almost every cell type of the kidney, the IL-1 pathway is essential to inhibit metabolic dysregulation, including excessive oxidative phosphorylation, and increased hypoxia, which favors fungal growth and pathology. Together, our data suggest distinct organ-specific mechanisms of IL-1-mediated control of *Candida* with a role of IL-1 in preventing hypoxia in the kidney.

## Results

### IL-1 is crucial for the immune defense against *C. albicans*

Once systemic infection is established, *C. albicans* penetrates and grows in various organs of the host. In murine systemic candidiasis, the primary targets of fungal growth are the kidneys and brain. *C. albicans* is also found in the liver and spleen, though to a lesser extent. Immediately after systemic infection with *C. albicans*, the expression of IL-1α, IL-1β, and IL-1R is rapidly upregulated in the kidneys (Fig. 1A), possibly suggesting a role in the host defense against fungal infection. Indeed, IL-1R-deficient mice show high susceptibility to *C. albicans*, indicated by rapid weight loss and approximately 1000-fold elevated fungal titers in the kidneys and brain, but not in the spleen or liver, three days post-infection with $10^5$ CFU *C. albicans* (Fig. 1B, C), when mice had to be euthanized to prevent undue suffering. *Il1r1-/-* mice exhibit significantly increased fungal titers in the kidneys and brain starting at 48 hours post-infection (supplementary Fig. 1D); however, no differences compared to controls were observed at 24 h p.i. (Fig. 1D).

To determine which IL-1 cytokine, IL-1α or IL-1β, is more critical in the anti-fungal immune response, we infected *Il1α-/-* and *Il1β-/-* mice. Interestingly, the absence of IL-1α only slightly increased susceptibility to the fungus, and this was observed only in the brain. However, the absence of IL-1β significantly impacted immune defense and led to elevated fungal growth in both the kidney and brain (Fig. 1E), which is consistent with a $10^7$-fold higher induction of *Il1b transcripts* compared to *Il1a* transcripts during the first two days post-infection. This indicates a crucial role for IL-1β, but not for IL-1α, in the immune defense against *C. albicans*.

To visualize the infection in-vivo, we performed histological staining for fungi, using Modified Grocott Methanamine Silver (GMS) staining, in kidney and brain sections from day 2 infected *Il1r1-/-* mice and *Il1r1+/-* controls. Histopathological analysis of the kidney revealed dense clusters of *C. albicans* hyphae located in inflammatory foci in the *Il1r1-/-* mice, while the control kidneys had only inflammatory foci, with almost no fungi (Fig. 1F). Interestingly, the analysis of the brain tissue of *Il1r1-/-* mice showed that *C. albicans* not only creates dense clusters of hyphae but is also disseminated throughout the tissue (Fig. 1G, panel A). In contrast, barely any *C. albicans* were detectable in control brains at day 2 (Fig. 1G).

Our results suggest that *Il1r1-/-* mice are highly susceptible to systemic candidiasis, especially in the kidney and brain.

### *Il1r1*-deficient mice exhibit a hyper-inflammatory state upon *C. albicans* infection

The immune response to systemic candidiasis consists of the fungi being recognized by tissue-resident cells, who subsequently respond by immediately releasing chemokines. This is followed by the infiltration of leukocytes belonging to the innate immune system, such as neutrophils and monocytes[21].

To obtain information about the inflammatory state of the *Il1r1-/-* mice, we analyzed the cytokine and chemokine protein levels in lysates of the kidney and brain of *WT* and *Il1r1-/-* mice 2.5 days post-infection (p.i.). The cytokine analysis revealed higher levels of IL-6, TNFα, and IL-23 in the infected kidneys of *Il1r1-/-* mice compared to the kidneys of *WT* mice, and an even more striking difference was seen in the brain tissue of the *Il1r1-/-* mice. The brain of infected *Il1r1-/-* mice contained dramatically higher levels of IL-6, TNFα, and MCP-1, compared to the brains of *WT* mice (Fig. 2A). The chemokine analysis revealed elevated levels of CCL3, CCL4, CCL5, CXCL1 and CXCL13 in the kidney and brain tissues of *Il1r1-/-* mice compared to controls. Additionally, levels of CXCL10 were up in the brain of *Il1r1-/-* mice (Fig. 2A). These data demonstrate that *Il1r1-/-* mice experience a hyperinflammatory state at this stage of the infection, indicating that IL-1R signaling is not necessary for the activation of a pro-inflammatory immune response during systemic *C. albicans* infection.

We next examined whether *Il1r1-/-* mice display differences in infiltration of specific cell types into the infected tissues, which might explain the elevated susceptibility. Therefore, we characterized CD45+ cells in the kidney and in the brain of *Il1r1-/-* mice and *WT* controls by high-dimensional flow cytometry at day 3 p.i.. As the phenotype is seen in the first days of infection, we focused on myeloid cells, which are known to infiltrate organs relatively early after initiating infection. To analyze the acquired data, we used an unbiased approach, including the dimensionality reduction algorithm "UMAP" and the clustering algorithm "Phenograph"[22].

The Phenograph algorithm identified 25 cell clusters in the kidney and 23 cell clusters in the brain (Fig. 2B, C). To annotate the identified cell types, we grouped the clusters identified in the kidney and in the brain into 8 cell types including neutrophils, eosinophils, cDC1, cDC2, Ly6C+ monocytes, MHCII+Ly6C-CD11b+ monocyte-derived inflammatory macrophages (Mo-Mac), CD11blow kidney macrophages and brain resident CD45lowCD64+ microglia (Fig. 2B, C).

Consistent with the overshooting inflammatory response, we found that *Il1r1-/-* mice exhibited significantly elevated numbers of neutrophils, Ly6C+ monocytes, and Ly6C- monocytes, both in the kidney and in the brain, (Fig. 2D, E). In contrast, Mo-Mac numbers were down in both organs. CD45lowCD64+ microglia were reduced in the brain, whereas CD11blow macrophages in the kidney were unaffected. Analysis using manual gating (supplementary Fig. 1A) and definition of the cell types using the same marker panel reproduced the results reliably (quantification of multiple experiments in supplementary Fig. 1B). Notably, our panel at this point included only the markers CD45 and CD64 to define microglia, and it might not reflect all microglia and resident immune populations of the brain. Later analysis, using a more detailed antibody panel, allowed better characterization of the brain immune populations and astrocytes (supplementary Fig. 2A–D, discussed below).

No differences in the number of tissue myeloid cell types in the kidney and brain were observed prior to infection in naïve mice, excluding the role of IL-1R signaling in the steady state (supplementary Fig. 1C).

Thus, consistent with elevated levels of several chemokines that are known to act on migration and/or polarization of myeloid cells, neutrophils, Ly6C+ and Ly6C- monocytes were massively increased in brain and kidney. Considering that neutrophils and inflammatory monocytes are fungicidal, increased *Candida* growth in the presence of elevated numbers of neutrophils may indicate that they are dysfunctional.

### Neutrophils and inflammatory monocytes are functionality intact in *Il1r1*-deficient mice

To explore whether IL-1R signaling is vital for the ability to kill *Candida*, we assessed the functionality of neutrophils and monocytes from *Il1r1-/-* and *WT* mice. Phagocytes eliminate engulfed fungi, using both

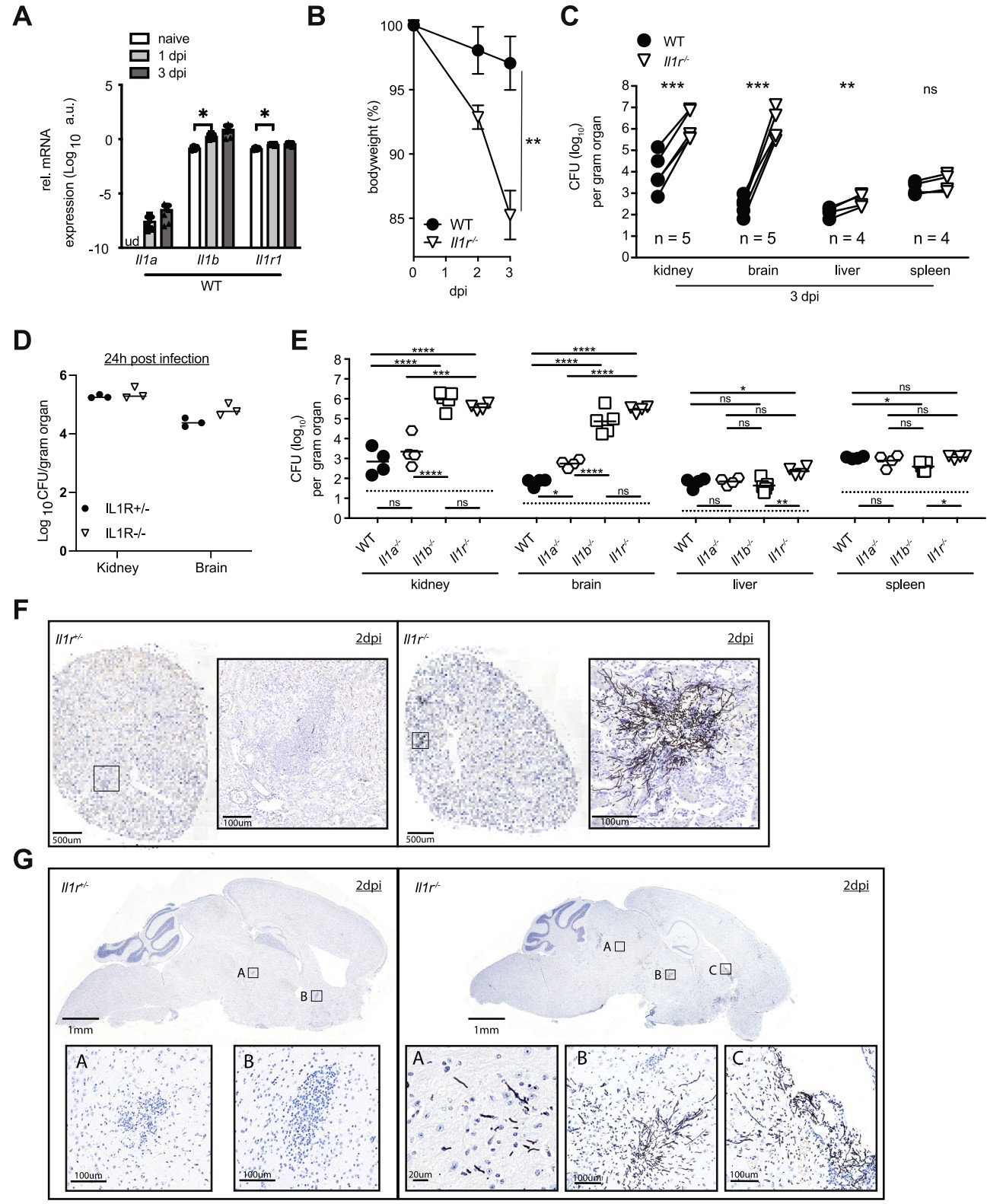

oxidative and non-oxidative killing mechanisms. The inducible nitric oxide synthase (iNOS) is an enzyme involved in producing reactive nitrogen intermediates (RNI), which have direct candidacidal activity[23]. Interestingly, the frequency of iNOS+Ly6C+ monocytes was dramatically increased in the kidney and brain of *Il1r1*[-/-] compared to *WT* mice (Fig. 3A), suggesting that the nitrosative killing capacity in the organ is increased rather than impaired by the lack of IL-1R, again consistent with elevated inflammatory signals and fungal burden in the KO mice.

To assess whether *Il1r1*[-/-] neutrophils have impaired candidacidal activity, we isolated neutrophils from blood and infected kidneys at 3 days p.i. and performed an ex-vivo killing assay with *C. albicans* yeast and hyphae. Figure 3B shows neutrophils from infected *WT* and *Il1r1*[-/-] mice displayed comparably potent *Candida* killing. However, to control *Candida* hyphae in-vivo, neutrophils mainly use NETs[24–26]. Thus, we assessed the capability of neutrophils from *Il1r1*[-/-] mice to produce NETs in response to a stimulus. Neutrophils isolated from

**Fig. 1 | IL-1 is crucial for the immune defense against *C. albicans*. A** Expression of *Il1r1* and its ligands in whole kidney of *WT* mice 1 and 3 days after infection with $10^5$ CFU *C. albicans*. *Il1a*, *Il1b* and *Il1r1* gene expression were measured using real time PCR, and normalized to *TBP* expression. (n = 4/5 mice per timepoint), ud=undetectable, au=arbitrary unit. Statistical test: Two-sided multiple unpaired t tests, P < 0.05 is marked with *. P[Il1b] = 0.0246, P[Il1r1] = 0.0002. Data are presented as mean values with +/- SD. **B** Bodyweight loss of *WT* and *Il1r1⁻/⁻* mice after infection with $10^5$ CFU *C. albicans*. (n = 3 mice per group, representative of more than 2 independent repeats), mean values +/- SD are indicated. Statistical test: Two-sided unpaired T test for Area Under Curve, P < 0.01 marked with **. P[AUC] = 0.007. **C** Compiled data of fungal burden in kidney, brain, liver and spleen of *WT* and *Il1r1⁻/⁻* mice, at 3 days p.i. with $10^5$ CFU *C. albicans*. Data from independent experiments (number of independed repeats is indicated). Icons represent the mean of a group in each experiment, means from the same experiment are connected with a line. n = 3/5 mice per group in each experiment. Statistical test: Two-sided paired t-test, P < 0.05 marked with *, P < 0.01 marked with **, P < 0.001 marked with ***.

P[kidney] = 0.0002, P[brain] = 0.0003. P[liver] = 0.0059, P[spleen] = 0.0538. **D** Fungal burden in kidney and brain of *WT* and *Il1r1⁻/⁻* mice 24 h after infection with $10^5$ CFU *C. albicans* (n = 3 mice per group, representative of more than 2 independent repeats). Statistical test: Two-sided multiple unpaired t tests, Individual variance assumption for each organ, Holm-Sidak method alpha = 0.05. **E** Fungal burden in kidney, brain, liver and spleen of *WT*, *Il1a⁻/⁻*, *Il1b⁻/⁻* and *Il1r1⁻/⁻* mice, at 3 days after infection with $10^5$ CFU *C. albicans*. (n = 4 mice per group, representative of 2 independent repeats) Statistical test: One-way ANOVA per organ with Tukey's multiple comparison test. P < 0.05 marked with *, P < 0.01 marked with **, P < 0.001 marked with ***, P < 0.0001 marked with ****. **F, G** GMS (Modified Grocott methenamine silver) staining of kidney (**F**) and brain (**G**) sections of *Il1r1⁺/⁻* and *Il1r1⁻/⁻* mice 48 h after infection with $10^5$ CFU *C. albicans*. Zoomed-in areas are marked with a square. (A representative picture from a single experiment with n = 3 mice per group is shown). Hematoxylin was used for background staining. Light microscopy is shown, no manipulations to the files were done.

bone marrow of *WT* and *Il1r1⁻/⁻* mice had comparable capacity to produce NETs in response to PMA, as well as in response to *C. albicans* (Fig. 3C). Notably, IL-1b stimulation did not affect the NET formation in *WT* or in *Il1r1⁻/⁻* mice. Next, we sought to determine whether neutrophils in the *Il1r1⁻/⁻* mice can produce NETs in response to *Candida* hyphae in vivo. For this, we used immunohistochemistry and stained consecutive slides of brain of infected *Il1r1⁺/⁻* and *Il1r1⁻/⁻* mice, at day 2 p.i., using anti-Ly-6G antibody and anti-Citrullinated Histone H3 antibody (specific for NETs) (Fig. 3D). This analysis confirmed the elevated neutrophil numbers in the *Il1r1⁻/⁻* mice compared to *WT* mice and showed citrullinated Histone H3 (Cit H3) staining surrounding the neutrophil clusters in *WT* and in *Il1r1⁻/⁻* mice, indicating that NET mediated *Candida* killing is not affected in *Il1r1⁻/⁻* mice.

Together, these data demonstrate that the fungicidal activity of neutrophils and iNOS expression by monocytes are not affected in *Il1r1⁻/⁻* mice, indicating that their severe susceptibility is less likely to be explained by defects in these well-established antifungal immune effector mechanisms.

To explain the high number of neutrophils seen in *Il1r1⁻/⁻* mice, we infected *WT* mice with increasing infection doses of *C. albicans* (supplementary Fig. 3A). As expected, we observed a positive correlation between the dose of infection and neutrophil numbers in the kidney and the brain at 3 days p.i. This correlation suggests that the observed numbers of neutrophils in *Il1r1⁻/⁻* mice at the height of infection is due to the severe fungal burden in these mice.

**Delayed neutrophil recruitment to the brain but not the kidney upon *C. albicans* infection of *Il1r1*-deficient mice**
Flow cytometry analysis of the brain at 48 h p.i., using a wider antibody panel, revealed significantly reduced numbers of both "activated" (Dectin⁺) and "homeostatic" microglia, as well as CD206⁺ border-associated macrophages and neutrophils in *Il1r1⁻/⁻* mice, compared to *Il1r⁺/⁻* littermates (supplementary Fig. 2A–C). At this time point (48 h post-infection), there was already a considerably higher fungal load in the brains of *Il1r1⁻/⁻* mice (supplementary Fig. 1D), which could indicate that microglia and border-associated macrophages might be especially vulnerable and die due to infection. Interestingly, we observed a significantly increased frequency of ACSA-2ˡᵒʷ astrocytes and decreased ACSA-2ʰⁱᵍʰ astrocytes at 48 h p.i. Decreased ACSA-2ʰⁱᵍʰ astrocytes were already observed in naïve *Il1r1⁻/⁻* mice compared to controls (supplementary Fig. 2D), indicating that steady state IL-1 production promotes the development or maintenance of the ACSA-2ʰⁱᵍʰ population. The heightened inflammation at 48 h p.i. may explain the increase in astrocyte numbers.

Since fungal titer in *Il1r1⁻/⁻* mice started to differ from *WT* mice already at 24 h p.i. (Fig. 1D), we suspected that a crucial role of IL-1R signaling may be instigated very early after infection. Indeed,

Ly6C⁺ monocyte and neutrophil recruitment to the brain were strikingly reduced in *Il1r1⁻/⁻* mice at 8 h, 16 h and 24 h p.i. (Fig. 3E–G). Notably, bulk RNA sequencing of the brain at 8 h p.i. showed significantly lower expression of molecules responsible for adhesion and extravasation into the tissue, such as *Icam*, *Lcn2*, *Sele* and *Vcam1* (supplementary Fig. 2E–H). Moreover, expression of the chemokines *Ccl1* and *Csf1*, which are known to promote monocyte recruitment and differentiation, was substantially reduced, as were the RNA and protein levels of CXCL1, a potent chemoattractant of neutrophils (Fig. 3H, supplementary Fig. 2 E–H).

These data suggest that the increased fungal titer in the brain of *Il1r1⁻/⁻* mice results from impaired monocyte and neutrophil recruitment due to reduced production of these chemokines.

On the other hand, in the kidneys of *Il1r1⁻/⁻* mice, compared to controls, we observed no differences in the number of monocytes and only a modest reduction of neutrophils at 8 h but not at 16 h and 24 h p.i. (Fig. 3F), which is unlikely to explain excessive *Candida* growth in this organ.

**IL-1 prevents severe susceptibility to *C. albicans* by acting on non-hematopoietic cells**
Depending on the context, *Il1r1* is expressed by many immune and non-hematopoietic cell types[27–29]. We wanted to examine which *Il1r1*-expressing cells are crucial for the defense against *C. albicans*. First, to assess the role of IL-1R in hematopoietic and non-hematopoietic cells, we used reciprocal bone marrow chimeras that were generated by reconstitution of sub-lethally irradiated *WT* mice with *Il1r1⁻/⁻* bone marrow (*Il1r1⁻/⁻* -> *WT*) and vice versa (*WT* -> *Il1r1⁻/⁻*). *WT* mice reconstituted with *WT* bone marrow (*WT* -> *WT*) and *Il1r1⁻/⁻* mice with *Il1r1⁻/⁻* bone marrow (*Il1r1⁻/⁻* -> *Il1r1⁻/⁻*) were used as controls. Interestingly, weight loss and fungal burden in mice with complete IL-1R deficiency (*Il1r1⁻/⁻* -> *Il1r1⁻/⁻*) was phenocopied by animals lacking IL-1R exclusively on non-hematopoietic cells (*WT* -> *Il1r1⁻/⁻*) (Fig. 4A). Surprisingly, mice lacking IL-1R exclusively in bone-marrow-derived leukocytes (*Il1r1⁻/⁻* -> *WT*) had only a very slight susceptibility compared to their *WT* -> *WT* counterparts. This suggests that IL-1R-signaling in radio-resistant non-hematopoietic cells is crucial for the early defense against *C. albicans*. However, as irradiation does not deplete brain resident microglia, we used the conditional KO mice *vavᶜʳᵉIl1r1ᶠˡ/ᶠˡ*, which have a deletion in the *Il1r1* gene locus in all hematopoietic cells, including brain microglia (supplementary Fig. 5A). A statistically significant elevated fungal load was only seen in the brain (Fig. 4B). However, the differences in fungal titer were much smaller compared to *Il1r1⁻/⁻* mice (i.e., 10-fold vs > 10,000-fold) (Fig. 1C), and similar to the *Il1r1⁻/⁻*→*WT* group from the bone marrow chimera experiment (Fig. 4A).

From these data, we conclude that the defense against systemic candidiasis strongly depends on *Il1r1* expression in non-hematopoietic

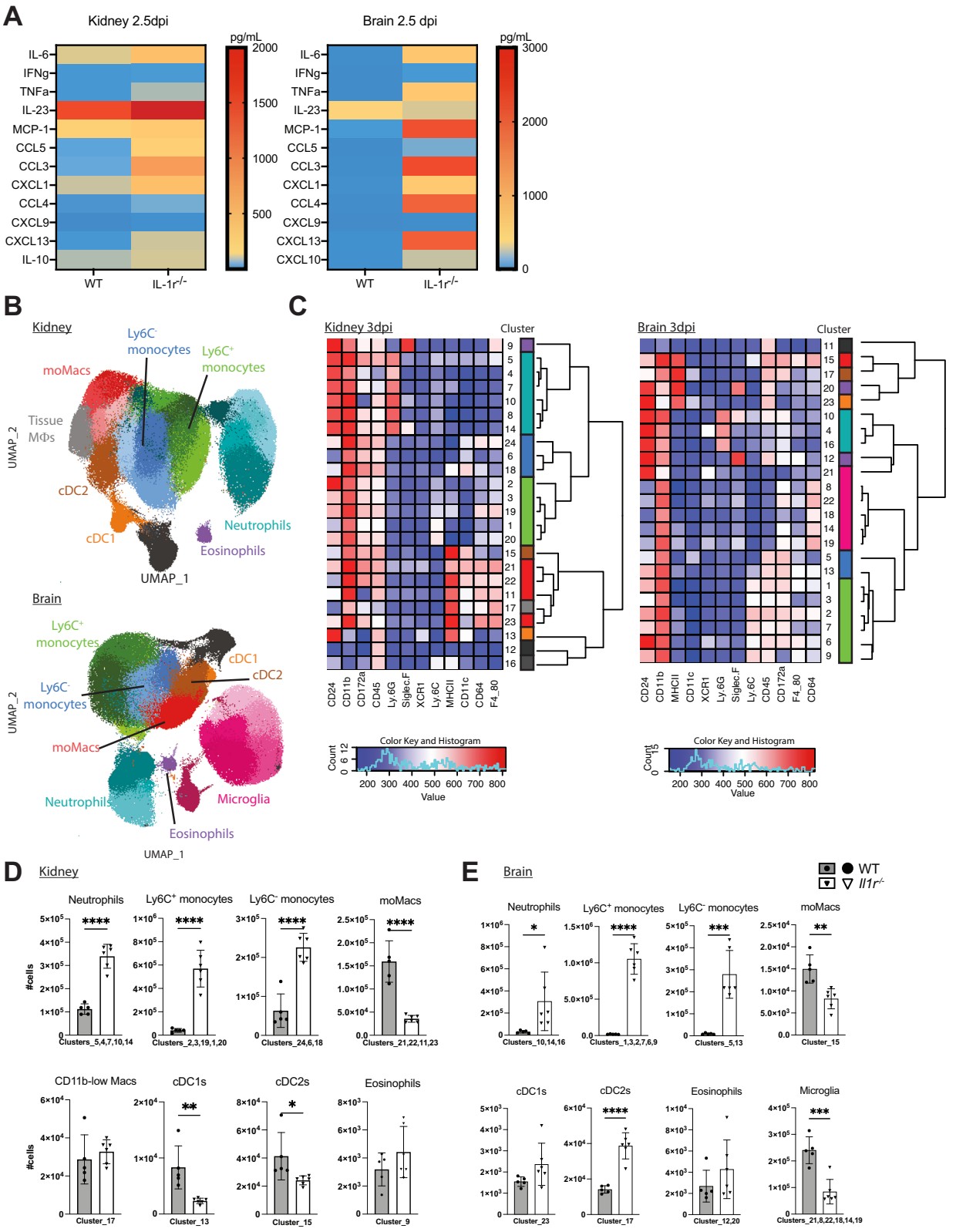

cells, while IL-1R signaling in hematopoietic cells plays a less critical role.

## IL-1R-signaling in kidney endothelial cells contributes to the control of *C. albicans* independent of neutrophils

IL-1R triggering induces a signaling cascade resulting in NF-κB activation[30]. To identify the non-hematopoietic cell types in the kidney and brain that contribute to IL-1R-mediated defense against *C. albicans*, we measured NF-κB activation in response to IL-1β stimulation as a proxy for IL-1R expression. To this end, we used recently generated NF-κB reporter (KappaBle) mice, allowing single-cell detection of NF-κB transcriptional activity monitored by expression of a destabilized EGFP[31]. Intravenous injection of IL-1β into naïve KappaBle mice resulted in a significant elevation of GFP expression after

**Fig. 2 | *Il1r1⁻/⁻* mice mount an exacerbated inflammatory response. A** Levels of inflammatory cytokines and chemokines in kidney and brain lysates from *WT* and *Il1r1⁻/⁻* mice, 2.5 days after infection with $10^5$ CFU *C. albicans*. n = 3/4 mice per group, mean values are indicated. Statistical test: Two-sided multiple unpaired t tests for each cytokine, Two stage step up (Benjamini, Krieger and Yekutieli), desired FDR (false discovery rate) = 1.00%. **B** Clustering using UMAP of flow cytometry data, obtained from kidney and brain of *WT* and *Il1r1⁻/⁻* mice, 3 days after infection with $10^5$ CFU *C. albicans*. Multi dimentional data was aquired from cells stained with a 14 antibody panel and analysed in Flow-Jo using the UMAP algorithm. **C** Heatmaps, visualizing mean expression of markers by 23 clusters, calculated using the Phenograph algorithm on data obtained from (**B**). Clusters are sorted acording to similarity and further sub-grouped and anotated based on the literature. Color coding allows visualization of the cell types in the UMAP plots in (**B**). **D**, **E** Quantification of the cell numbers in *WT* vs. *Il1r1⁻/⁻* mice. n = 5 (*WT*) and n = 6 (*Il1r1⁻/⁻*) mice per group. (Representative of 2 independent repeat). Statistical test: Two-sided unpaired t test for each cluster group. P < 0.05 marked with *, P < 0.01 marked with **, P < 0.001 marked with ***, P < 0.0001 marked with ****.

4–5 h in kidney and brain cells compared to PBS-treated KappaBle controls. The GFP-positive cells in the kidney could be characterized as CD45⁻CD31⁺ endothelial cells and CD45⁻EpCam⁺ epithelial cells, as well as other non-hematopoietic CD45⁻ cells that are CD31⁺EpCam⁺ double-positive and CD31⁻EpCam⁻ double negative (Fig. 4C and supplementary Fig. 4A). In the brain, we found IL-1R-signaling mediated NF-κB activation in border-associated macrophages (BAM) (GFP⁺CD45⁺CD206⁺), astrocytes (GFP⁺CD45⁻ACSA-2⁺) and in undefined CD45⁻CD31⁻EpCam⁻ cells, but not in endothelial and epithelial cells (Fig. 4D and supplementary Fig. 4B, C).

To assess whether IL-1R-signaling in endothelial cells is critical for protection from systemic candidiasis, we generated mice allowing inducible deletion of *Il1r1*, specifically in endothelial cells. To this end, we crossed *Il1r1^flox* mice[32] to *Pdgfb^icreERT2* mice[33]. We treated *Pdgfb^icreERT2 Il1r1^fl/fl* mice and *Il1r1^fl/fl* littermates with Tamoxifen on five consecutive days and infected the mice with *C. albicans* four days after the last treatment. We confirmed efficient deletion of the *Il1r1* gene in endothelial cells from the kidney and the brain by qPCR (supplementary Fig. 5B). On day 3 p.i., the fungal burden was up to 10-fold higher in the kidneys of *Pdgfb^cre Il1r1^fl/fl* mice compared to their *Il1r1^fl/fl* littermates. Intriguingly, a similar effect was not seen in the brain (Fig. 4E). Like global *Il1r1⁻/⁻* mice, at 3 days p.i., *Pdgfb^icreERT2 Il1r1^fl/fl* mice showed elevated number of neutrophils and inflammatory monocytes in the kidney, relatively to their *Il1r1^fl/fl* littermates (Supplementary Fig. 5C). These results demonstrate that lack of IL-1R on endothelial cells contributes to dramatically elevated (up to 1000-fold) fungal load observed in the kidney of global *Il1r1⁻/⁻* mice at day 3 p.i.

Since we have established that IL-1R is needed for the very early recruitment of neutrophils to the site of infection (Fig. 3E), we were interested in investigating whether this process depends on endothelial IL-1R. Therefore, we quantified neutrophil infiltration at 8 h p.i. in *Pdgfb^icreERT2 Il1r1^fl/fl* mice, compared to their cre negative littermates and to *Il1r1⁻/⁻* mice. Surprisingly, neutrophil infiltration in *Pdgfb^icreERT2 Il1r1^fl/fl* mice was comparable to the *Il1r1^fl/fl* mice (Fig. 4F), suggesting that neutrophil recruitment upon *Candida* infection is independent of endothelial IL-1R. We next treated naive *Il1r1^fl/fl* and *Pdgfb^icreERT2 Il1r1^fl/fl* mice with IL-1β intravenously and quantified the myeloid cells in the kidney and brain after 4 h. Interestingly, IL-1β treatment was sufficient to induce neutrophil infiltration to the kidney and brain of wild-type (*Il1r1^fl/fl*) mice, and a comparable recruitment was observed in *Pdgfb^icreERT2 Il1r1^fl/fl* mice (Fig. 4G), suggesting that the recruitment of myeloid cells is independent of endothelial expression of IL-1R.

These data suggest that IL-1R signaling in kidney endothelial cells contributes to fungal defense independent of neutrophil recruitment. However, IL-1R signaling in other non-hematopoietic cells in the kidney and brain is involved in protecting against *Candida* overgrowth.

### *C. albicans* infection induces massive transcriptional upregulation of uniform pathways across all kidney cell types in the absence of IL-1R

To uncover the transcriptional response to *C. albicans* infection in *WT* mice and differential gene expression in the absence of IL-1R at the single cell level, we performed single-nucleus RNA sequencing (snRNA-seq) on the kidneys of *Il1r1⁺/⁻* and *Il1r1⁻/⁻* mice at 8 h p.i. and naive

controls. We opted for snRNA-seq as this method has been shown to supply comparable gene detection to single-cell RNAseq[34], while reducing transcriptional biases that originate in the tissue dissociation process and allowing better annotation of rare kidney cell types[35]. We sequenced a total of 69,820 kidney cells. Following filtering steps, 47,243 cells passed QC with ~2270 genes and ~4070 reads on average per cell (Supplementary Fig. 6A, B, Methods QC). Using previously published kidney datasets[35,36], we annotated our dataset to the 20 unique kidney cell types recapitulating most of the nephron-forming cells (Fig. 5A; Supplementary Fig. 6C, D). We did not observe differences in the abundance of a particular cell type, except for an increased proportion of ascending-loop of Henle/proximal tubule cells (CTAL-PT cells). This cell cluster showed similarity both to the proximal tubules and thick ascending loop of Henle in the cortex (Fig. 5A; Supplementary Fig. 6C). Using data imputation, we were able to detect expression of the *Il1r1* gene in at least 5 cell types, with the highest levels in mesangial cells (MC), endothelial cells (EC), connective tubule (CNT) cells, principal cells (PC), and intercalated cells (Supplementary Fig. 6E). However, due to limited sensitivity of snRNA-seq and low abundance detection of IL-1R in the dataset, IL-1R is probably expressed at a low level in other kidney cell types.

Notably, *Candida* infection in *WT* mice resulted in relatively minor transcriptional changes (~50 genes up or down), with endothelial cells and cell types localized in the loop of Henle, showing more inhibition of gene expression (Fig. 5B). In contrast, infection of *Il1r1⁻/⁻* mice induced a massive upregulation of genes in every kidney cell type compared to infected *WT* and naive *Il1r1⁻/⁻* mice, in particular the cells in the proximal tubule (i.e., PT-S1 to S3) and in the loop of Henle (i.e., CTAL1/2), which showed upregulation of 400–500 genes (Figs. 5B, 6A). Genes upregulated in the infected *Il1r1⁻/⁻* mice were highly overlapping between different kidney cell types, suggesting that in these mice, infection causes an upregulation of non-cell-specific gene programs. In contrast, downregulated genes upon infection were rather cell-specific (Fig. 5C).

To determine whether our observations of the transcriptional response could be secondary to kidney damage, we analyzed the kidney function over the course of infection. We collected blood and urine of infected *Il1r1⁻/⁻* and control mice at 10 h, 24 h and 48 h p.i. (Fig. 5D, E). The analysis revealed high blood urea and macro-albuminuria (ACR > 300 mg/g) at 48 h p.i. in the *Il1r1⁻/⁻* mice, indicating that they suffered from severe kidney damage at that time point. However, no signs of kidney damage were observed at 10 h p.i., and only slight microalbuminuria (ACR > 30 mg/g) was seen at 24 h p.i. in the *Il1r1⁻/⁻* mice. Additionally, the fungal burden in the kidneys of *Il1r1⁻/⁻* and control mice at 8 h p.i. is not different (supplementary Fig. 7A). These results confirm that the transcriptional phenotype we observe at 8 h p.i. is likely a direct consequence of the absence of IL-1R and the presence of *C. albicans*, and it is not secondary to kidney damage, or differences in fungal burden.

### Dysregulated metabolism in kidney cells from *C. albicans* infected *Il1r1*-deficient mice

Analysis of the differentially expressed genes and pathways upon infection in *Il1r1⁻/⁻* and in control mice using Metascape[37] uncovered an interesting transcriptional response in mice lacking IL-1R (Fig. 6A).

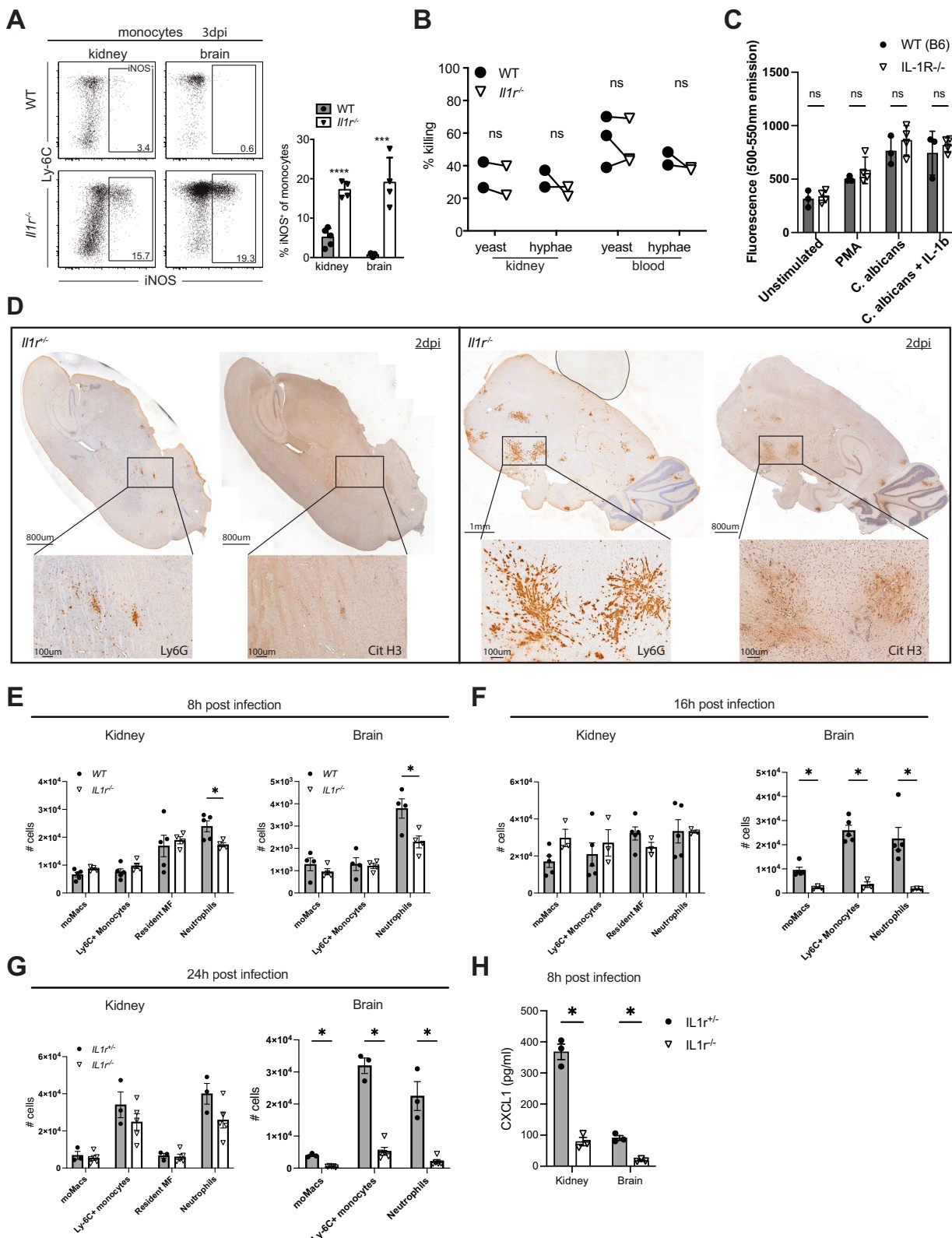

While *Candida* infection of *Il1r1*-sufficient mice mainly results in downregulation of pathways such as respiratory electron transport and sodium transport in most cell types, infection of *Il1r1*-deficient mice results in a strong upregulation of aerobic respiration in all cell types, except for macrophages (Fig. 6A). Additionally, there is what seems to be a metabolic crisis, as indicated by the upregulation of ribosomal genes, aerobic respiration, carbon and amino-acid

metabolism, glycolysis and gluconeogenesis, oxidative stress and redox pathway, fatty acid degradation and other (Fig. 6A, C supplementary Fig. 7B, C). These observations were consistent with the analysis of signature scores of KEGG metabolic pathways (Fig. 6C, supplementary Fig. 7C). Oxidative phosphorylation is one of the highest upregulated pathways. It is markedly up in all the kidney cell types except macrophages in infected *Il1r1*-/- mice (Fig. 6C). A multitude

**Fig. 3 | Neutrophils and inflammatory monocytes are functionality intact in _Il1r1_$^{-/-}$ mice. A** Flow cytometric analysis of intracellular expression of iNOS by monocytes, isolated from kidney and brain of _WT_ and _Il1r1_$^{-/-}$ mice, infected with $10^5$ CFU _C. albicans_, at 3 days p.i.. (n = 5(_WT_), n = 4(_Il1r1_$^{-/-}$) mice per group). Left: representative dot plot pregated on monocytes (gating is described in Supplementary Fig. 1A). Right: each dot represents a mouse, bars indicate mean values (representative experiment out of 5 repeats is shown), Statistical test: Two-sided unpaired t test per organ. Data are presented as mean values with +/- SD. **B** Neutrophils were sorted from kidneys or blood of infected _WT_ and _Il1r1_$^{-/-}$ mice at 3 days p.i. and co-cultured in-vitro with opsonised _C. albicans_ yeast or hyphae at 5:1 effector:target ratio. Each dot represents mean killing of _C. albicans_ in an independent experiment, data from the same experiment are connected with a line (n = 2/n = 3 mice per group in each experiment). Statistical test: Two-sided paired t test per organ. **C** In-vitro NET formation by bone marrow neutrophils from _WT_ and _Il1r1_$^{-/-}$ mice. Neutrophils were isolated from the bone marrow of uninfected mice, and stimulated for 2.5 h with either PMA, _C. albicans_ (1:1 cell to cell ratio) or _C. albicans_ (1:1) + IL-1b. NET formation was assessed by Sytox green staining (arbitrary units of fluorescence emission at 500-550 nm are used). (n = 3(_WT_), n = 4(_Il1r1_$^{-/-}$) mice per group, representative of two independent repeats), each dot represents an animal. Plating was performed with 3 technical replicates per condition.

Statistical test: Two-sided multiple unpaired t tests, with no correction for multiple comparisons. Data are presented as mean values with +/- SD. **D** In-vivo NET formation by neutrophils in the brain of _Il1r1_$^{+/-}$ and _Il1r1_$^{-/-}$ mice, 2 days p.i. with $10^5$ CFU _C. albicans_. Subsequent slides from brains were stained with either Ly6G or Citrulinated Histone H3 (Cit H3) antibodies. Left panel: _Il1r1_$^{+/-}$ mouse, Right panel: _Il1r1_$^{-/-}$ mouse. Within each panel: left section: Ly6G immunostaining, right section: Cit H3 immunostaining. (A representative picture from a single experiment with n = 3 mice per group is shown). **E–G** Analysis of myeloid cells in kidney and brain of infected _WT_ and _Il1r1_$^{-/-}$ mice 8, 16 and 24 h after infection with $10^5$ CFU _C. albicans_. Gating strategy is visualized in supplementary Fig. 1. (E: n = 5(_WT_), n = 4(_Il1r1_$^{-/-}$) mice per group, F: n = 5(_WT_), n = 3(_Il1r1_$^{-/-}$) mice per group, G: n = 3(_WT_), n = 5(_Il1r1_$^{-/-}$) mice per group, all time points are performed as independent experiments), data from representative experiments is displayed. Statistical test: Two sided multiple unpaired t tests, with no correction for multiple comparisons. Data are presented as mean values with +/- SEM. **H** CXCL1 levels in kidney and brain lysates of _Il1r1_$^{+/-}$ and _Il1r1_$^{-/-}$ mice 8 h after infection with $10^5$ CFU _C. albicans_. (n = 3 mice per group). Statistical test: Two-sided multiple unpaired t tests, Two-stage step-up (Benjamini, Krieger, and Yekutieli), Desired FDR 1.00%. Data are presented as mean values with +/- SEM.

of genes encoding for subunits of complexes I, III, IV, and V of the electron transport chain (ETC) were up in all kidney cells within 8 h after infection of _Il1r1_$^{-/-}$ mice compared to controls (supplementary Fig. 7E). In addition, we see elevation in genes involved in glycolysis (Fig. 6D) as well as genes involved in targeting and import of proteins into the mitochondria, such as Timm50[38] and Tomm20[39]; Prohibitin (Phb), which forms a complex on the mitochondrial membrane and functions as a protein stabilizing chaperon[40,41]; and the mitochondrial citrate carrier Slc25a1 (Fig. 6E). These observations indicate an activation of mitochondrial activity and respiration.

Pathway activity analysis with PROGENy database (Fig. 6B) showed reduced activity of a few main pathways upon infection of _Il1r1_$^{-/-}$ mice, such as NF-κB, Wnt, and PI3K. A reduction in NF-κB pathway activity is consistent with the absence of IL-1R and is observed to a significant extent in 11 cell types out of 18, suggesting the importance of IL-1R in most kidney cell types.

Interestingly, this analysis highlighted an increase in the hypoxia pathway for the majority of the cell types in the infected _Il1r1_$^{-/-}$ mice (Fig. 6B). Transcription factor specific analysis using DoRothEA spotlighted increased HIF-1α, PPARα, Myc and Sp1 regulon activity in _Il1r1_$^{-/-}$ kidney cells (Supplementary Fig. 7D), suggesting elevated transcriptional regulation of hypoxia-related target genes.

Under conditions of very low oxygen content in a tissue (hypoxia), cells increase glycolysis to promote survival. This metabolic switch is initiated by stabilization and nuclear translocation of hypoxia inducible factor 1 (HIF-1), which induces upregulation of several enzymes of glycolysis[42]. Indeed, all kidney cell types from _Candida_-infected _Il1r1_-deficient mice showed strikingly increased transcription of almost every enzyme involved in the breakdown of glucose to pyruvate including 6-phophofructokinase I (_Pfkl_), aldolase A, B and C (_Aldoa, Aldob, Aldoc_), triosephoshphate isomerase (_Tpi1_), glyceraldehyde 3-phosphate dehydrogenase (_Gapdh_), phosphoglycerate kinase (_Pgk1_), phosphoglycerate mutase (_Pgam1_), enolase (_Eno1_), and pyruvate kinase M (_Pkm_) (Fig. 6D). Moreover, there was a transcriptional increase of lactate dehydrogenase (_Ldha_) mediating the conversion of pyruvate to lactate and back. Notably, promotor and enhancer regions of the genes _Pfkl, Aldoa, Pgk1, Eno1, and Ldha_ contain HIF-1 binding sites (HRE), suggesting that HIF directly upregulates them in response to hypoxia[43,44].

Interestingly, in parallel to the ramp-up of glycolysis genes, there was a significant increase in the transcription of pyruvate dehydrogenases a and b (_Pdha1_ and _Pdhb_) and dihydrolipoamide dehydrogenase (_Dld_), which together catalyze the conversion of pyruvate to acetyl-CoA and $CO_2$ in the mitochondria, thereby providing

a substrate for the TCA cycle. The purpose of hypoxia-driven HIF-1α guided transcriptional activity is to reduce oxygen consumption by the cell via inhibition of oxidative phosphorylation and boosting of glycolysis for energy needs. In the absence of _Il1r1_$^{-/-}$, however, HIF-1α is activated but seems insufficient for inhibiting oxidative phosphorylation.

To assess the possibility that IL-1R intrinsically influences kidney cell metabolism, we used a human kidney podocyte cell line (differentiated AB8/13 cells[45,46]). We performed an ATP assay on the Agilent Seahorse XF96 analyzer, which allows the precise quantification of the rate of adenosine triphosphate (ATP) production from glycolysis and mitochondria in live cells[47]. Before analysis, we treated the cells overnight with either IL-1β or the IL-1R antagonist Anakinra (Fig. 7A, B). The assay revealed that podocytes increase the glycolytic source of ATP upon IL-1β treatment, while Anakinra treatment caused a significant increase of the mitochondrial (oxidative) source of ATP in the cells (Fig. 7C). Quantification of the mitochondrial ATP fraction over glycolytic ATP fraction demonstrated a significant effect of IL-1R signaling on the cell ATP production. In other words, while activation of IL-1R causes increased glycolytic ATP, blocking of IL-1R (as is happening in _Il1r1_$^{-/-}$ mice) causes increased respiratory ATP.

These results demonstrate that IL-1R intrinsically influences kidney cell metabolism, specifically shifting away from respiratory ATP production. Respectively, in _Il1r1_$^{-/-}$ mice the kidney cells seem to respond to _C. albicans_ infection by increased oxidative respiration.

## IL-1R1 protects from rapid kidney hypoxia upon _C. albicans_ infection

The kidney is one of the best-perfused organs. However, it is not very efficient in extracting oxygen from the blood (~10–20% only from delivered oxygen)[48]. This makes the kidney highly vulnerable to hypoxia. To confirm hypoxia in the kidney, we used Pimonidazole, which creates bonds with cellular macromolecules when the oxygen levels are under 1.3%. We injected Pimonidazole into _Il1r1_$^{-/-}$ mice and _Il1r1_$^{+/-}$ controls 24 h p.i. and visualized hypoxia in the kidneys. We used anti-pimonidazole and fluorescent secondary antibodies to visualize the Pimonidazole-positive areas. We then used a _C. albicans_-specific antibody to visualize the fungus. Since both antibodies have been generated in the same species, we could not co-stain the same slides. Thus, we stained subsequent slides (10μm apart) and approximated the co-localization of hypoxia and _C. albicans_ (Fig. 7D). Consistent with the transcriptional analysis, the kidneys of _Il1r1_$^{-/-}$ mice had significantly more hypoxic areas than the _Il1r1_$^{+/-}$ controls (Fig. 7E).

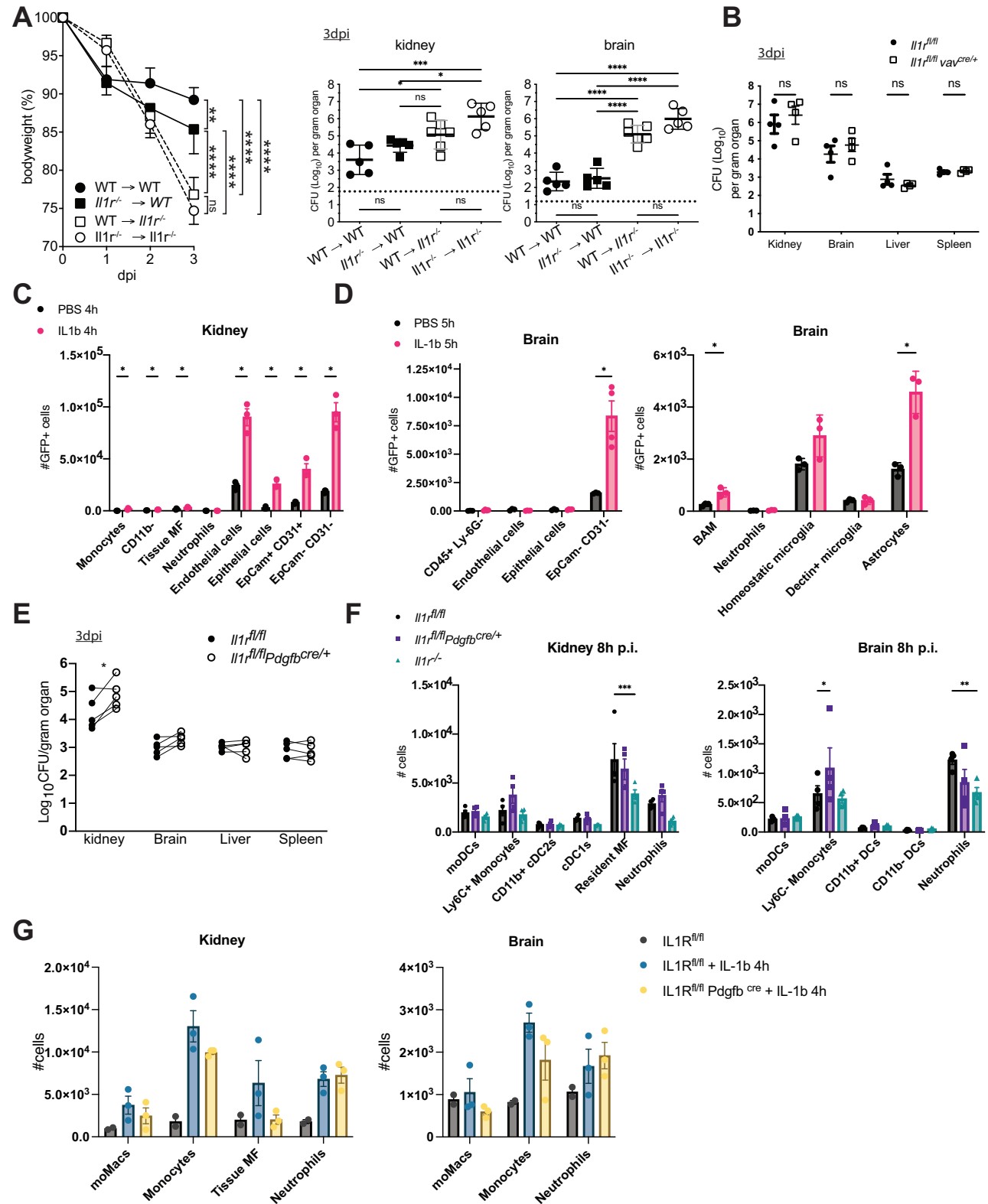

Interestingly, we found co-localization of hypoxic regions and infection foci (Fig. 7D) more in the *Il1r1*[-/-] mice than in the *Il1r1*[+/-] controls. This observation suggests that *C. albicans* itself could contribute to the creation of hypoxia by overconsumption of oxygen. However, since IL-1R seems to prevent most of the hypoxia, as seen in *Il1r1*[+/-] control mice, we believe that the kidney cells have a significant role in its exacerbation. We suggest that the metabolic response of the kidney cells to the infection in the absence of IL-1R exacerbates the depletion

of oxygen in the infection foci and contributes to the creation of local hypoxia, which in turn activates HIF-1α. The HIF-1α activation, however, is not enough to combat the hypoxia.

**Hypoxia promotes fungal invasiveness**

*C. albicans* rapidly changes its morphology in response to environmental cues such as nutrient availability, temperature, $CO_2$ and $O_2$ levels. At higher temperatures (37 °C, compared to room temperature)

**Fig. 4 | IL-1 acts via non-hematopoietic cells in the immune defense against *C. albicans*. A** Reciprocal bone marrow chimeras were generated from *WT* and *Il1r1*[-/-] mice, and infected with $10^5$ CFU *C. albicans*, 10–12 weeks after reconstitution. Left: Bodyweight loss during the infection. Right: Fungal burden at 3 days p.i.. Each dot represents an animal. (n = 5 mice per group, representative of two independent repeats) Statistical analysis was performed using one way ANOVA with Bonferroni's multiple comparisons. P < 0.05 marked with *, P < 0.01 marked with **, P < 0.001 marked with ***, P < 0.0001 marked with ****. Data are presented as mean values with +/- SD. **B** Fungal burden in *Il1r1*[fl/fl] *vav*[cre] and *Il1r1*[fl/fl] mice at 3 days after infection with $10^5$ CFU *C. albicans*. (n = 4 mice per group, two independent repeats were performed) Statistical test: Two-sided multiple unpaired t tests per organ, with no correction for multiple comparisons. **C, D** Quantification of GFP[+] cells in kidney and brain of KappaBle mice, 4–5 h after i.v. injection of IL-1b or PBS as control. (n = 3 **C**, n = 3/4 **D** mice per group, data from a representative experiment of two independent repeats are shown). Gating of cells is described in supplementary Fig. 4. Statistical test: Two-sided multiple unpaired t tests, with no correction for multiple comparisons. P < 0.05 marked with *. Data are presented

as mean values with +/- SEM. **E** Compiled data of fungal burden of *Il1r1*[fl/fl] *Pdgfb*[creERT2] and *Il1r1*[fl/fl] mice, at 3 days p.i. with $10^5$ CFU *C. albicans*. Inducible cre recombinase was activated by 5 consecutive i.p. injection of Tamoxifen. Both groups recieved the same treatment, and were infected 4 days after the last Tamoxifen injection. Data from 5 independent experiments are shown, (n = 3–5 mice per group). Data from the same experiment are connected with a line. Statistical test: Multiple paired t tests, with no correction for multiple comparisons. P[kidney]= 0.036784. **F** Quantification of myeloid cell infiltration into kidney and brain, 8 h after infection of *Il1r1*[fl/fl], *Il1r1*[fl/fl] *Pdgfb*[creERT2] and *Il1r1*[-/-] mice with $10^5$ CFU *C. albicans*. Gating according to the scheme in supplementary Fig. 1. (n = 4 mice per group). Statistical test: 2-way ANOVA with Tukey's multiple comparisons test. P < 0.05 marked with *, P < 0.01 marked with **, P < 0.001 marked with ***. Data are presented as mean values with +/- SEM. **G** Quantification of myeloid cell infiltration into kidney and brain, 4 h after i.v. injection of IL-1b or PBS, into *Il1r1*[fl/fl] and *Il1r1*[fl/fl] *Pdgfb*[creERT2] mice. (n = 3 mice (*Il1r1*[fl/fl] + Il-1b, *Il1r1*[fl/fl] *Pdgfb*[creERT2] + Il-1b), n = 2 mice (*Il1r1*[fl/fl] + *PBS*)). No statistical test was used, as data was not large enough. Data are presented as mean values with +/- SEM.

*C. albicans* readily creates hyphae and upregulates virulence genes[49]. This possibly happens when the fungi enter the host environment. $CO_2$ and oxygen availability have also been shown to affect *C. albicans* virulence and hyphae formation[50,51].

We asked whether the hypoxic environment in the kidneys of *Il1r1*[-/-] mice during candidiasis can promote fungal hyphae formation. Since Pimonidazole staining is positive at oxygen levels below 1.3%, we decided to grow *C. albicans* in YPD agar in similar levels of hypoxia (1% oxygen) or normoxia in 37 °C, 5% $CO_2$, and visualized the colonies. As predicted, while in normoxic conditions *C. albicans* did not create any hyphae, hypoxia promoted significant hyphae formation (Fig. 7F).

Since hyphae formation is crucial for *C. albicans'* pathogenicity[52], we suggest that the hypoxia that is created in the infection foci in the absence of IL-1R results in higher invasiveness of *C. albicans*, exacerbating the infection.

## Discussion

Our study was initiated with the goal of investigating the mechanism by which IL-1R activation protects from systemic candidiasis. Using *IL1a*[-/-], *IL1b*[-/-], and *Il1r1*[-/-] mice, we demonstrated the crucial role of IL-1β and, to a minor extent, also IL-1α, for efficient inhibition of *C. albicans* growth in kidney and brain, in keeping with previous studies[19,20,53]. Infection of *Il1r1* KO/WT criss-cross bone marrow chimeras and *vav*[cre]*Il1r1*[fl/fl] mice revealed that IL-1R-signaling in non-hematopoietic cells plays the decisive role in protecting against candidiasis. Also, IL-1R was not required to generate a full-blown pro-inflammatory response to *Candida* infection, which was exaggerated in its absence. Neutrophils are key players in the innate response to systemic and mucosal *Candida* infection by production of ROS, antimicrobial peptides, and NETS[9]. IL-1 cytokines have been suggested to protect from mucosal and invasive candidiasis mainly by recruitment and induction of antifungal effector mechanisms of neutrophils through cell-intrinsic and cell-extrinsic IL-1R/MyD88-signaling[17,19,20,53]. Our results rule out a neutrophil-intrinsic role of IL-1R and argue against a defective neutrophil effector function in the absence of IL-1R when considering comparably efficient ex vivo killing of *Candida* yeast and hyphae with neutrophils harvested from the kidney of infected *Il1r1*[-/-] and control *WT* mice. Also, massive NETs and elevated neutrophil numbers were observed in the brains and kidneys of *Il1r1*[-/-] mice 2–3 days after infection, indicating no general defect in neutrophil recruitment and function.

However, consistent with a previous report[20], we did observe a 24 h delay in the recruitment of neutrophils to the brain in *Il1r1*[-/-] mice, which may cause a head-start of fungal growth that cannot be compensated for despite massive infiltration later. It is well established that IL-1 cytokines can guide neutrophil trafficking indirectly by induction of the CXCR2 ligands CXCL1 and CXCL2 in different cell types[54].

Drummond et al. showed that microglia drive neutrophil recruitment to the CNS by production of IL-1β and CXCL1 in response to the *Candida* toxin Candidalysin in the CNS, although it remained unclear whether IL-1R expression on microglia or other cells in the brain was essential for neutrophil migration[20]. In addition to reduced levels of CXCL1 in *Il1r1*[-/-] mice, we observed reduced levels of other chemokines, as well as adhesion and extravasation molecules in sequencing data obtained from the brain.

IL-1R is expressed by a variety of cells in the CNS[29]. While expression on endothelial cells was found to be necessary and sufficient for sickness behavior (such as reduced food intake, locomotion, and social behavior) in response to central IL-1, neutrophil recruitment was partially dependent on both endothelial and ventricular (choroid plexus and ependymal) IL-1R, and monocyte recruitment relied entirely on ventricular IL-1R[29]. Furthermore, endothelial IL-1R1 has been proposed to drive leukocyte infiltration to the CNS during autoimmune encephalitis[55]. Our results show that the absence of IL-1R on endothelial cells had neither an effect on the recruitment of neutrophils and monocytes to the brain, nor on fungal growth upon systemic infection with *Candida*. While several reports question IL-1R expression on microglia[29,56,57], genetic IL-1R ablation on microglia affects their self-renewal capacity and morphology in vivo[58,59], suggesting low-level but biologically relevant expression. Indeed, it has been suggested that fewer than 20 IL-1-receptors per cell are sufficient for signaling[60]. Using highly sensitive NF-κB reporter mice, we showed that IL-1β treatment triggered IL-1R-signaling in border-associated macrophages, astrocytes and to a lower extent in microglia in vivo. Considering a slight but statistically significant increase of fungal burden in the brain but not the kidney of *vav*[cre]*Il1r1*[fl/fl] mice, our results indicate a partial contribution of microglia and astrocytes IL-1R-signaling for protection from fungal growth in the brain.

While delayed neutrophil recruitment in global *Il1r1*-deficient mice may explain the defect in controlling exaggerated fungal growth in the brain, the situation in the kidney is different. Selective genetic ablation of IL-1R in endothelial cells increases fungal growth 5- to 10-fold (the increase in global KO is 100–1000-fold) compared to littermate controls. Yet neutrophil recruitment was not affected, indicating that endothelial IL-1R partially contributed to fungal defense in the kidney, independent of neutrophil infiltration.

To identify relevant cell types that respond to *Candida* infection and IL-1, we performed transcriptomics of the entire kidney at the single cell level at 8 h p.i., when differences in neutrophil recruitment but none in fungal burden between WT and KO mice were observed. Single nucleus RNA sequencing revealed 20 clusters that could be assigned to known cell types. We were surprised to see striking transcriptional differences between *Il1r1*[-/-] and *Il1r1*[+/-] mice so early upon infection. IL-1R-sufficient mice seem to downregulate the

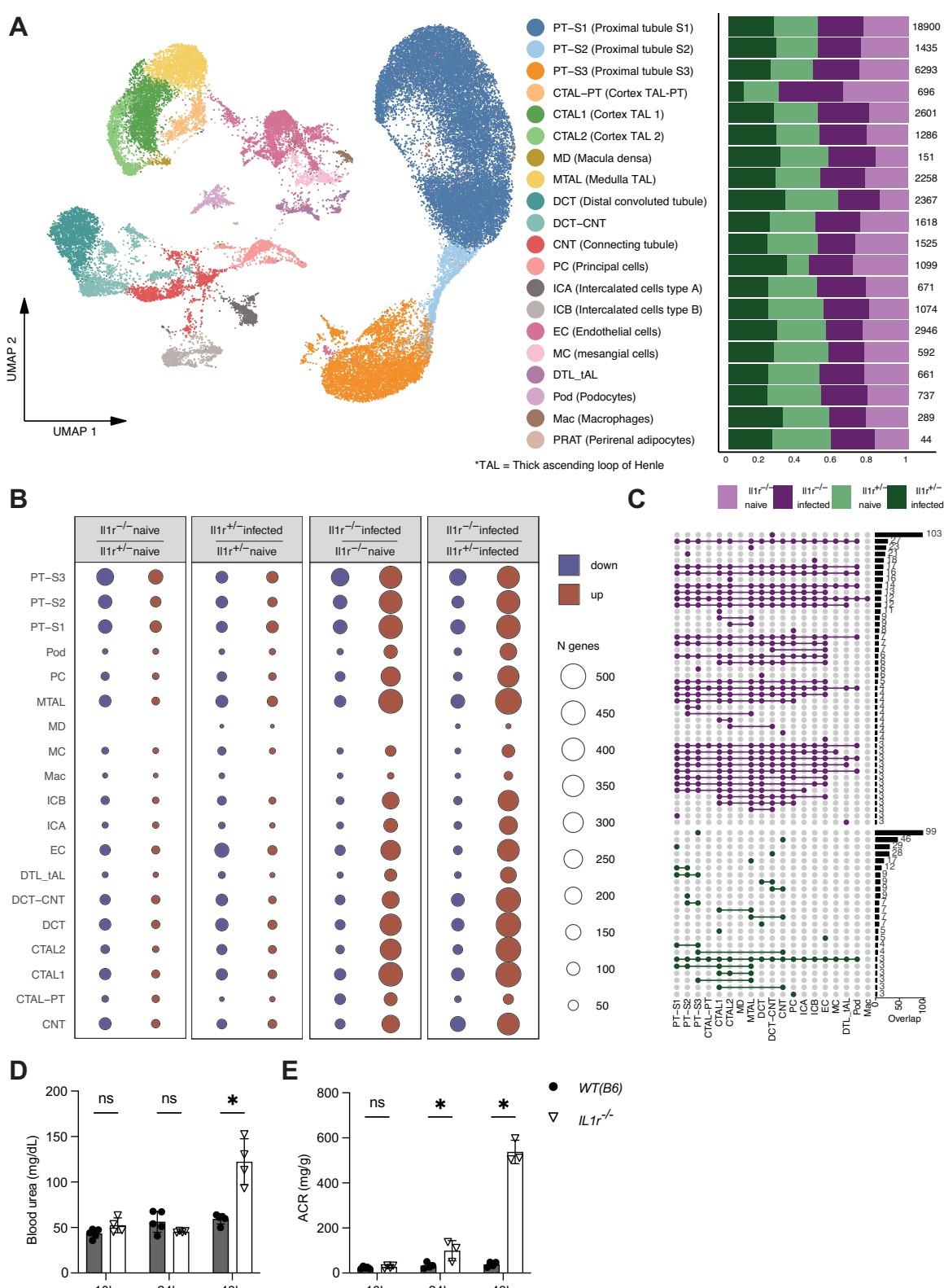

respiratory electron transport chain and sodium transport in 15 kidney cell types upon infection. In contrast, in IL-1R-deficient mice, every kidney resident cell type except macrophages massively increased expression of genes involved in protein synthesis, fatty acid degradation, glycolysis, redox stress, and oxidative phosphorylation, as if a metabolic dysregulation occurred uniformly across all cells.

Visualization of poorly oxygenated areas in histological sections confirmed that the kidneys of *Il1r1*[-/-] mice became rapidly hypoxic upon infection. This explains HIF-1α activation and enhanced expression of genes related to glycolysis[42]. However, HIF-1α activity appears insufficient to negatively regulate oxidative phosphorylation in *Il1r1*[-/-] kidney cells. IL-1R-signaling in adipocytes was previously shown to induce translocation of IRAK2 to the mitochondria, where it disrupts

**Fig. 5 | Massive and uniform transcriptional upregulation across all kidney cell types in *Il1r1*[-/-] mice immediately upon *C. albicans* infection.** Mice were infected with *C. albicans* ($10^5$ CFU) and kidneys harvested 8 h later and processed for single nuclear RNA sequencing as described in Methods. **A** UMAP of annotated celltypes with barplot for abundance per condition. PT-S1 - S1 segment of proximal tubule; PT-S2 - S2 segment of proximal tubule; PT-S3 - S3 segment of proximal tubule; CTAL-PT - thick ascending limb of loop of Henle in cortex associated with proximal tubules; CTAL1 - thick ascending limb of loop of Henle in cortex 1; CTAL2- thick ascending limb of loop of Henle in cortex 2; MD - macula densa; MTAL - thick ascending limb of loop of Henle in medulla; DCT - distal convoluted tubule; DCT-CNT - intermediate between DCT and CNT; CNT - connecting tubule; PC - principal cells; ICA - type A intercalated cells of collecting duct; ICB - type B intercalated cells of collecting duct; EC - endothelial cells; MC - mesangial cells; DTL_tAL - thin descending limb of loop of Henle + thin ascending limb of loop of Henle; Pod - podocytes; Mac - macrophages; PRAT - perirenal adipocytes; **B** Differentially expressed (DE) genes per pairwise comparison between conditions per celltype; **C** Shared genesets between celltype specific DE genes for *Il1r1*[-/-] infected (top) versus *Il1r1*[+/-] infected (bottom) comparison; **D** Urea and **E** ACR (albumin to creatinine ratio) levels in blood of *Il1r1*[+/-] and *Il1r1*[-/-] mice infected with $10^5$ CFU *C. albicans* at indicated time points after infection (n = 4/5 mice per group). Statistical test: Two-sided multiple unpaired t tests, set P value threshold = 0.05%. P[Urea 48 h] = 0.000908, P[ACR 24 h] = 0.033929, P[ACR 48 h] = 0.000007. Data are presented as mean values with +/- SD.

mitochondrial respiratory chain super-complex formation and thereby oxidative phosphorylation by interaction with prohibitin (Phb) and optic atrophy protein 1 (Opa1)[61]. We demonstrate that IL-1β treatment alone suppresses the mitochondrial ATP production in a kidney cell line, while Anakinra-mediated inhibition of IL-1R signaling, caused increased oxidative phosphorylation. This may explain the significantly upregulated oxidative phosphorylation genes in all kidney cells in the absence of IL-1R. However, how IL-1R-signaling causes transcriptional inhibition of OXPHOS and other mitochondrial genes in the kidney remains to be investigated.

In the absence of IL-1R, increased oxidative phosphorylation is associated with stronger hypoxia, suggesting the possibility that kidney cells consume more oxygen upon infection. IL-1 is critical to prevent hypoxia in the kidney microenvironment around *C. albicans* infection foci and, thereby, *Candida* invasiveness. Indeed, in agreement with others[50,51], we showed that a hypoxic environment of 1% oxygen promotes *Candida albicans* hyphae formation. It has been also suggested that in hypoxic conditions, *C. albicans* masks the cell wall component β-glucan, making it less exposed to C-type lectin receptors, the main pattern recognition receptors that sense *Candida* infection[62]. Evasion of recognition could potentially explain why increased neutrophil numbers in kidneys of *Il1r1*[-/-] mice observed from day 2 p.i fail to limit fungal growth despite normal killing activity ex vivo. Accumulation of neutrophils, on the other hand, may also be the consequence of hypoxia, which has been shown to promote their survival[63].

Per gram of tissue, the kidneys are the best-perfused organs of the body. However, the renal cortex and medulla are poorly oxygenated due to a few unique physiological and anatomical characteristics and, therefore, are highly susceptible to hypoxia[48,64,65]. Indeed, hypoxia is a prominent pathophysiological feature of acute kidney injury and chronic kidney diseases that have been associated with increased HIF-1α and/or HIF-2α expression[64,65]. HIF-1α has been detected in tubular epithelium and ECs following exposure to acute hypoxia. At the same time, HIF-2α is predominantly expressed in ECs and glomerular cells and in peritubular interstitial cells, where it protects from hypoxia-induced damage and anemia[66,67]. It is tempting to speculate that the relatively poor oxygenation in the kidney is the underlying reason for favorable *Candida* growth compared to the spleen and liver.

Our data indicate that upon infection with *C. albicans*, local IL-1R signaling inhibits a dysregulated metabolic response in the kidney cells and oxygen depletion in the infection foci. Our data suggest that increased hypoxia in the absence of IL-1R benefits the pathogen, making it more successful and promoting fungal overgrowth. On the other hand, in an IL-1R-sufficient environment, the kidney cells resist the metabolic shift, possibly protecting the microenvironment from becoming hypoxic, which can protect the kidney from *C. albicans*.

Taken together, our data reveal two distinct mechanisms by which IL-1R protects from early stages of invasive *C. albicans* infection dependent on the tissue. In the brain, IL-1 triggers the immediate recruitment of myeloid cells to the infected tissue, exerting a classical innate immune function. In the kidney, IL-1R has a non-conventional role independent of immune cell recruitment or activation. By acting on most non-hematopoietic cells of the kidney, IL-1R-signaling suppresses a metabolic response to the infection, which includes increased oxidative phosphorylation and, by that, combats the development of hypoxia in the microenvironment. The local hypoxia in *Il1r1*[-/-] mice allows the *Candida* to create more hyphae and become more pathogenic and immune evasive.

## Methods

### Animals

All animal work was performed in the ETH Phenomics Center (EPIC, Zürich, Switzerland). C57BL/6J (Jackson laboratory), B6 Cd45.1 (JAX 002014), *Il1r1*[-/- 68], *Il1a*[-/-; 69], *Il1b*[-/-69], *Il1r1*[fl/fl] mice[32] (JAX 028398), *vav*[cre/+70], Kappable[31] and *Pdgfb*[creERT2/+33] mice were kept in individually ventilated cages under specific pathogen free conditions and bred at the EPIC facility. *Il1r1*[fl/fl] mice were bred with *vav*[cre/+] mice and with *Pdgfb*[icreERT2/+] mice in house. The following mouse genotypes were bred as littermates: *Il1r1*[-/-] and *Il1r1*[+/-], *Il1r1*[fl/fl]*vav*[cre/+] and *Il1r1*[fl/fl]*vav*[+/+], and *Il1r1*[fl/fl]*pdgfb*[icreERT2/+] and *Il1r1*[fl/fl]*pdgfb*[+/+]. For experiments, age-matched mice were used at 8 to 12 weeks. Sex was not considered in the study design and both female and male mice were used interchangeably; only sex-matched animals were compared in each experiment. Animals were fed ad-libitum with chow diet (Granovit, 3437.PX.L15 M/R) and euthanized by exposure to $CO_2$). All experiments were approved by the local animal ethics committee (Kantonales Veterinäramt Zürich) by licenses for infection experiments (ZH134/18) and for organ removal of naïve animals (ZH104/2021). All experimental procedures were performed according to local guidelines (TschV, Zurich) and the Swiss animal protection law (TschG).

### Activation of inducible Cre recombinase

Il1r1[fl/fl]pdgfb[icreERT2/+] and Il1r1[fl/fl]pdgfb[+/+] mice were injected with 2 mg Tamoxifen (Sigma, T5648), dissolved in corn oil, i.p. for 5 consecutive days. 3–7 days after the last Tamoxifen injection the mice were infected and analyzed.

### Bone marrow chimera

Animals were sub-lethally irradiated with a total dosage of 950 RAD using an RS 2000 X-ray irradiator (Rad Source Technologies) and reconstituted one day later with freshly isolated bone marrow cells flushed from tibias and femurs of donor mice by intravenous injection. Mice were treated with antibiotics (0.024% Borgal, MSD Animal Health, in the drinking water) for 6 weeks and were used for infection experiments 10 to 12 weeks post reconstitution.

### Infection

*Candida albicans* (strain SC5314) was inoculated in YPD medium (Sigma-Aldrich) to an $OD_{600}$ of 0.1 and then grown for 16 h at 30 °C and 200 rpm. The *Candida* suspension was washed twice with PBS and the cell concentration was determined using a Neubauer chamber. Animals were infected intravenously by an injection of $10^5$ CFU *Candida*

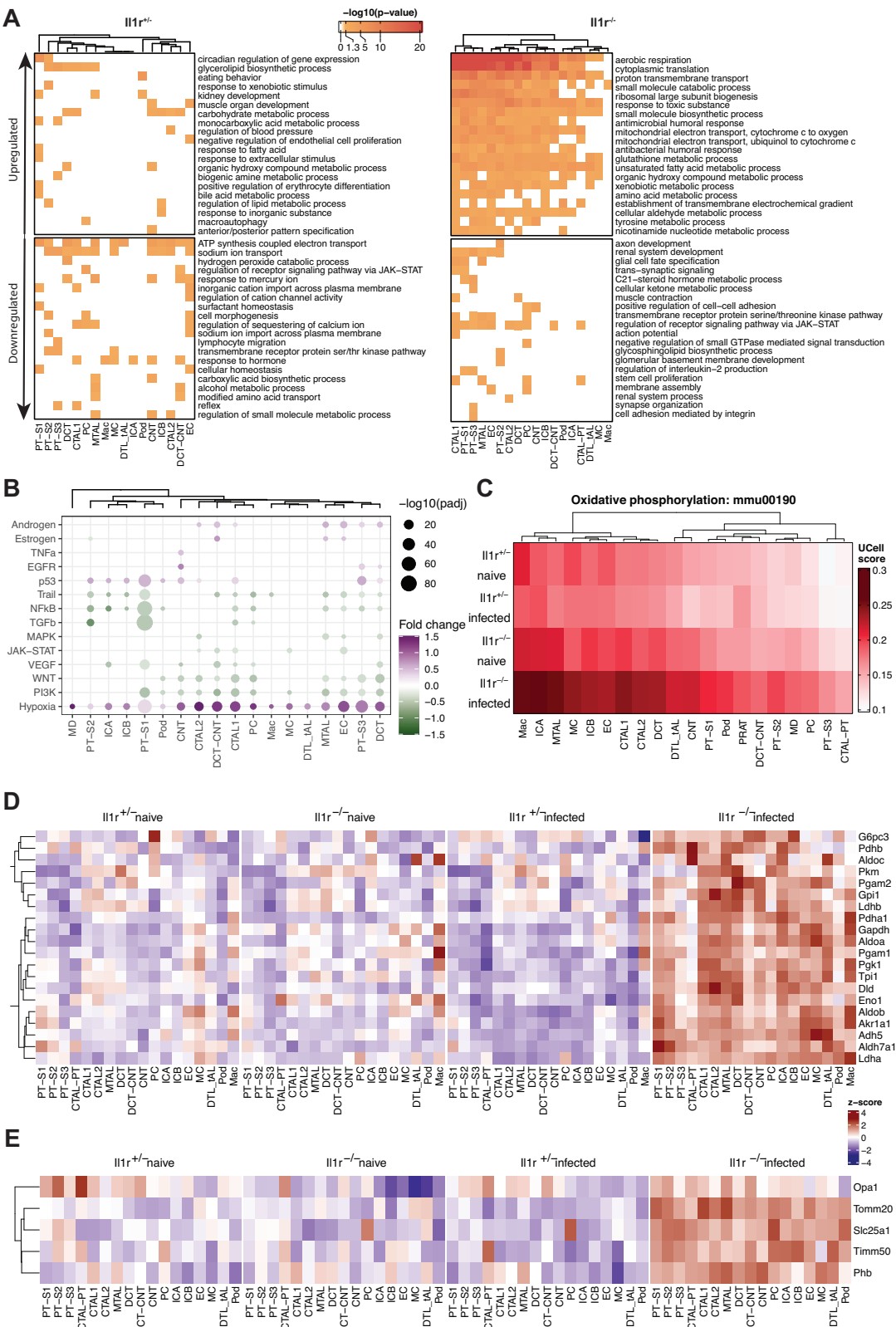

**Fig. 6 | Increase in oxidative phosphorylation upon infection with *C. albicans* for kidney cell types in *Il1r1⁻/⁻* mice. A** Combined celltype specific Metascape ontology analysis for *Il1r1⁻/⁻* (right) and *Il1r1⁺/⁻* (left) based on differentially expressed genes (adj. p < 0.05) for infected (top) versus naive (bottom) conditions. Statistical test: two-sided Wilcoxon rank sum test followed by Bonferroni p-value adjustment. **B** Differentially active (adj. p-value < 0.1) signaling pathways for *Il1r1⁻/⁻* infected versus *Il1r1⁺/⁻* infected. Statistical test: two-sided Wilcoxon rank sum test followed by Bonferroni p-value adjustment. **C** UCell enrichment scores for oxidative phosphorylation geneset signature. **D, E** Heatmaps showing expression of selected genes **D** from glycolysis pathway (mmu00010) and **E** from mitochondrial proteins per condition and cell type.

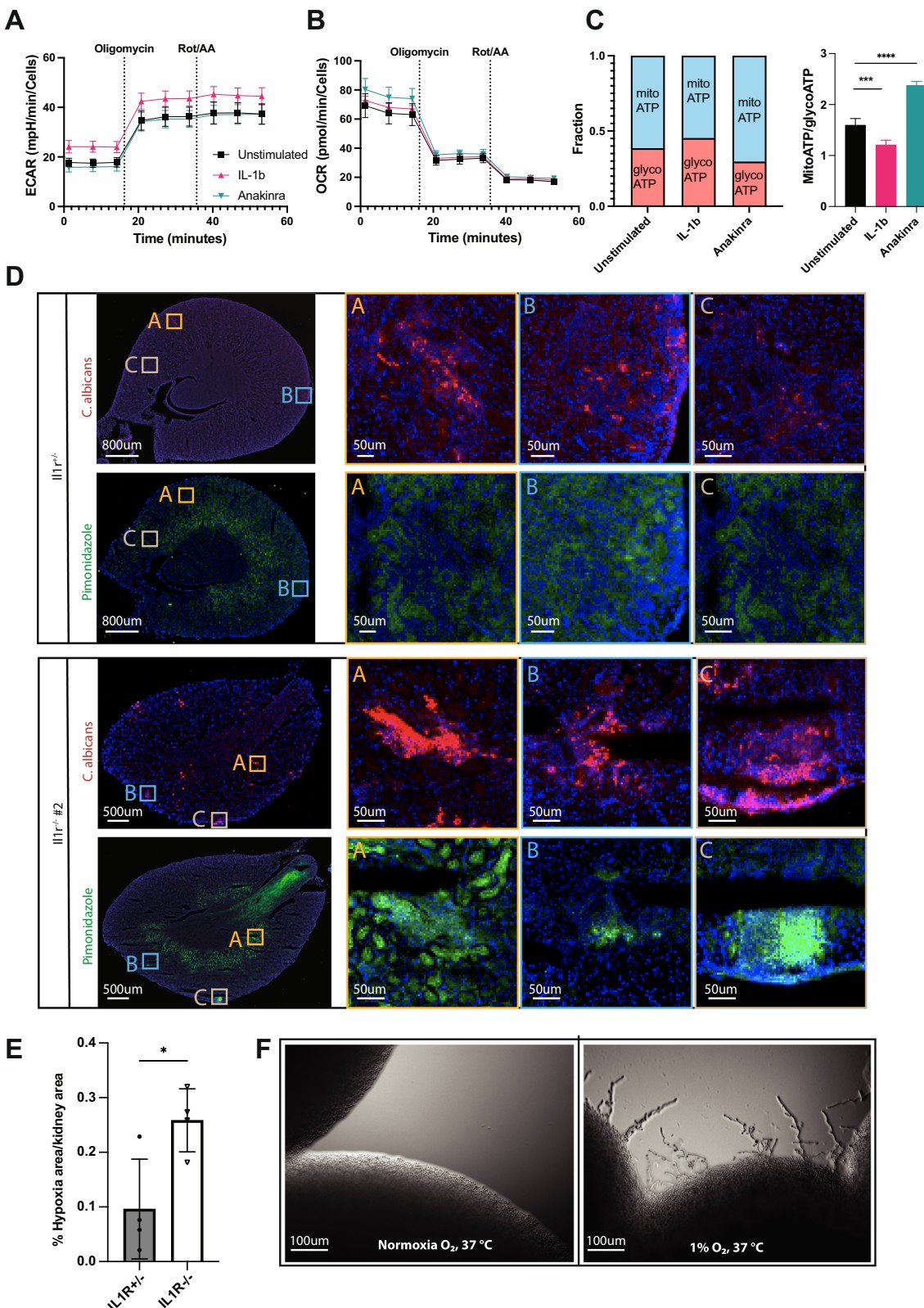

*albicans* into the lateral tail vein. The actual infection dosage was confirmed by plating an aliquot on YPD agar plates.

**Fungal burden**
Fungal burden was analysed at indicated time points. Organs were removed and homogenized in sterile PBS containing 0.05% Triton X-100 (Sigma-Aldrich) using a TissueLyser (Qiagen) for 10 min at 25 Hz.

Serial dilutions were plated on YPD agar containing Penicillin (100 U/mL) and Streptomycin (100 µg/mL) (Pen/Strep, Life Technologies). Colony forming units (CFU) were counted after incubation for 36 to 48 h at 30 °C and normalized to the weight of the organ. The limit of detection (LOD) correlates to one colony divided by the weight of the heaviest organ analysed in the experiment. For samples with a CFU value below the LOD, the value was set to LOD/2.

**Fig. 7 | Hypoxia in kidneys of *Il1r1-/-* mice promotes pathogenicity of *C. albicans*.**
**A** ECAR and OCR **B** values from a Seahorse XF96 ATP assay analysis of differentiated AB8/13 cells, treated overnight with either IL-1b, Anakinra or neither. 5 well per condition were analyzed (representative of two independent repeats is shown). Data are presented as mean values with +/- SD. **C** Calculated mitochondrial/glycolytic ATP source from the experiment in (**A**, **B**). Calculations were performed as described in the Seahorse XF user guide[47] (5 well per condition were analyzed, representative of two independent repeats is shown). Statistical test: two tailed unpaired t tests on MitoATP/GlycoATP, comparing stimulation to control. P[Il-1b] = 0.0005, P[Anakrin] < 0.0001. Data are presented as mean values with +/- SD. **D** Kidney sections from infected *Il1r1+/-* (top) and *Il1r1-/-* (bottom) mice, 24 h after infection with $10^5$ CFU *C. albicans* stained with antibodies against Pimonidazole (green) and *C. albicans* (red) and with Hoechst (blue). Merged images of 2 color-channels (Hoechst and Cy5) are shown. While both were stained using the same secondary antibody (fluorophore Cy5), different colors are chosen to display Pimonidazole and *C. albicans* staining. **E** Quantification of the high hypoxia areas from (**D**). Total area of high hypoxia was calculated using ImageJ2 (version 2.14.0) and normalized to the total area of the kidney section. To define high hypoxia, a color threshold was defined for green (Pimonidazole), with brightness above physiological hypoxia. All samples were analyzed using the same color and brightness thresholds. Each point represents an animal, n = 4. Statistical test: two tailed unpaired t test on normalized areas. P = 0.0237. Data are presented as mean values with +/- SD. **F** *C. albicans* colonies grown on YPD agar plates for 48 h, 37 °C, 5% $CO_2$, in normal atmospheric oxygen levels (left) and in hypoxia (1% $O_2$) (right). Imaging done by light microscopy. (A representative picture from a single experiment with n = 3 plates per condition is shown).

## Organ preparation for single cell suspensions

Single cell suspensions were obtained from kidney and brain at indicated time points after infection. Animals were perfused with 10 mL PBS. Kidney and brain were removed, minced and then digested for 60 min at 37 °C and 120 rpm in IMDM medium (Life Technolgies) containing digestion enzymes (for kidney: Collagenase IV (600 U/mL, Worthington), for brain: Collagenase D (0.02 mg/mL Roche) and DNaseI (0.02 mg/mL, Sigma-Aldrich)). Digested tissue was passed through a 70 μm cell strainer (Corning) and washed. All washing steps were performed at 4 °C in FACS buffer (PBS + 2% FCS) and with a speed of 300 g. For analysis and FACS sorting of CD45+ cells, leukocytes were enriched using a Percoll (GE Healthcare) gradient with centrifugation at 860 g for 30 min at room temperature (for kidney 40%/70% Percoll, for brain 30%/70% Percoll was used). After centrifugation, the leukocytes were collected from the interphase, washed and pelleted at 470 g and resuspended in FACS buffer. Lastly, all samples were filtered through a 35 μm cell strainer. Single cell suspensions were then stained with antibodies and analyzed or sorted using flow cytometry.

## Preparation of brain for single-cell suspensions enriched for microglia and astrocytes

Single-cell suspensions were obtained from brain tissue at the indicated time point after infection. Animals were perfused with 10 mL PBS. Brain was removed, minced, and then digested for 40 min at 37 °C and 120 rpm in HBSS medium, containing Collagenase IV (0.4 mg/mL) and DNAse I (0.2 mg/mL). after digestion. The tissue was homogenized using 5 mL syringes through G19 needles, by resuspending >5 times, and subsequently filtered through a 100 μm filter. Cells were centrifuged at 400 g > 15 min at 4 °C. To remove the myelin, the cell pellet was resuspended in 32% percoll and centrifuged at 1632 g 30 minutes at 4°C without break. The myelin ring was discarded, cells were washed with PBS and pelleted by centrifugation at 485 g for 25 min at 4 °C.

## Endothelial cell isolation

For endothelial cell isolation from brain tissue, mouse brains were removed without perfusion for endothelial cell isolation from brain. The cortex and hippocampus were isolated on ice and minced using a surgical blade. The minced tissue was digested in a 37 °C water bath for 20 min, in HBSS, containing Glucose (1.55 g/L, Sigma-Aldrich), Liberase DH (0.08 W U/mL, Roche) and DNase I (100 U/mL, Sigma-Aldrich). 10 min after start of digestion the samples were gently mixed using a glass Pasteur pipette (HuberLab). Digested tissues were then passed 20 times through a 20 G needle for homogenization. Homogenized tissued were filtered through a 40 μm filter, washed with PBS and pelleted at 300 g. Myelin removal was performed using Myelin Removal Beads (Miltenyi Biotec, 130-096-733) and LS columns (Miltenyi Biotec, 130-042-401) according to manufacturer's protocol, using MACS separators for magnetic cell isolation (Miltenyi Biotec). Myelin depleted samples were pelleted at 300 g, resuspended in blocking solution, containing PBS + 2% FCS and anti-CD16/CD32 mAb (2.4G2, homemade) to block Fc gamma receptors and filtered through a 35 μm cell strainer.

Staining for surface markers was performed for 30 min at 4 °C, using PerCP/Cyanine5.5 anti-mouse CD326 (Ep-CAM) Antibody (BioLegend), APC anti-mouse CD31 Antibody (BioLegend) and PE/Cyanine7 anti-mouse CD45 Antibody (BioLegend). For exclusion of dead cells, SYTOX™ Blue was added to the stained single cell suspensions during sorting. Sorting was done using BD FACS Aria III. Endothelial cells were defined as CD31+CD45-EpCam-Sytox-, and sorted into FCS treated tube at 4 °C. Sorted cells were immediately lysed using RNAzol RT (Sigma-Aldrich) and stored at −80 °C until analysis.

For endothelial cell isolation from kidneys, single cell suspension from kidneys were obtained as indicated above in "Organ preparation for single cell suspensions", without performing leukocyte enrichment with Percoll. Cells were pelleted at 300 g, resuspended in blocking solution, containing PBS + 2% FCS and anti-CD16/CD32 mAb (2.4G2, homemade) to block Fc gamma receptors and filtered through a 35 μm cell strainer. Staining for surface markers and sorting of endothelial cells was performed as described above for brain endothelial cells.

## Flow cytometry

Single-cell suspensions from kidney and brain were stained for analysis by flow cytometry or for FACS sorting. Washing steps were carried out at 1880 g, if cells were analysed by flow cytometry or at 300 g, if cells were FACS sorted. All washing and incubation steps were performed in FACS buffer (PBS + 2% FCS). Prior to surface staining with antibodies, cells were incubated with anti-CD16/CD32 mAb (2.4G2, homemade) to block Fc gamma receptors. All surface stainings were performed for 20 min at 4 °C. For dead cell exclusion, the antibody mix included Zombie Aqua™ Fixable Viability Kit (BioLegend, 1:500). All used antibodies are listed in Table S2. Biotin-labeled antibodies were stained after a washing step with APC-labeled or BV711-labeled Streptavidin (eBioscience and BD Bioscience, respectively). For analysis by flow cytometry, the surface staining was followed by a 10 min fixation step using 4% Formalin (Sigma-Aldrich) at room temperature. In case of an intracellular staining for iNOS (NOS2), cells were subsequently permeabilized with permebilization buffer (PBS + 2% FCS + 0.1% saponin) and stained for 10 min on ice. Finally, cells were washed with permeabilization buffer and then with FACS buffer. For flow cytometry analysis, samples were acquired using a BD LSRFortessa instrument or Cytek Aurora instrument; for FACS sorting BD FACS Aria III or BD FACS Aria IIIu were used. For determination of the absolute cell count, a sample aliquot was analysed separately from the stained sample using the high throughput sampler (HTS). Flow cytometry data was analysed using FlowJo 10.7 software unless specified otherwise.

## Phenograph

Flow cytometry data was gated on single cells, viability-CD45+Lineage-, then subsampled (50,000 cells from each sample) and concatenated into one file. This file was then used for identification of cell clusters using a PhenoGraph[22] and UMAP[71] plug-ins in FlowJo 10.7 software. Heatmaps and dendrograms were created using R (Bioconductor, Seattle, WA).

Grouping of the clusters into cell types was done in the following manner: in the kidney, cluster 9, expressing Siglec-F, CD24, CD11b and F4/80 was defined as eosinophils. Cell clusters 5, 4, 7, 10, 8, and 14 all displayed high expression of Ly6G, CD11b and CD24, while presenting variability in CD172a expression; based on their high Ly6G expression, we decided to group these clusters together, and define them as neutrophils. Clusters 2, 3, 19, 1 and 20 expressed similarly high level of CD11b and CD172a, displayed relatively intermediate levels of F4/80 and CD64 and relatively high levels of Ly-6C. These clusters were grouped together and defined as Ly6C$^+$ monocytes, noticing that their expression of CD24 was heterogeneous. Clusters 24, 6 and 18 expressed high CD11b, CD172a and CD11c, with variable levels of CD64 and F4/80 and no Ly6C. We therefore grouped those cells together and defined them as Ly6C$^-$ monocytes. As clusters 21, 22 and 11 were high in MHCII, CD11b, CD11c, F4/80 and CD64, we defined those as monocyte-derived macrophages (Mo-Mac). Cluster 13 represents MHCII$^+$CD11c$^+$XCR1$^+$ conventional dendritic cells type 1 (cDC1s), and cluster 15 are MHCII$^+$CD11c$^+$CD172a$^+$CD11b$^+$ cDC2s. Cluster 17 was mostly similar to the Mo-Mac with intermediate level of CD11b indicating that these are tissue resident macrophages. Finally, clusters 12 and 16 expressed only CD45 and low amounts of Ly6C and MHCII, which made it difficult define them; however, these clusters contained very low number of cells and were not different between the two genotypes.

For the analysis of brain CD45$^+$ cells, we could group the 23 clusters into 8 cell types, one of which was more heterogeneous, and we considered it to consist of tissue resident immune cells, since they all had relatively low expression of CD45. Thus, clusters 21, 8, 22, 18, 14 and 19 were defined as tissue resident CD45$^{low}$CD64$^{int}$ cells, or microglia. Clusters 12 and 20 expressed high Siglec-F, CD11b, CD24 and variable levels of MHCII. We defined these two clusters as eosinophils. Clusters 10, 4 and 16 were defined as neutrophils, as they expressed Ly6G, CD11b and CD24, and had similar expression pattern as the neutrophils in the kidney. Clusters 1, 3, 2, 7, 6 and 9 resembled the Ly6C$^+$ monocytes of the kidney, and clusters 5 and 13 had similar expression to the Ly6C$^-$ monocytes of the kidney. Cluster 15 was unique in its high MHCII expression combined with the markers CD11b, CD172a, CD64 and F4/80, and therefore was defined as Mo-Mac. Clusters 17 and 23 are MHCII$^+$CD11c$^+$, and were defined as cDC1s (cluster 23, which was CD11b$^{low}$) and cDC2s (cluster 17, which was CD11b$^+$).

## High dimensional flow cytometry analysis of brain data

Single cell stained data from brain tissue was acquired as described above (enriched for microglia and astrocytes). Data for live mouse brain cells were exported as csv files and imported into the R 4.3.2 using the Spectre[72] v1.0.0 *read.files()* function. For each marker 1% percentile outliers were removed and *asinh* transformation with 5000 cofactor was used for downstream analysis. Afterwards, data was subseted into CD45+ and CD45- groups and clustered them separately using FlowSOM[73] v2.8.0. The resulting clusters were manually annotated and merged for further analysis. For heatmap visualization of marker intensities ComplexHeatmap[74] v2.18.0 was used. To estimate the significant differential abundance for each annotated cluster we use the *create.sumtable()* function from the Spectre Bioconductor package.

## Quantitative real-time PCR (qRT-PCR)

Analysis of gene expression by qRT-PCR was performed from whole kidney tissue (naïve, 1 dpi, 3 dpi). Kidney tissue was frozen on dry ice and stored at −80 °C until further processing. For extraction of RNA, RNAzol RT (Sigma-Aldrich) was added to the tissue after thawing, the tissue was homogenized using a TissueLyser (Qiagen) for 5 min at 25 Hz and RNA was isolated according to the manufacturer's protocol.

Quantification of *Il1r1* expression in conditional knock-outs and control mice was performed in sorted cell populations. Kidney and brain tissues were prepared as indicated in the "Organ preparation for single cell suspensions" section. CD45$^+$Sytox$^-$ cells were sorted into FCS-treated 1.5 mL tubes at 4 °C. For sorting of endothelial cells, single cell suspensions from kidney and brain were prepared as indicated in "Endothelial cell isolation". CD45$^-$Sytox$^-$CD31$^+$ cells were sorted into FCS-treated 1.5 mL tubes at 4 °C. Directly after sorting, cells were spun at 300 *g* for 10 min, resuspended in RNAzol RT (Sigma-Aldrich) and frozen at −80 °C until extraction of RNA. After RNA isolation, the transcription into cDNA was performed using GoScript Reverse Transcriptase (Promega) according to the manufacturer's instructions. qRT-PCR was performed with Brilliant II SYBR Green (Agilent Technologies) on an iCycler (BioRad). Expression was normalized to *TBP* as indicated in the Fig. legends.

The sequences of all primers used are listed in Table S1.

## Bulk brain RNA-seq analysis

Total RNA was isolated from whole mouse brain (*Il1r*$^{-/-}$ and *Il1r*$^{+/-}$ mice, 8 h after infection with 10$^5$ CFU *C. albicans*) using RNeasy plus micro kit (Qiagen) according to the manufacturer's protocol. Libraries were prepared using SMART-Seq RNA Stranded sample kit (Takara Bio) and sequenced by NovaSeq X Plus (Illumina) by the Functional Genomics Center Zurich. Sequencing was done using paired-end 150 bp reads with a final sequencing depth of 25 M paired-end reads per sample on average. Sequenced libraries were aligned with STAR[75] to the mouse reference genome GRCm38.p6 (release M23) and FeatureCounts[76] was used to count mapped reads per gene. Raw gene counts were imported in R v4.3.2 and analyzed with DESeq2[77] v1.40.2 to generate normalized read counts for visualization as heatmaps. Significantly differentially expressed genes (p < 0.01) were used with Metascape[37] v3.5.20240101 analysis.

## Histology

Organs were removed at day 2 after infection and fixed for 24 h in 10% Formalin (Sigma-Aldrich) for *Candida* specific staining, or 25% acetic acid/ethanol for immunohistochemistry, in both cases followed by incubation for 24 h in 70% Ethanol. The tissues were embedded in Paraffin (Leica) using a Tissue Processor (Thermo Scientific) and cut at a 4um thickness using a microtome (HM 355S (Microm AG)). Tissue sections were deparaffinized and stained with Silver Stain (Modified GMS) Kit (Sigma-Aldrich HT100A) for *Candida* specific staining. For immunohistochemistry staining, deparaffinized sections were washed with 0.1% Tween/PBS, then blocked with 12% BSA in Tween/PBS for 1 h at room temperature. After blocking, sections were incubated over night at 4 °C with primary antibody (Anti mouse neutrophil Ly6g/ Ly6c (abcam, ab25377) at 1:100 in blocking solution, Anti-Histone H3 (citrulline R2 + R8 + R17) (abcam, ab5103) at 1:100 in blocking solution). The next day, the sections were washed in PBS, and incubated 45 min at room temp with a secondary antibody in blocking solution (Rabbit anti-rat biotinylated (Vector laboratories, BA4001) at 1:100, anti-rabbit biotinylated (abcam, ab6720) at 1:250. The antibodies were then detected using an ABC-HRP Kit (VECTASTAIN®, PK-4000) and DAB Peroxidase Substrate Kit (3,3'-diaminobenzidine - LS-J1073), according to manufacturer's protocol. All sections were counterstained using Hematoxylin Solution, Gill No. 3 (Periodic Acid-Schiff (PAS) Kit, Sigma-Aldrich 395B), then dehydrated and mounted, using a Xylene based mounting media (Eukitt, Sigma-Aldrich 03989).

## Microscopy

Slides were scanned, using a slide scanner (Slide Scanner Panoramic 250 (3D Histech) for light microscopy, or Slide Scanner Evident/ Olympus VS200 for fluorescent slides). Light microscopy images were prepared using QuPath software. Fluorescent images were prepared

and analyzed using ImageJ2 (version 2.14.0). No manipulations were done on the images.

Analysis of hypoxic areas was done using ImageJ2 (version 2.14.0) on merged images (as they are displayed in the figure). In short, the color threshold was set for green (hypoxia positive staining) and the brightness threshold was set to include all areas brighter than background hypoxia. All images were analyzed using the same thresholds (color and brightness). The area was calculated and normalized to the total area of the tissue.

## Cytokine and chemokine assay
Kidney and brain were harvested at day 2 and 8 h p.i.. The tissues were immediately homogenized in sterile PBS (Sigma-Aldrich) using a TissueLyser (Qiagen) for 10 min at 25 Hz. Tissue lysate was analyzed using LEGENDplex™ Mouse Inflammation Panel (13-plex) (BioLegend 740446) and LEGENDplex™ Mouse Proinflammatory Chemokine Panel (13-plex) (BioLegend 740007), according to manufacturer's protocol, and acquired using a BD LSRFortessa.

## Killing assay
Killing assays were performed with neutrophils isolated from blood as well as with neutrophils, monocytes and moDCs isolated from kidneys of infected animals 3 days p.i.

Blood was taken from the vena cava and coagulation was prevented by addition of heparin (Drossapharm AG). ACK buffer (see above) was used to lyse red blood cells from the blood. Subsequently, blood neutrophils were isolated using biotin-labeled anti-Ly-6G mAb (1A8, BioLegend) in combination with Streptavidin MACS beads (Miltenyi Biotec) according to the manufacturer's protocol. Purity after MACS enrichment was > 90%. Kidneys were processed as described above (digestion, mashing, ACK lysis, Percoll gradient and viability and surface staining) and cells were FACS sorted using the following marker combinations: Neutrophils were CD11b$^+$ Ly-6G$^+$, monocytes were Ly-6G$^-$ CD11b$^+$ MHCII$^-$ F4/80$^+$ and moDCs were Ly-6G$^-$ CD11b$^+$ MHCII$^+$ F4/80$^+$. *Candida* was grown over night at 30 °C in YPD medium and was washed twice with PBS prior to opsonization with 5% naive serum in PBS for 30 min at 37 °C. For the killing assay, 10.000 *Candida* cells and 50.000 phagocytes were co-incubated in RPMI medium (with 2mM L-Glutamine (Sigma-Aldrich), 100 U/mL Penicillin and 100 µg/mL Streptomycin (Life Technologies), 10 mM HEPES (Lonza), 50µM 2-Mercaptoethanol (Thermo Fisher Scientific)) in a total volume of 200 µL in Protein LoBind Tubes (Eppendorf) at 37 °C, 5% CO$_2$. After mixing *Candida* and the phagocytes, cells were spun for 3 min at 300 $g$ to ensure their contact in the pellet. After incubation of 2.5 h, the phagocytes were lysed by adding 800 µL water containing 0.05% Triton X-100 and the number of surviving *Candida* was determined by plating the suspension on Pen/Strep-containing YPD agar. *Candida* samples without phagocytes served as controls to calculate the killing activity of the phagocytes. For the killing assay with *Candida* hyphae, *Candida* was aliquoted in RPMI medium with supplements (see above) into Protein LoBind Tubes and then incubated for 3 h at 37 °C, 5% CO$_2$ to induce hyphae formation. The successful induction of hyphae was visualized under a light microscope. Then, neutrophils were added and the assay was performed as described above.

## Assessment of neutrophil NET formation in-vitro
Neutrophils were isolated from bone marrow of the indicated mice, using Histopaque density gradient as previously described[78]. In short, bone marrow was harvested from both femur bones by flushing the bone with RPMI supplemented with 10% FCS and 2 mM EDTA (Sigma-Aldrich, E8008). The cells were strained through a 100 µm cell strainer. The cells were then pelleted by centrifugation at 427 $g$ at 4 °C for 7 min. To lyse red blood cells, the pellet was resuspended in 20 mL 0.2% NaCl (Sigma, S5886), followed immediately by the addition of 1.6% NaCl. The remaining cells were washed using RPMI (10% FCS, 2 mM EDTA).

Cells were resuspended in ice-cold PBS. For separation of neutrophils using density gradient, the cells were overlayed onto a Histopaque gradient (lower layer containing Histopaque 1119 (Sigma-Aldrich, 11191), upper layer containing Histopaque 1077 (Sigma-Aldrich, 10771)) and centrifuged at 872 $g$ at room temperature for 30 min. Neutrophils were then collected from the interface between the two densities, washed and collected in RPMI (10% FCS, penicillin/streptomycin (Gibco, 15140122)). The isolated neutrophils were then plated in 96 well plates, in a density of $2 \times 10^5$ cells per well, in HBSS containing CaCl$_2$ and MgCl$_2$. Plated neutrophils were then stimulated for 2.5 h at 37 °C with either Phorbol 12-myristate 13-acetate (PMA) (100 ng/mL, Sigma, P8139) and Ionomycin (1 µM, Cayman Chemical, 10004974), with $2 \times 10^5$ CFU *C. albicans* yeast, or with $2 \times 10^5$ CFU *C. albicans* yeast together with IL-1b (10 ng/mL, abcam, ab259421). After the stimulation, NET formation was quantified by the addition of Sytox green (160 nM, Thermo Fischer, S7020) and reading using a GloMax spectrophotometer (Promega), excitation 485 nm, emission 535 nm. Unstimulated neutrophils were used as controls.

## In-vivo treatment with IL-1b of NFκB reporter mice
KappaBle mice[31] were injected intravenously with 1 µg IL-1b in PBS (abcam, ab259421) or PBS as control. Four to five hours after injection, kidney or brain were analyzed as described above by flow cytometry. Brain was analyzed in separate experiments, using two different cell isolation protocols, one with endothelial isolation and the other with enrichment for astrocytes and microglia (both methods of single cell preparation are described above). GFP expression by single cells was captured by flow cytometry and GFP$^+$ cell percentage was quantified.

## Assessment of kidney damage
For blood urea and ACR measurements, blood and urine were collected from infected mice, at 10 h, 24 h and 48 h post intravenous infection with $10^5$ CFU *C. albicans*. Serum and urine were stored at −20 °C until analysis. Urea in the serum was measured according to the referred methodology[79]. Creatinine in the urine was measured as described before[80–82]. Urine albumin was measured using Urine/CSF Albumin reagent as described[83,84]. ACR was calculated by dividing urine albumin levels (in milligrams) by urine creatinine levels (in grams). Microalbuminuria was defined in the range of ACR between 30 mg/g and 300 mg/g, and macroalbuminuria was defined for ACR > 300 mg/g.

## Single nuclear RNA sequencing
For single nuclear RNA sequencing, animals were euthanized 8 h p.i. and perfused with PBS. Kidneys were transferred into RNAlater solution (Sigma Aldrich, R0901) and kept at 4 overnight. The next day, the RNAlater solution was removed, and the tissues were snap frozen in liquid nitrogen until processing. Processing of the tissue was done at the day of sequencing. In short, tissues were thawed, then cut into small pieces using a surgical blade. Next, the tissues were lysed using a Dounce homogenizer with Nuclei EZ lysis buffer (Sigma Aldrich NUC-101). The tissues were ground with a loose pestle 12–15 times, then another 12–15 times with a tight pestle. Homogenized tissue was incubated 5 min on ice, filtered using a 40 µm sieve and washed with Nuclei EZ lysis buffer. Nuclei were then pelleted at 500 $g$ for 5 min and washed once with 1% BSA containing RNAse inhibitor (New England BioLabs, 0.2 U/µl). Nuclei were briefly vortexed and strained through a 30 µm strainer. The quality of the nuclei was assessed under the microscope. Library preparation and RNA sequencing was performed by the Functional Genomics Center Zurich (FGCZ) using Drop-seq platform from 10X Genomics.

## snRNA-seq data preprocessing
Raw data was demultiplexed, filtered and mapped to the mouse reference genome (GRCm39, Ensembl release 106) using CellRanger

v7.0.0 with parameters expectedCells = 10,000 and includeIntrons=TRUE. Afterwards filtered feature matrices were imported to R v4.2.2 with Seurat v4.3.0 and merged to one object. Cells with fewer than 500 genes, fewer than 10,000 or bigger than 25,000 reads and more than 5% of mitochondrial and ribosomal genes were filtered out for the downstream analyses. We normalized the dataset using SCTransform v2[85] with the glmGamPoi[86] method using the 3000 most variable genes and estimated doublets using the DoubletFinder v2.0.3[87] R Bioconductor package with default parameters for each sample separately. After doublets removal we normalized once more with the SCTransform function and used RunHarmony[88] to integrate samples based on the condition variable with default parameters. We used the FindNeighbors and FindClusters Seurat functions with 0.8 resolution and removed cluster 13 that had, on average, a low number of detected genes (-1285), as well as high percentage of mitochondrial (-3.2%) and ribosomal (-4.4%) genes. To annotate celltypes, we called marker genes using the FindAllMarkers Seurat function with default parameters in the SCT assay and the "wilcox.test" was used for the significance test. For the celltype specific differential expression analysis between conditions we used the FindMarkers Seurat function with the following parameters: assay = SCT; test.use = "wilcox"; logfc.threshold = 0.2; min.pct = 0.05. For the data imputation we used the MAGIC algorithm[89] implemented with Rmagic v2.0.3.999 in the SCT assay and for all genes.

### Decoupler

To estimate pathway and transcription factor activity from the snRNA-seq data we used DecoupleR v2.3.2[90]. For the pathway activity estimation, we used the run_wmean function (min.size=5; mor='weight'; times = 100) with the top100 target genes per pathway from mouse PROGENy v1.17.13 database. For the transcription factor activity calculation, we used the DoRothEA v1.7.1 transcription factor - target gene database with 'A,B,C' confidence levels and the run_wmean function with the same parameters as described above.

### UCell signature scores

To calculate geneset signature enrichment scores for the specified KEGG ontology terms (Release 105.0 + /03-17, Mar 23) we used KEGGREST v1.36.3 and UCell v2.0[91] with the 'SCT' assay and default parameters.

### Metascape functional enrichment analysis

We used significantly differentially expressed genes (adjusted p < 0.05) from the pairwise comparisons between conditions (infected versus naive), as input to the functional enrichment analysis tool Metascape[37]. In order to run Metascape we downloaded MsBio Docker containers v3.5.20230101 and used each set of upregulated or downregulated genes per comparison separately. Based on the filtering by accumulative hypergeometric p-values and enrichment factors, the top 20 ontology terms were visualized per side and comparison.

### Pimonidazole treatment in-vivo and visualization

Kidney hypoxia was detected with Hypoxyprobe-Red549 Kits (Dylight™549-Mab) (Hypoxyprobe, Inc, HP7-100Kit) according to the manufacturer's protocol. Briefly, mice received an intraperitoneal injection of 100 mg/kg pimonidazole HCl (20 mg/mL in 0.9% NaCl). One hour after injection, kidneys were harvested, embedded in Tissue-Tek, and frozen in liquid N2-cooled isopentane. Kidney cryosections (10 μm) were blocked 10 min in ice-cold acetone, then washed twice with PBS and subsequently incubated for 1 h in the blocking buffer (3% BSA with 0.1% Triton X-100) at room temperature. Thereafter, samples were incubated overnight at 4 °C with primary anti-*C. albicans* antibody (clone: PA1-27158) (Invitrogen, 1:450), anti-pimonidazole rabbit anti-sera (HP3-Kit, 1:500). Slides were subsequently washed in PBS and

incubated for 1 h in blocking buffer with Goat anti-Rabbit IgG (H + L) Highly Cross-Adsorbed Secondary Antibody, Alexa Fluor™ 647 (A-21245, 1:1000) at room temperature. Nuclei were stained with Hoechst. Analysis and quantification of hypoxic areas is described above in Microscopy.

### ATP assay on agilent seahorse XF

The human podocyte cell line (Human Podocytes Ab8 13, supplied by ATCC[45]) was cultured in RPMI (Sigma-Aldrich) containing 10% heat-inactivated fetal calf serum (Gibco, Thermo Fischer Scientific), 50IU/ML penicillin, 50 μg/mL streptomycin (Invitrogen), 5 μg/mL of each insulin, transferrin and sodium selenite (ITS; Roche, 11074547001). AB8/13 cells were propagated at 33 °C and differentiated for 10–14 days at 37 °C in a humidified incubator containing 5% $CO_2$. Differentiated cells were seeded in an Agilent Seahorse XF96 well plate at a density of $6 \times 10^3$ cells per well, 5 well per condition. The day before the assay, the cells were stimulated with either recombinant human IL-1b (Thermo Fischer Scientific, A42508) or Anakinra (kindly provided by Profs. Thomas Kündig, University Hospital, Zürich and Leo Joosten, Radbout University, Nijmegen), or left unstimulated. The ATP assay was performed according to the protocol from Agilent Seahorse XF. Acquired data were analyzed in the Agilent Seahorse analytics platform, and the ATP rate was calculated as described in the Agilent Technologies white paper[47].

### *C. albicans* growth in hypoxic conditions

*C. albicans* was grown on YPD agar plates for 48 h under hypoxic conditions (1% oxygen, 37 °C, 5% $CO_2$) or normoxic conditions (21% oxygen, 37 °C, 5% $CO_2$) before imaging using Leica M205 FA stereomicroscope.

### Statistics

Statistical analyses of experiments were performed using Graph-Pad Prism software. For pairwise comparisons multiple unpaired Student t-tests were performed; For multiple comparisons, a two-stage step-up correction (Benjamini, Krieger and Yakutieli) was performed. For comparisons of three or more groups, one-way ANOVA was performed, with Bartlett's test for checking equal variances and Bonferroni's Multiple Comparison Test for defining significance. Significance levels were defined: $P < 0.05$ marked with *, $P < 0.01$ marked with **, $P < 0.001$ marked with ***, $P < 0.0001$ marked with ****.

### Reporting summary

Further information on research design is available in the Nature Portfolio Reporting Summary linked to this article.

## Data availability

The Next generation sequencing data generated in this study (kidney snRNA-seq and brain bulk RNA-seq) are available at the Gene Expression Omnibus (GEO) repository under the accession number (GSE282799). Processed brain FACS data for the Supplementary Fig. 2 has been deposited in the Zenodo database (https://doi.org/10.5281/zenodo.14894266). All other data shown in this study are provided in the Source Data file. Source data are provided with this paper.

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

## Acknowledgements
We thank the teams of the ETH Flow Cytometry Core Facility for cell sorting, the EPIC mouse facility for animal husbandry, and the Functional Genomics Centre Zürich for RNA sequencing. We also thank Prof. Roland Wenger, University Zürich, for scientific discussions and Franziska Ampenberger, Nadine Nägele, and Patrick Spielmann for expert technical help. O.D. is supported by the Research Priority Program (URPP) ITINERARE at the University of Zurich. M.K. is supported by ETH Zurich Project SKINTEGRITY.CH.

## Author contributions
S.H., M.S., F.P., and J.Z. performed the experiments. SH performed the data analysis and wrote the manuscript and I.B. performed bioinformatic analysis, K.D.B. discussed the results, S.N.K. provided reagents and discussed the results, OD performed analysis of kidney damage, J.K. performed data analysis and M.K. provided conceptualization, study design, funding, and manuscript writing.

## Funding

## Competing interests
The authors declare no competing interests.
