## [Transparent Peer Review file · Nature Communications]

IL-1 protects from fatal systemic candidiasis in mice by inhibition of oxidative phosphorylation and hypoxia

Corresponding Author: Professor Manfred Kopf

Version 0:

Reviewer comments:

Reviewer #1

(Remarks to the Author)

In this article, the authors attempt to delineate the mechanism of IL-1-mediated protection in the kidney and brain during invasive *Candida* infection. They use a variety of advanced techniques including sn-RNAseq and multi parameter flow cytometry. They show that mice lacking IL-1R are more susceptible to infection, and equate this to expression of IL-1R by epithelial/endothelial cells in the kidney which regulates their metabolic profile. Lack of IL-1R leads to a hypoxic environment which the authors speculate is caused by over-active OXPHOS by kidney structural cells and endothelium. These mechanisms are not explored in the brain. Overall, I thought this was a nice paper with some interesting observations. There are some statements that I think need to be softened/addressed regarding mechanism in the brain, and more experiments to clarify how IL-1R regulates metabolism would be helpful. Specific comments below.

Major

1. My main issue with the paper is around the role of microglia and IL-1R signalling and how the authors have framed the brain data. A role for IL-1R in regulating early PMN recruitment to the brain has been described before and this finding is therefore not novel. The authors make definitive statements about how microglia are not involved in the IL-1R KO phenotype because their conditional KO mice (Vav-Cre) retained some of the phenotype. However endo-specific KO of IL-1R had no phenotype in brain (but did in kidney). Also, their NFkB reporter mice show that one of the main cells to respond to IL-1b injection is microglia. The only other cell type to respond to IL-1b was an undefined CD31-EpCam- cell. My guess is that these could be astrocytes, since it was shown before astrocytes respond to IL-1b to drive microglia production of CXCL1 (Nat Immunol 2019, reference 19). I think a lot of the authors data point to a role for microglia (even if not exclusive) but the writing comes across that the authors are trying to rule this out... I don't really agree and think this should be softened. I also wonder if the authors had considered role of astrocytes given the published data and whether they looked via FACS to see if these cells are responding to IL-1b in their experiments?
2. Are the areas of hypoxia in the kidney shown in Fig 7 correlating with areas of fungal growth? Is the fungal growth creating the hypoxic areas, rather than the lack of IL-1R signalling? The proposed sequence of events isn't clear – the lack of IL-1R might mean an inability to control infection via some other method leading to more fungal growth and hypoxia, thus leading to the metabolic signatures seen. I think the authors are trying to propose that it is IL-1R that regulates the metabolism and thus the hypoxia and fungal growth stem from this, but I don't think they've conclusively shown this. Some suggested experiments to help could be to use Seahorse assays or similar to determine how IL_1b stimulation affects OXPHOS or glycolysis of cultured kidney or endothelial cells. The discussion indicates that IL-1b would drive OXPHOS, but is that actually known and occurs in the relevant cell types? Would blocking hypoxia therefore rescue the IL_1R KO mice?
3. Hypoxia is already known to drive hyphae formation in *Candida*, but also requires high CO₂. What CO₂ concentrations were these experiments done in (couldn't see it in methods)? Some discussion of this prior literature should also be included and acknowledged.
4. Definitions of the mono-macs and tissue-resident macs are a bit hazy. The tissue-resident macs are mostly defined by lower CD11b (at least in kidney). Knowing the recent developments about how monocytes can take on tissue-resident phenotypes (and therefore not true tissue-resident), I think some rewording of the subsets might help future-proof the results since ontogeny studies would be needed to be sure what is recruited and what is not.
5. Related to above, was CX3CR1 used in the brain panel for data shown in Fig 2? This would help distinguish microglia better and pick out border macrophage populations.
6. I'm not really convinced by the CitH3 staining shown in Fig 3. Can this staining be affected by tissue damage in the kidney caused by the infection? How can we be sure there are no artefacts caused by the fungus itself etc? An in vitro NETosis

assay or additional controls would be helpful here.

7. Fig 1c confused me a bit – are the lines connecting average CFU from each experiment (i.e. an average of all WT and KO tested in that experiment), where each experiment had 3-5 mice, or do the icons mean individual mice? If the latter, I don't understand how the values could be paired. It would be helpful if the legend clarified total number of mice tested and across how many experiments.

Minor

1. Fig S4 – no details given about n number or how many times these experiments were done.

2. I found Fig 5 really difficult to follow. Can the authors relabel some of the cell clusters with real labels as opposed to acronyms? I think this would help make the data easier to follow for non-kidney specialists

Reviewer #2

(Remarks to the Author)

The study by Horn et al explores the roles of IL-1 signalling in host defences during fungal infection. The focus is on the common human pathogen *Candida albicans*.

Strengths:

The authors do a nice job of showing that IL-1 signalling in epithelial, rather than immune cells, is responsible for controlling *Candida* infection in the kidney. The situation in the brain is more complex, with the likely scenario being that both immune and other cell types contribute. Transcriptomic data implicate a major metabolic reprogramming in the kidney in the absence of IL-1 signalling, and staining shows increased local hypoxia.

The proposed model is that the hypoxic environment drives invasive hyphal growth by *Candida*, which in turn contributes to disease. In other words, IL-1 signalling maintains normal metabolic programs in kidney cells, which ensures that a metabolic environment is maintained to suppress invasive hyphal growth by *Candida* and reduce disease.

The reported discoveries that a) IL-1 signalling in non-immune cells is protective in fungal infection and b) IL-1 signalling might be creating the correct metabolic environment for controlling fungal infections, are both important and potentially very exciting.

Weaknesses:

- The study provides little mechanistic insight on how IL-1 signalling might control metabolism in the kidney.
- The metabolic shift is inferred from transcriptomics data only. Many metabolic changes are observed in *Il1r*-deficient kidneys: e.g. glycolysis is increased, but so is mitochondrial respiration and fatty acid beta oxidation. As such, the precise metabolic reprogramming under these conditions remains unclear.
- The statement "metabolic cataclysm" needs clarification, both experimental and conceptual.
- Hypoxia is a known trigger for hyphal growth in *C. albicans*. As such, Fig 6B does not add new knowledge.
- While it is possible that increased hypoxia in *Il1r*-deficient kidneys is causing higher hyphal growth, other scenarios are also possible, and the data so far is not sufficient to conclude that this is the major reason for increase susceptibility of *Il1r*- mice to infection. I therefore feel this part – connecting the metabolic change in *Il1r*- mice to the fungal phenotypes - needs more work.
- For example, what are the transcriptional effects on *Candida* in these conditions, i.e. is there evidence that a hyphal hypoxic response is activated (see Sellam lab work)? Are there any *Candida* mutants in the hyphal hypoxic response that could be used to corroborate the idea that specifically this pathway is impacted in *Il1r*- mice?
- There is a large increase in fungal CFUs in *Il1r*- and *Il1b*- kidneys. This indicates that fungal growth, not just morphology, is impacted. In other words, I feel that the higher CFUs cannot be explained only by increased hyphal growth due to a hypoxic environment. Assessing the fungal response under these conditions might provide important information.

Reviewer #3

(Remarks to the Author)

The manuscript "IL-1 protects from fatal systemic candidiasis by inhibition of oxidative phosphorylation and hypoxia" by Horn, et al, is focused on identifying the mechanism(s) through which IL-1 receptor signaling acts to protect a "healthy" host from invasive candidiasis. The manuscript uses knock out mice and bone marrow chimera mice to dissect the immune responses to *C.albicans* infection. This manuscript addresses an important topic in the field of host-fungal interactions. In particular, the attempt to identify the mechanism of action through which defense occurs rather than just identifying the pathways involved is very appealing. There are some concerns about how strongly to interpret the results. Furthermore, the manuscript would be stronger with some additional data and some modifications in how the data is presented and how the manuscript is written.

Key Results:

The manuscript begins with data demonstrating that IL-1R1 signaling, particularly via IL-1b, is critical for defense against systemic candidiasis. In particular, IL-1R1 knockout mice have more disease and higher fungal burdens in the brain and kidneys. The authors demonstrate that IL-1R1 knockout mice can mount a strong inflammatory response and recruiting inflammatory cells to the site of infection. Furthermore, the ability of monocytes to express iNOS and of neutrophils to form NETs *in vivo* and kill *C.albicans* organisms *ex vivo* is IL-1R1 independent. Some of this data is consistent with previously

published data but the authors have carried out a systematic analysis of this that is important to the subsequent studies. The most novel and potentially impactful findings in the manuscript are 1) that the requirement for IL-1R1 in defense against systemic candidiasis is in non-hematopoietic cells; 2) recruitment of neutrophils to the site of infection in the kidney is independent of endothelial cell IL-1R1, 3) IL-1R1 deficient mice that are infected with *C.albicans* undergo a substantial change in kidney gene expression related to aerobic respiration, carbon metabolism, oxidative stress, and the response to hypoxia, and 4) the kidney of IL-1R1 deficient mice infected with *Candida* is substantially hypoxic.

Validity

My major concern about the manuscript is that the authors present one possible interpretation of their results but there are other interpretations that may be equally likely. This does not remove all interest in their results, but it does require that these conclusions either be validated further or be viewed in a more preliminary light than they are presented. Main concerns include:

1. The IL-1R1 knock out mice have high organism burden and high inflammation. How does this impact the findings? Cytokine and chemokine production, immune cell recruitment, gene expression in infected tissues, etc. all can be a function of the level of disease in the tissue, not just the specific change being studied. For example, when the authors state that there was an elevated number of neutrophils in the knockout compared to WT mice, is that because of the change in IL-1R1 signaling directly or because a neutrophil independent factor caused increased organism burden in the knockout mouse and that increased burden results in different responses that are actually IL-1R1 independent?

This is a challenge in any type of research like this but it requires extreme care during interpretation of results. In relation to this manuscript, this is extremely important for interpretation of the authors' conclusions about the metabolic derangement and hypoxia that occur in the knockout mice. Is that because the knockout mice have severe disease and the gene expression changes/hypoxia are due to the massive tissue damage that is occurring in response to the disease or is it because of the lack of IL-1R1 signals? The gene expression studies were done at 8 hours post infection, which suggests that it's not because of overwhelming disease nor global tissue necrosis. However—in order to make the argument that the gene expression changes are occurring prior to development of significant disease, the authors would need to at least present information about organism burden and conventional histology and/or other measures of tissue damage at the same time point as the gene expression studies.

2. A large part of the manuscript involves looking at brain organism burden, cell recruitment, etc. in response to CNS infection with *C.albicans*. Is there any data to indicate that IL-1R1 signaling influences organism dissemination into the brain rather than just to responses that occur once the organism has crossed the blood brain barrier?

Minor concerns:

1. The authors state that their data in figure 3A-C “indicate that phagocytic and fungicidal activity of monocytes and neutrophils are not affected in *Il1r1*^{-/-} mice.” They have not presented any data on phagocytosis and have not presented data on the fungicidal activity of monocytes.

Data and methodology

In general, the experiments are well designed with good techniques and controls. I have some specific concerns about the methodology:

1. In figure 1a, the authors normalize gene expression data to *g6pdx*. This gene undergoes changes in expression in conditions of hypoxia, oxidative stress, and metabolic derangements. Given the results presented later in the paper, is this a valid gene to use as a housekeeping control?

2. In figure 1f/g – the use of hematoxylin as a background stain is very useful for identifying inflammatory infiltrates occurring in the regions of large clusters of *C.albicans* hyphae. However, it makes it very difficult to identify whether or not there are individual hyphae and/or clusters of yeast at the magnification presented. This would be stronger if the inserts had higher magnification and/or the authors include images with adjusted contrast (applied systematically, of course), to allow discrimination between small black dots (yeast or cross-sectional hyphae) and small dark purple dots (immune cells).

3. The authors state that *C.albicans* is distributed throughout brain tissue as seen in Fig 1g. The images presented could be the edge of a cluster – it doesn't mean the organisms are truly distributed throughout the tissue.

4. In figure 2a – it would be helpful if the cytokines/chemokines data includes the data for the same set of cytokines for both organs – this would allow better comparison between the organs. Furthermore, the authors refer to data about CCL22, CXCL10, and CCL17 in the text but these aren't included in the figure.

5. Microglia are incredibly difficult to study as they are a heterogeneous population of cells with a variety of unusual responses. The authors use CD64+CD45^{lo} staining to identify microglia. Microglia can also be CD45^{int} and have a variety of other phenotypes. The authors should either include a statement that they are using only this staining and therefore the data may not reflect all microglia populations or present data with additional markers showing that this population really does reflect all microglia in their system (see Honarpisheh et al *J Neuroinflammation* 2020). Furthermore – the authors mention that irradiation / bone marrow transplant doesn't impact microglial populations. This is not always true (see Okonogi et al, *J Radiat Res*, 2014).

Analytical approach

No concerns

Suggested improvements

The authors should either a) include the gene expression data as is and add a significant discussion about all possible explanations such as stating that it is possible that IL-1R1 signaling suppresses the massive changes in gene expression or that the level of disease in the kidney at 8 hours is already sufficient to prompt broad scale expression changes (and any other possible explanations of the data), b) include data to demonstrate that the level of kidney disease in the WT and knockout mice is the same or both. They should also carefully edit the manuscript for other instances when they make

conclusions about the role of IL-1R1 when they may be describing IL-1R1 independent effects that occur because of the different level of organism burden / disease.

They should also address the questions raised in the data and methodology section, above.

Clarity and context

The discussion does a very good job of presenting how their data aligns with previously published data and within our current understanding of *C.albicans* disease. However, there are multiple instances where they need to be more precise with their language. For example, there are statements such as “mean expression is displayed in heatmaps” when they are really displaying average number of cells; “oxidative” is written in a discussion about “nitrosative” damage, and “host derived cells” should be non-hematopoietic cells.

References

References are appropriate.

Reviewer #4

(Remarks to the Author)

With multiple types of evidence, the authors suggest that IL-1R-signaling is crucial for innate protective response independent of neutrophils in the kidney and brain and it is required to prevent fatal candidiasis by inhibiting a metabolic cataclysm including excessive oxidative phosphorylation and hypoxia. Overall, the story is solid and well-organised. Please find the comments and questions/suggestions as followings:

1. In figure 2C, among all 7 celltypes shared by kidney and brain, quantification of cell numbers of five celltypes showed a similar pattern in WT and *Il1r1*^{-/-} mice, but an opposite trend was observed in brain and kidney for celltype cDC1s and cDC2s. Is there an assumption to explain the observation?

2. For figure 1E, figure 2 and figure 3B, the authors evaluated the phenotypes at DPI=3. However, DPI=2 was chosen as the time point for figure 1F, 1G (GMS staining) and figure 3C (Ly6G and Cit H3 staining). Is there a specific reason to choose different DPI for different assays?

According to figure 1D and 1E, an elevated fungal growth in brain has been observed in the first 24hrs and it lasted at least for 72hrs. It would be nice to check each individual at 24hrs, 48hrs and 72hrs post infection.

3. Figure 1G showed the hyphae locations in a mouse brain at DPI=2, it seems there's a higher fungi density in the hypothalamus. It would be nice to show more staining results of various sagittal slices of the same individual from one side to another, or showing the same region of other individuals to indicate if there's a specific brain region/layer more sensitive to *C. albicans* infection.

Similarly, it would be nice to show more slices with NETs locations in figure 3C.

4. Suggestion, not required: For the first 16hrs post infection, the number of neutrophils in the brain of *Il1r1*^{-/-} mice are lower than WT mice. However, the number of neutrophils in the brain of *Il1r1*^{-/-} mice are much higher than WT mice 3 days post infection.

Is it possible to collect the infected brain samples at an early time point (8hrs or 16 hrs post infection) and a later time point (48hrs or 72hrs post infection) of *Il1r1*^{-/-} mice and perform the single cell sequencing? Maybe it can give a clue about how neutrophils and other celltypes reacted when infection progressed.

5. For Figure 2B, CD45+ cells including brain specific cell type microglia were sorted out and taken for analysis. However, astrocytes which cannot be enriched by CD45+ selection are also considered as key immune effector cells in CNS and they can become activated in pathological states. I wonder if the authors have ever checked the cell number difference of astrocytes in WT and *Il1r1*^{-/-} post infection.

6. For figure 5C, it's a little bit confusing to understand according to the simple description of legend and the main text.

The Figure.5C legend says “(C) Shared genesets between celltype specific DE genes for *Il1r1*^{-/-} infected (top) versus *Il1r1*^{+/-} infected (bottom) comparison”.

The statement made about Figure 5C in the main text: “Genes upregulated in the infected *Il1r1*^{-/-} mice were highly overlapping between different kidney cell types, suggesting a similar uniform response in the whole kidney.”

Does it mean that the DEG sharing showed in figure.5C top panel was plotted based on up-regulated genes in *Il1r1*^{-/-} infected cells, while figure.5C bottom panel was plotted based on down-regulated in *Il1r1*^{-/-} infected cells (up-regulated genes in *Il1r1*^{+/-} infected cells) ?

If that is the case, it's kind of weird to give the conclusion like “suggesting a similar uniform response in the whole kidney” only based on up-regulated genes.

7. Minor question for figure.6A: As metascape uses all genes of the genome as the background gene set, the enrichment analysis might miss some terms which could be significant if the background list is limited to the genes expressed in the cells analysed in this study (for example, only consider the genes expressed in >5% of the cells and reads >= 2).

Version 1:

Reviewer comments:

Reviewer #1

(Remarks to the Author)

The authors have added significant amounts of new data including high dimensional flow and Seahorse metabolic data. I think the textual revisions made are satisfactory and the authors have sufficiently responded to each of my prior comments. Their responses are thorough and I have no further comments.

Reviewer #2

(Remarks to the Author)

I thank the authors for considering my comments and responding to them experimentally and by modifications to the text.

As the authors acknowledge, the link between the metabolic changes in the host dependent on IL-1 signalling and fungal phenotypes (hyphal growth, virulence) remains to be directly shown. Nevertheless, it is my opinion that the current study presents a new angle in our understanding of the interface between immune factors, host metabolism and fungal pathogenesis.

Reviewer #3

(Remarks to the Author)

The revised manuscript "IL-1 protects from fatal systemic candidiasis by inhibition of oxidative phosphorylation and hypoxia" by Horn, et al, is focused on identifying the mechanism(s) through which IL-1 receptor signaling acts to protect a "healthy" host from invasive candidiasis. The manuscript uses knock out mice and bone marrow chimera mice to dissect the immune responses to *C.albicans* infection. This manuscript addresses an important topic in the field of host-fungal interactions. In particular, the attempt to identify the mechanism of action through which IL-1 signaling impacts host defense rather than just identifying the pathways involved is very appealing.

The biggest concern I have about the revised manuscript is that it is now overly full of potentially unconnected data, there are a huge number of interpretations of different findings, sometimes the authors link one finding to another that isn't especially relevant to it, and their conclusions are often overstating their findings. This is particularly prevalent given the many different comparisons that are happening across the different time points after infection. The main point(s) they are trying to make are now lost.

In an attempt to re-focus on the information the authors feel is most important, I went back to the abstract. Unfortunately, the abstract seems to be suffering the same problem and seems to have significant problems itself. Here are several statements from the abstract and specific concerns about them:

Lines 26/27: In the kidney, endothelial IL-1R contributes to fungal clearance independent of neutrophils...

They have not presented any data on what happens in the absence of neutrophils. There is an increase in fungal burden at 3 days after infection in *PdgfbicreERT2Il1r1fl/fl* mice compared to control mice but other data in the manuscript indicates that at 3 days after infection, there is substantial kidney damage, near overwhelming infection and a hyperinflammatory state so it's hard to know what the role of the endothelial cells was or when it was important. Furthermore, the authors are comparing the organism burden at day 3 with the neutrophil recruitment data from 8 hours after infection. They present data that indicates that the number of neutrophils in the kidney is tightly correlated to organism burden and the organism burden at 8 hours is the same in WT and IL-1R1^{-/-} mice. Thus, it's very difficult to untangle what might be happening and what can be attributed to the endothelial cells.

Lines 27-29: In the brain, IL-1R-signaling indirectly recruits neutrophils and monocytes by regulating chemokines and adhesion molecules...

There is data showing that CXCL1 levels in both brain and kidney are substantially lower at 8 hours after infection in IL-1R1-deficient mice and at the same time point, there are fewer neutrophils in both brain and kidney as well. Given the data in the paper and that presented by others (who previously demonstrated the same findings), it is certainly likely that it is the CXCL1 effect that's altering neutrophil recruitment. This isn't novel nor is it a completely explored mechanism. More concerning is the statement that IL-1R signaling regulates adhesion molecules which then recruit neutrophils. The only times the word "adhesion" appears in the manuscript are line 29 in the abstract and in the gene ontology list within Figure 6A.

Lines 29-30: Single-nucleus-RNA-sequencing revealed Il1r1-mediated NF-κB activation in distinct kidney cell types.

This is supported by Figure 4D but of unclear significance to the remainder of the investigation.

Lines 30-32: Our data reveal excessive ribosomal activity and oxidative phosphorylation across all cell types in the kidney of Il1r1-deficient mice within a few hours upon infection, causing localized hypoxia at infection foci...

There is no data on ribosomal activity in the manuscript. There is upregulation of oxidative phosphorylation gene pathways in the kidney of IL-1R1-deficient mice at 8 hours after infection but no *in vivo* data on the process itself. Furthermore, there is no data to demonstrate that it is increased oxidative phosphorylation that is causing the hypoxia which is seen at 24 hours after infection.

In the revised manuscript, the authors have included information correlating levels of neutrophil recruitment with organism burden and added an interpretation that the number of neutrophils in the tissue "at the height of infection" is due to the fungal burden. They also added data on BUN and ACR that demonstrates that at 10 hours, there is no significant difference between WT and knockout mice in terms of kidney damage. This is very supportive of their conclusions that their 8 hour gene expression findings are related to IL-1R signaling and not to overwhelming infection. However – it is still not at all clear that the hypoxia they demonstrate occurring at 24 hours after infection (Fig 7D) is a direct effect of IL-1R signaling and not either organism burden or neutrophil activity. Figure 1D indicates that the organism burden in the kidney at 24 hours after infection is statistically "not different" from that seen in WT mice although the mean burden in IL-1R1-deficient mice is approx. 0.3 logs higher than WT and with the distribution of the data points, one wonders if that would be "significant" if more

points had been included. In figure 7, though, the areas with increased pimonidazole staining co-locate with areas that clearly have a much higher local organism burden. Based on other studies, those foci of infection also typically are strong foci of neutrophil recruitment. Given all the extra activity that is going on in those areas, isn't it possible that the hypoxia seen in the kidney is an indirect effect stemming from the host/pathogen interaction rather than from IL-1R1-deficiency?

The additional data goes a long way to argue that the changes in gene expression seen at 8 hours after infection in the kidney are a function of IL-1R1 signaling. This is a very interesting finding. The finding that it is a non-hematopoietic requirement for IL-1R1-deficiency that is driving the ability to control infection is also very interesting. If the authors could focus on these points, they could make this argument clearly and relatively cleanly. Without full mechanistic data, no one could "prove" this is a direct effect, but their data is interesting enough to warrant publication and further investigations. In order to do that, they need to substantially trim and organize the data and their discussion of it. I don't think that the inclusion of data on brain infection is substantially contributing to their points and it is certainly "muddying the waters." It should be moved to a different manuscript. Likewise, I'm not sure that the data in Figures 2 and Figure 4C, D, and G contribute substantially to the manuscript.

The data on hypoxia is relevant to the gene expression data and certainly interesting but still a bit underwhelming. This includes my concern above about the co-localization of the hypoxia with dense organism numbers and, one expects, high density of neutrophils. An additional concern is the seahorse data on a kidney podocyte cell line. If they feel that endothelial cells are the location at which presence or absence of IL-1R1 matters, why did they test seahorse data on podocytes? Why isn't there any statistical analysis of this data? Is the amount of difference observed either statistically or biologically relevant? They include untreated cells and cells treated with IL-1B or anakinra. What was the anakinra doing in the cell cultures that should have had very low levels of IL-1B in them to start with? The anakinra argument would have been much more convincing if they had treated cells with both anakinra AND IL-1B.

There is very interesting data in this manuscript but it is still hampered with a variety of over-conclusions and inaccurate statements.

There are still a large number of places in the results and discussion which involve results obtained from 48 hours, 2.5 days, and 3 days after infection – this is the time that the authors have shown to line up with significant kidney damage and very high organism burden and neutrophil density. How much weight should be given to those results?

There are still a variety of overstatements such as:

Lines 508/509... tissue hypoxia in the microenvironment around *C. albicans* infection foci, which we and others showed to increase *C. albicans* pathogenicity

The authors only showed that hypoxia increases invasive growth, not pathogenicity.

Lines 534/535: Taken together, our data reveal a non-conventional role for IL-1b, and to a minor extent IL-1a, during early stages of *C. albicans* invasion into the kidney.

There is no data on how IL-1a may be involved.

Other concerns:

-EVERY figure should include a label in that figure about what time point the data came from.

-Data on how many replicates per experiment and how many repeats of each experiment should be included

Reviewer #4

(Remarks to the Author)

The authors achieved a significant improvement compared to the last version.

Please find some points need to be addressed as follows:

In Line 191-199, "Flow cytometry analysis of the brain at 48h p.i., using a wider antibody panel, revealed significantly reduced numbers of both "activated" (Dectin+) and "homeostatic" microglia, as well as CD206+ border-associated macrophages and neutrophils in Il1r1^{-/-} mice, compared to Il1r^{+/-} littermate controls (supplementary Fig. 2A-C). Interestingly, we observed significantly increased ACSA-2low astrocytes and decreased ACSA-2high astrocytes....."

Is there a cell number difference in astrocytes/BAM can be observed between Il1r1^{-/-} mice and Il1r^{+/-} I mice without infection? It was only mentioned that there's no difference in the cell types shown in supplementary figure 1C between Il1r1^{-/-} and WT mice, but no similar statement for supplementary Fig. 2A-C.

2. Line 267-269:

"Like global Il1r1^{-/-} mice, at 3 days p.i., PdgfbicreERT2 Il1r1^{fl/fl} mice showed elevated number of neutrophils and inflammatory monocytes in the kidney, relatively to their Il1r1^{fl/fl} littermates (Supplementary Fig. 5C)."

No p-value/significance indicated in the figure S5C.

3. In the response to reviewers, the authors suggested "Attached are three slices from three Il1r1^{-/-} mice, capturing different areas of the brain. The staining is of CitH3 (neutrophil NETs)."

Cannot find any attached slices in the response file or supplementary file in terms of CitH3 staining. Please specify.

4. minor point: Line 102: Il1r1^{-/-} should be Il1r1^{-/-}

Version 2:

Reviewer comments:

Reviewer #3

(Remarks to the Author)

The authors revision of the manuscript appropriately addresses my previous concerns. I agree with the authors that their data compromises new and interesting findings that will prompt many future questions in the field of *C.albicans* host-pathogen interactions.

A small concern is that the majority of the figure legends, including all of the figure legends for the primary (non-supplemental figures) do not indicate what p value is used to designate a *, **, or *** in the figures.

I recommend that the following very minor inconsistencies, which might be typographical errors, be addressed:

Lines 86-87: The text indicates that a “small but significant difference to controls was observed in the brain as early as 24 h p.i (Fig 1D) but the figure doesn’t have any markings to indicate significance and a p value is not mentioned in the figure legend.

Lines 89-90: The text reports that “the lack of IL-1a did not impair the ability to fight the fungus...” but in Figure 1E, brain samples, there is an apparent difference between the values for WT and IL-1a^{-/-} mice and there is a * for the comparison between those samples.

Lines 101-102: The text reports that the data in Fig 1G is from day 3 but the figure indicates 2 dpi

Reviewer #4

(Remarks to the Author)

The authors made essential revisions to improve the manuscript and supplied elaborate responses to the reviewers. I have no further comments.

Point-to-point reply to the reviewers.

Reviewer #1:

- 1. My main issue with the paper is around the role of microglia and IL-1R signalling and how the authors have framed the brain data. A role for IL-1R in regulating early PMN recruitment to the brain has been described before and this finding is therefore not novel. The authors make definitive statements about how microglia are not involved in the IL-1R KO phenotype because their conditional KO mice (Vav-Cre) retained some of the phenotype. However endo-specific KO of IL-1R had no phenotype in brain (but did in kidney). Also, their NFkB reporter mice show that one of the main cells to respond to IL-1b injection is microglia. The only other cell type to respond to IL-1b was an undefined CD31-EpCam- cell. My guess is that these could be astrocytes, since it was shown before astrocytes respond to IL-1b to drive microglia production of CXCL1 (Nat Immunol 2019, reference 19). I think a lot of the authors data point to a role for microglia (even if not exclusive) but the writing comes across that the authors are trying to rule this out... I don't really agree and think this should be softened. I also wonder if the authors had considered role of astrocytes given the published data and whether they looked via FACS to see if these cells are responding to IL-1b in their experiments?**

Authors:

We readily admit that the brain data regarding immune cell recruitment are not entirely novel, and we quoted the relevant work from Lionakis and colleagues. Therefore, we focused our study mainly on the kidney to reveal the mechanism.

However, as requested by the reviewers, in the revised version, we have better-characterized microglia, astrocytes, border-associated macrophages, and the myeloid compartments in the brain 48h p.i. by high-dimensional flow cytometry 4. This analysis *".... revealed significantly reduced numbers of both "activated" (Dectin⁺) and "homeostatic" microglia, as well as CD206⁺ border-associated macrophages and neutrophils in Il1r1^{-/-} mice, compared to Il1r1^{+/-} littermate controls (supplementary Fig. 2A-C). Interestingly, we observed significantly increased ACSA-2^{low} astrocytes and decreased ACSA-2^{high} astrocytes. At this time point (48h post-infection), there was already a considerably higher fungal load in the brains of Il1r1^{-/-} mice (supplementary Fig. 1D), which could indicate that microglia and border-associated macrophages might be especially vulnerable and die due to infection. The heightened inflammation may explain the difference in astrocyte numbers"* page 7, lines 191-199.

Furthermore, bulk RNA sequencing of the brain at 8h p.i. showed significantly lower expression of molecules responsible for extravasation into the tissue, such as Icam, Lcn2, Sele, and Vcam1. Also, the expression of the chemokines Ccl1 and Csf1 was substantially reduced, which explains the delayed recruitment and differentiation of monocytes in the absence of IL-1R1. Page 7, lines 203-207.

These data are entirely novel!

Regarding the role of microglia. We did not intend to say that they are not involved in protection from candidiasis. Instead, in the discussion or the original version, we wrote that "our results may indicate a partial contribution of microglia IL-1R-signaling for protection from fungal growth in the brain".

A contribution of hematopoietic cells in the brain is clearly indicated by a 10-fold difference in fungal load compared to WT mice (Fig 4b). However, comparing global IL-1R KO and WT, there is >10,000-fold difference. That's why we stated that hematopoietic cells make a minor contribution.

In the revised manuscript, we have changed the text as follows:

*"However, as irradiation does not deplete brain resident microglia, we used the conditional KO mice $vav^{cre}Il1r1^{fl/fl}$, which have a deletion in the *Il1r1* gene locus in all hematopoietic cells including brain microglia (supplementary Fig. 5A). A statistically significant elevated fungal load was only seen in the brain (Fig. 4B). However, the differences in fungal titer were smaller compared to global $Il1r1^{-/-}$ mice (i.e. 10-fold vs > 10000-fold) (Fig. 1C), mirroring the $Il1r1^{-/-} \rightarrow WT$ group from the bone marrow chimera experiment (Fig. 4A).*

*From these data, we conclude that the defense against systemic candidiasis strongly depends on *Il1r1* expression in non-hematopoietic cells, while IL-1R signaling in hematopoietic cells plays a less critical role."* (page 8, line 228)

In the discussion, we now state that

"..... our results indicate a partial contribution of microglia and astrocytes IL-1R-signaling for protection from fungal growth in the brain" (page 16, line 476).

(See below why astrocytes)

Nonetheless, for this revision, we have explored more what is happening in the brain.

We first analyzed again the brain populations at the height of infection to better differentiate microglia, border macrophages, and astrocytes:

"Flow cytometry analysis of the brain at 48h p.i., using a wider antibody panel, revealed significantly reduced numbers of both "activated" (Dectin⁺) and "homeostatic" microglia, as well as CD206⁺ border-associated macrophages and neutrophils in $Il1r1^{-/-}$ mice, compared to $Il1r1^{+/+}$ littermate controls (supplementary Fig. 2A-C). Interestingly, we observed significantly increased ACSA-2^{low} astrocytes and decreased ACSA-2^{high} astrocytes. At this time point (48h post-infection), there was already a considerably higher fungal load in the brains of $Il1r1^{-/-}$ mice (supplementary Fig. 1D), which could indicate that microglia and border-associated macrophages might be especially vulnerable and die due to infection. The heightened inflammation may explain the difference in astrocyte numbers." (page 7, line 191).

Next, we used this antibody panel to address the response to IL-1b again, using the NFkB reporter mice:

"In the brain, we found IL-1R-signaling mediated NF-κB activation in Border-associated macrophages (BAM) (GFP⁺CD45⁺CD206⁺), astrocytes (GFP⁺CD45⁻ACSA-2⁺) and in undefined CD45⁻CD31⁻EpCam⁻ cells, but not in endothelial and epithelial cells (Fig. 4D and supplementary Fig. 4B-C)." (page 9, line 251).

Indeed, astrocytes had the most significant response to IL-1b, indicating the possibility that these cells contribute to IL-1R-driven protection.

2. Are the areas of hypoxia in the kidney shown in Fig 7 correlating with areas of fungal growth?

Authors:

We repeated our hypoxia staining in the kidney and included an anti-*C. albicans* antibody:

"We used an anti-pimonidazole antibody and a fluorescent secondary antibody to visualize the Pimonidazole-positive areas. We then visualized the C. albicans, using a C. albicans-specific antibody. Since both antibodies have been generated in the same species, we could not co-stain the same slides. Thus, we stained subsequent slides (10µm apart) and approximated the co-localization of hypoxia and C. albicans (Fig. 7D). Consistent with the transcriptional analysis, the kidneys of Il1r1^{-/-} mice had significantly more hypoxic areas than the Il1r1^{+/-} controls (Fig. 7E). Interestingly, we found that hypoxic regions co-localized with C. albicans in many of the infection foci (Fig. 7D), more in the Il1r1^{-/-} mice than in the Il1r1^{+/-} controls." (page 14, line 401).

It is essential to mention that mild hypoxia staining was present in the kidney medulla both in KO and control mice. Consultation with kidney experts confirmed that the kidney medulla normally has some levels of physiological hypoxia, and this level of staining is expected. Quantification of the areas with bright hypoxia staining confirmed higher levels in the Il1r1^{-/-} mice (Figure 7B).

Is the fungal growth creating the hypoxic areas, rather than the lack of IL-1R signalling?

Authors:

"This observation suggests that C. albicans itself could contribute to the creation of hypoxia by over-consumption of oxygen. However, since IL-1R seems to prevent most of the hypoxia, as seen in Il1r1^{+/-} control mice, we believe that the kidney cells have a significant role in its creation." (page 14, line 409).

The proposed sequence of events isn't clear – the lack of IL-1R might mean an inability to control infection via some other method leading to more fungal growth and hypoxia, thus leading to the metabolic signatures seen. I think the authors are trying to propose that it is IL-1R that regulates the metabolism and thus the hypoxia and fungal growth stem from this, but I don't think they've conclusively shown this. Some suggested experiments to help could be to use Seahorse assays or similar to determine how IL-1b stimulation affects OXPHOS or glycolysis of cultured kidney or endothelial cells.

Authors:

The metabolic signature is seen in the cells at 8h after infection. At this early time point, there is no difference in fungal burden between KO and control animals (supplementary Fig. 7A). Additionally, for this revision, we monitored kidney damage throughout the infection and showed that in the first 24h, there is no damage to kidney seen yet (Fig. 5D-E). Importantly, we took the suggestion to use the Seahorse device to determine how IL-1b stimulation affects OXPHOS and glycolysis in cultured kidney cells:

"To assess the possibility that IL-1R intrinsically influences kidney cell metabolism, we used a human kidney podocyte cell line (differentiated AB8/13 cells). We performed an ATP assay on Agilent Seahorse XF96 analyzer, which allows the precise quantification of the rate of adenosine triphosphate (ATP) production from glycolysis and mitochondria in live cells⁴⁵. Before analysis, we treated the cells overnight with either IL-1b or the IL-1R antagonist Anakinra (Figure 7A, B). The assay revealed that podocytes increase the glycolytic source of ATP upon IL-1b treatment, while Anakinra treatment caused a significant increase of the mitochondrial (oxidative) source of ATP in the cells (Fig. 7C). Quantification of the mitochondrial ATP fraction over glycolytic ATP fraction demonstrated a significant effect of IL-1R signaling on the cell ATP production. In other words, while activation of IL-1R causes increased glycolytic ATP, blocking of IL-1R (as is happening in Il1r1^{-/-} mice) causes increased respiratory ATP. These results demonstrate that IL-1R intrinsically influences kidney cell metabolism, specifically shifting away from respiratory ATP production. In Il1r1^{-/-} mice, however, the kidney cells respond to C. albicans infection by increased cellular respiration." (page 13, line 380).

"We suggest that the metabolic response of the kidney cells to the infection in the absence of IL-1R exacerbates the depletion of oxygen in the infection foci, and contributes to the creation of local hypoxia, which in turn activates HIF-1 α . The HIF-1 α activation, however, is not enough to combat the hypoxia." (page 14, line 413).

The discussion indicates that IL-1 β would drive OXPHOS, but is that actually known and occurs in the relevant cell types? Would blocking hypoxia therefore rescue the IL_1R KO mice?

Authors:

The discussion indicates that IL-1 inhibits OXPHOS, this is something that has been shown in adipose cells (<https://www.ncbi.nlm.nih.gov/pmc/articles/PMC7566776/>) by activation of an unconventional signaling pathway that directly impacts mitochondrial metabolism.

Our seahorse data can go together with this idea, as blocking of IL-1R (with Anakinra) increased OXPHOS.

Regarding blocking of hypoxia in-vivo to rescue the *Il1r1*^{-/-} mice: the only way to reduce hypoxia in-vivo that we know of is by treating with DMOG (Dimethyloxallyl Glyci), which is a HIF stabilizing drug that has been shown to reduce hypoxia in vivo.

We tried to reduce hypoxia by treating *Il1r1*^{-/-} mice with DMOG continuously during the infection (we tried two doses, the results here are of the higher dose), unfortunately we could not affect the hypoxic state of the kidneys.

Figure: Stabilization of HIF in-vivo in infected *Il1r1*^{-/-} and control mice

Il1r1^{-/-} and control mice were infected with 10⁵ CFU *C. albicans* and treated during the infection with either 8mg DMOG or vehicle, by an i.p. injection every 24h, starting one day before the infection.

(A) *C. albicans* titer in specified organs, 2 days after infection.

(B) Expression of the HIF-1 target genes Phd3 and Vegfa in the kidney, at 2 days post infection.

(C) Quantification of hypoxic areas in the kidneys at 2 days post infection. 1h prior to sacrifice, the mice received i.v. Pimonidazole injection. The kidneys were then harvested, prepared for immunohistology and stained with anti-pimonidazole antibody. Stained areas were quantified and normalized to total area of the kidney.

3.

Authors:

We edited the main text according to the suggestion (and we used 5% CO₂ in both hypoxic and normoxic conditions):

“C. albicans rapidly changes its morphology in response to the environment, nutrient availability, temperature, CO₂ and O₂ levels all influence the decision of morphology. It has been shown that at higher temperatures (37°C, compared to room temperature) C. albicans readily creates hyphae and upregulates virulence genes⁴⁷. This possibly happens when the fungi enter the host environment. CO₂ levels have also been shown to affect C. albicans virulence and hyphae formation^{48, 59}. We then asked whether the hypoxic environment in the kidneys of Il1r1^{-/-} mice during candidiasis promotes fungal invasion and/or pathogenicity.

We thus wanted to assess whether the hypoxia levels in the Il1r1^{-/-} kidneys can potentially promote fungal hyphae formation. Since Pimonidazole staining is positive in any oxygen levels under 1.3%, we decided to grow C. albicans in YPD agar in similar levels of hypoxia (1% oxygen) or normoxia in 37°C, 5% CO₂, and visualize the colonies. As predicted, while in normoxic conditions C. albicans did not create any hyphae, hypoxia promoted significant hyphae formation (Fig. 7F).” (page 14, line 418).

4. Definitions of the mono-macs and tissue-resident macs are a bit hazy. The tissue-resident macs are mostly defined by lower CD11b (at least in kidney). Knowing the recent developments about how monocytes can take on tissue-resident phenotypes (and therefore not true tissue-resident), I think some rewording of the subsets might help future-proof the results since ontogeny studies would be needed to be sure what is recruited and what is not.

Authors:

We agree with the reviewer (our group recently addressed the difference between fetal-derived vs bone marrow-derived alveolar macrophages (<https://pubmed.ncbi.nlm.nih.gov/35776802/>)), we cannot claim that the CD11b^{low} F4/80+ population in the kidney is a homogenous population of tissue-resident macrophages.

This definition of CD11b-low kidney macrophages may possibly include both fetal-derived and monocyte-derived macrophages, however in this particular study we do not have to be sure of their ontogeny.

We revised the text:

“we grouped the clusters identified in the kidney and in the brain into 8 cell types including neutrophils, eosinophils, cDC1, cDC2, Ly6C⁺ monocytes, MHCII⁺Ly6C⁺CD11b⁺ monocyte-derived inflammatory macrophages (Mo-Mac), CD11b^{low} kidney macrophages and brain resident CD45^{low}CD64⁺ microglia (Fig. 2B,C).” (page 5, line 130).

5. Related to above, was CX3CR1 used in the brain panel for data shown in Fig 2? This would help distinguish microglia better and pick out border macrophage populations.

Authors:

For this revision, we re-analyzed the brain populations with a different panel, that included CX3CR1. This re-analysis is added in supplementary figure 2A-C, and discussed above.

6. I'm not really convinced by the CitH3 staining shown in Fig 3. Can this staining be affected by tissue damage in the kidney caused by the infection? How can we be sure there are no artefacts caused by the fungus itself etc? An in vitro NETosis assay or additional controls would be helpful here.

Authors:

To address this concern, we performed an in-vitro NETosis assay using neutrophils from *Il1r1*^{-/-} and control mice:

“Thus, we assessed the capability of neutrophils from Il1r1^{-/-} mice to produce NETs in response to stimuli. Neutrophils isolated from bone marrow of WT and Il1r1^{-/-} mice had comparable capacity to produce NETs in response to PMA, as well as in response to C. albicans (Fig. 3C). Notably, IL-1b stimulation did not affect the NET formation in WT or in Il1r1^{-/-} mice.” (page 6, line 168)

Additionally, to show that the CitH3 staining is specific, and stains individual cells, here are zoomed-in images of CitH3 staining (from an *Il1r1*^{-/-} mouse):

7. Fig 1c confused me a bit – are the lines connecting average CFU from each experiment (i.e. an average of all WT and KO tested in that experiment) , where each experiment had 3-5 mice, or do the icons mean individual mice? If the latter, I don’t understand how the values could be paired. It would be helpful if the legend clarified total number of mice tested and across how many experiments.

Authors:

Yes, the lines indicate paired data – they connect average CFUs from the same experiment, where each experiment had 3-5 mice per group. We updated the figure legend to make it clearer.

Minor:

1. Fig S4 – no details given about n number or how many times these experiments were done.

Authors:

The information is updated.

2. I found Fig 5 really difficult to follow. Can the authors relabel some of the cell clusters with real labels as opposed to acronyms? I think this would help make the data easier to follow for non-kidney specialists

Authors:

We added the cell names in Fig. 5., the full description is specified in the figure legend.

Reviewer #2

1. The study provides little mechanistic insight on how IL-1 signalling might control metabolism in the kidney.

Authors:

Unfortunately, we didn't unravel the ultimate mechanism(s), which turned out to be very difficult considering the dysregulation of many metabolic pathways in virtually all types of kidney cells.

However, we have achieved to come closer by answering some open questions (see below)

There are also publications supporting the idea that IL-1 can have a direct influence on mitochondrial respiration:

*<https://www.ncbi.nlm.nih.gov/pmc/articles/PMC7566776/>

*<https://pubmed.ncbi.nlm.nih.gov/33804447/>

2. The metabolic shift is inferred from transcriptomics data only. Many metabolic changes are observed in *Il1r*-deficient kidneys: e.g. glycolysis is increased, but so is mitochondrial respiration and fatty acid beta oxidation. As such, the precise metabolic reprogramming under these conditions remains unclear.

Authors:

To better understand the metabolic shift in kidney cells of *Il1r1*^{-/-} mice, we made use of a Seahorse metabolic analyzer, working with an in-vitro kidney cell culture:

*"To assess the possibility that IL-1R intrinsically influences kidney cell metabolism, we used a human kidney podocyte cell line (differentiated AB8/13 cells). We performed an ATP assay on Agilent Seahorse XF96 analyzer, which allows the precise quantification of the rate of adenosine triphosphate (ATP) production from glycolysis and mitochondria in live cells⁴⁵. Before analysis, we treated the cells overnight with either IL-1b or the IL-1R antagonist Anakinra (Figure 7A, B). The assay revealed that podocytes increase the glycolytic source of ATP upon IL-1b treatment, while Anakinra treatment caused a significant increase of the mitochondrial (oxidative) source of ATP in the cells (Fig. 7C). Quantification of the mitochondrial ATP fraction over glycolytic ATP fraction demonstrated a significant effect of IL-1R signaling on the cell ATP production. In other words, while activation of IL-1R causes increased glycolytic ATP, blocking of IL-1R (as is happening in *Il1r1*^{-/-} mice) causes increased respiratory ATP. These results demonstrate that IL-1R intrinsically influences kidney cell metabolism, specifically shifting away from respiratory ATP production. In *Il1r1*^{-/-} mice, however, the kidney cells respond to *C. albicans* infection by increased cellular respiration."* (page 13, line 380)

The pathway analysis, which showed HIF-1 to be one of the most strongly upregulated pathways (Fig. 6 B), originally led us to suspect hypoxia in the kidney. HIF-1 activation explains the upregulation of glycolytic genes in *Il1r1*^{-/-} mice. We suspect that the HIF response in *Il1r1*^{-/-} mice is insufficient to block the upregulated oxidative phosphorylation.

3. The statement "metabolic cataclysm" needs clarification, both experimental and conceptual.

Authors:

We agree that this term might be confusing and decided to use instead the terms "metabolic shift" and "dysregulated metabolism" throughout the paper. Indeed, some of the metabolic pathways that we see upregulated in the single-cell transcriptomics data are not addressed, as we decided to focus on the most striking hypoxia and OXPHOS pathways. We could successfully experimentally show that IL-1R affects mitochondrial respiration (see previous point) and that *Il1r1*^{-/-} kidneys have significantly stronger hypoxia. Hopefully, this version will convey our hypothesis more clearly.

4. Hypoxia is a known trigger for hyphal growth in *C. albicans*. As such, Fig 6B does not add new knowledge.

Authors:

We cultured the *C. albicans* in the oxygen percentage that would simulate the situation in-vivo (1% oxygen), to make sure that these conditions indeed support the morphological shift.

5. While it is possible that increased hypoxia in *Il1r*-deficient kidneys is causing higher hyphal growth, other scenarios are also possible, and the data so far is not sufficient to conclude that this is the major reason for increase susceptibility of *Il1r*^{-/-} mice to infection. I therefore feel this part – connecting the metabolic change in *Il1r*^{-/-} mice to the fungal phenotypes - needs more work. For example, what are the transcriptional effects on *Candida* in these conditions, i.e. is there evidence that a hyphal hypoxic response is activated (see Sellam lab work)?

Authors:

We tried to address this by measuring the expression of some virulence genes in *C. albicans* in vivo, from infected kidneys of *Il1r1*^{-/-} and *Il1r1*^{+/-} mice.

Figure: expression of chosen *Candida* virulence genes in infected kidneys of *Il1r1*^{-/-} and *Il1r1*^{+/-} mice at 24h post infection.

**ECE1* = Candidalysin (fungal toxin)

**Als3* = surface protein, adhesin (attachment to cells), biofilm formation, invasion (binds to integrins and causes endocytosis)

**HWP1* = hyphal wall protein 1, morphology (important for virulence)

**Sap5* = Secreted aspartic protease 5, important for virulence (protein degradation, cell-cell adhesion)

Fungal RNA is of very low abundance in the infected kidney, and we have no means to measure a large sum of genes at once, without sacrificing a large number of knock out animals. Thus, we declare that the link between the metabolic change and the fungal virulence is yet to be proven, and we offer an associative link between IL-1R signaling and fungal “success”, hypoxia and a metabolic shift of the kidney stroma.

6. Are there any *Candida* mutants in the hyphal hypoxic response that could be used to corroborate the idea that specifically this pathway is impacted in *Il1r*^{-/-} mice?

Authors:

C. albicans creates hyphae in both *Il1r1*^{-/-} and control mice (see Fig. 7 D). We do not want to claim that *C. albicans* only makes hyphae in the *Il1r1*^{-/-} environment.

7. There is a large increase in fungal CFUs in *Il1r*^{-/-} and *Il1b*^{-/-} kidneys. This indicates that fungal growth, not just morphology, is impacted. In other words, I feel that the higher CFUs cannot be explained only by increased hyphal growth due to a hypoxic environment. Assessing the fungal response under these conditions might provide important information.

Authors:

By being more successful as a pathogen, *C. albicans* multiplies and grows more in the *Il1r1*^{-/-} host, creating more CFUs. We claim that *Il1r1*^{-/-} mice have fungal overgrowth due to favorable conditions for *C. albicans* to establish infection.

One aspect that makes *C. albicans* more successful as a pathogen is hyphae formation. The longer and more advanced the hyphae are, the harder it is for the host immune system to combat the fungi.

Another aspect that allows fungal overgrowth is immune evasion (referred in the discussion).

Morphological changes into hyphae cause masking of beta-glucans, which are molecules comprising the fungal wall and are the main molecules to be recognized by pattern recognition receptors (PRRs) on immune cells.

Taken together, morphology changes will impact the ability of the host to control the infection and in turn support fungal growth.

Reviewer #3

1. The IL-1R1 knock out mice have high organism burden and high inflammation. How does this impact the findings? Cytokine and chemokine production, immune cell recruitment, gene expression in infected tissues, etc. all can be a function of the level of disease in the tissue, not just the specific change being studied. For example, when the authors state that there was an elevated number of neutrophils in the knockout compared to WT mice, is that because of the change in IL-1R1 signaling directly or because a neutrophil independent factor caused increased organism burden in the knockout mouse and that increased burden results in different responses that are actually IL-1R1 independent?

Authors:

We believe that the high organism burden, seen starting from the second day post infection, indeed affects the findings at these time points. The elevated number of neutrophils, the high levels of pro-inflammatory cytokines, are all a factor of fungal burden. And we indeed try to make it clear in the paper, that the mice are suffering from a hyper inflammatory state due to fungal overgrowth in the kidney and the brain. Additionally, we try to confirm that neutrophil numbers are indeed a mirror of fungal burden:

*“Together, these data demonstrate that the fungicidal activity of neutrophils and iNOS expression by monocytes are not affected in $Il1r1^{-/-}$ mice, indicating that their severe susceptibility is less likely to be explained by defects in these well-established antifungal immune effector mechanisms. To explain the high number of neutrophils seen in $Il1r1^{-/-}$ mice, we infected WT mice with increasing infection doses of *C. albicans* (supplementary Fig. 3A). As expected, we observed a positive correlation between the dose of infection and neutrophil numbers in the kidney and the brain at 3 days p.i.. This correlation suggests that the observed numbers of neutrophils in $Il1r1^{-/-}$ mice at the height of infection is merely due to the severe fungal burden in these mice.” (page 7, line 180)*

This observation does not impact our findings, which eventually focus on the events in the first hours of infection. In the first hours of infection, we can attribute all the observations to be directly related to the lack in IL-1R signaling, rather than to the fungal burden (see next point).

a. This is challenge in any type of research like this but it requires extreme care during interpretation of results. In relation to this manuscript, this is extremely important for interpretation of the authors conclusions about the metabolic derangement and hypoxia that occur in the knockout mice. Is that because the knockout mice have severe disease and the gene expression changes/hypoxia are due to the massive tissue damage that is occurring in response to the disease or is it because of the lack of IL-1R1 signals?

b. The gene expression studies were done at 8 hours post infection, which suggests that it's not because of overwhelming disease nor global tissue necrosis. However –in order to make the argument that the gene expression changes are occurring prior to development of significant disease, the authors would need to at least present information about organism burden and conventional histology and/or other measures of tissue damage at the same time point as the gene expression studies.

Authors:

To make sure our observations are not secondary to disease progression, we analyzed fungal titer and kidney damage in the first 24h post infection:

*“To determine whether our observations of the transcriptional response could be secondary to kidney damage, we analyzed the kidney function over the course of infection, and collected blood and urine of infected *Il1r1^{-/-}* and control mice at 10h, 24h and 48h p.i. (Fig. 5D,E). The analysis revealed high blood urea and macroalbuminuria (ACR>300mg/g) at 48h p.i. in the *Il1r1^{-/-}* mice, indicating that they suffered from severe kidney damage at that time point. However, no signs of kidney damage were observed at 10h p.i., and only a slight microalbuminuria (ACR>30mg/g) was seen at 24h p.i. in the *Il1r1^{-/-}* mice. Additionally, the fungal burden in the kidneys of *Il1r1^{-/-}* and control mice at 8h p.i. is not different (supplementary Fig. 7A). These results confirm that the transcriptional phenotype we observe at 8h p.i. is likely directly affected by the absence of IL-1R and the presence of *C. albicans*, and it is not secondary to kidney damage, or differences in fungal burden.”* (page 11, line 318).

2. A large part of the manuscript involves looking at brain organism burden, cell recruitment, etc. in response to CNS infection with *C.albicans*. Is there any data to indicate that IL-1R1 signaling influences organism dissemination into the brain rather than just to responses that occur once the organism has crossed the blood brain barrier?

Authors:

That’s an interesting point. Looking at fungal burden in the brain at 8h post infection (supplementary Fig. 7A), we recovered comparable CFU of *C. albicans* from a perfused brain. If anything, there was a slight tendency to slower dissemination of *C. albicans* into the *Il1r1^{-/-}* brain, which was caught up by 16h. However, the number of CFUs in the brains was similar enough to cause recruitment of immune cells, which was severely impaired in *Il1r1^{-/-}* mice.

Minor concerns:

1. The authors state that their data in figure 3A-C “indicate that phagocytic and fungicidal activity of monocytes and neutrophils are not affected in *Il1r1^{-/-}* mice.” They have not presented any data on phagocytosis and have not presented data on the fungicidal activity of monocytes.

Authors:

We rephrased this part:

*“Together, these data demonstrate that the fungicidal activity of neutrophils and iNOS expression by monocytes are not affected in *Il1r1^{-/-}* mice, indicating that their severe susceptibility is less likely to be explained by defects in these well-established antifungal immune effector mechanisms.”* (page 7, line 180).

Data and methodology

1. In figure 1a, the authors normalize gene expression data to *g6pdx*. This gene undergoes changes in expression in conditions of hypoxia, oxidative stress, and metabolic derangements. Given the results presented later in the paper, is this a valid gene to use as a housekeeping control?

Authors:

We appreciate this comment. The figure is updated with normalization to TBP as a housekeeping gene.

2. In figure 1f/g – the use of hematoxylin as a background stain is very useful for identifying inflammatory infiltrates occurring in the regions of large clusters of *C.albicans* hyphae. However, it makes it very difficult to identify whether or not there are individual hyphae and/or clusters of yeast at the magnification presented. This would be stronger if the inserts had higher magnification and/or the authors include images with adjusted contrast (applied systematically, of course), to allow discrimination between small black dots (yeast or cross-sectional hyphae) and small dark purple dots (immune cells).

Authors:

We updated the figure with higher magnification, and we hope that it is now better possible to distinguish between fungal hyphae and cell infiltrates.

3. The authors state that *C.albicans* is distributed throughout brain tissue as seen in Fig 1g. The images presented could be the edge of a cluster – it doesn't mean the organisms are truly distributed throughout the tissue.

Authors:

In response to this concern, we updated Fig. 1G and added an enlarged area that exactly shows what we mean.

In Fig. 1G, right panel: while enlarged areas B and C show a dense growth of *C. albicans* in infection foci, the enlarged area A is an example for what we see outside of infection foci. There, *C. albicans* is dispersed and distributed in the form of single hyphae, and there are no infection foci in the proximity.

4. In figure 2a – it would be helpful if the cytokines/chemokines data includes the data for the same set of cytokines for both organs – this would allow better comparison between the organs.

Authors:

We updated the plots, and they are now more easily comparable (Fig. 2A). We indeed performed the same set on both organs; however we could not detect IL-10 in the brain, nor CXCL10 in the kidney, this is why we put those in the bottom.

Furthermore, the authors refer to data about CCL22, CXCL10, and CCL17 in the text but these aren't included in the figure.

Authors:

Thank you for this comment, it seems as through the versions we have missed this mistake. The text is now updated to describe the presented data in figure 2A.

5. Microglia are incredibly difficult to study as they are a heterogenous population of cells with a variety of unusual responses. The authors use CD64+CD45-lo staining to identify microglia. Microglia can also be CD45int and have a variety of other phenotypes. The authors should either include a statement that they are using only this staining and therefore the data may not reflect all microglia populations or present data with additional markers showing that this population really does reflect all microglia in their system (see Honarpisheh et al J Neuroinflammation 2020).

Authors:

We agree that our flow staining panels have been relatively simplistic, given what is known about microglia. Thus, we updated our statements in the text, and we discuss specifically CD64+CD45lo microglia. Additionally, for this revision we performed an additional analysis, including a few more flow markers that allow better characterization of brain populations (supplementary Fig. 2A-C).

Furthermore – the authors mention that irradiation / bone marrow transplant doesn't impact microglial populations. This is not always true (see Okonogi et al, J Radiat Res, 2014).

Authors:

When we performed the bone marrow chimera experiments, we were interested to know whether microglia were depleted by irradiation. Thus, we analyzed the chimerism of microglia in the brain. Here are the results, based on which we state that microglia populations are not impacted.

Suggested improvements

The authors should either a) include the gene expression data as is and add a significant discussion about all possible explanations such as stating that it is possible that IL-1R1 signaling suppresses the massive changes in gene expression or that the level of disease in the kidney at 8 hours is already sufficient to prompt broad scale expression changes (and any other possible explanations of the data), b) include data to demonstrate that the level of kidney disease in the WT and knockout mice is the same or both. They should also carefully edit the manuscript for other instances when they make conclusions about the role of IL-1R1 when they may be describing IL-1R1 independent effects that occur because of the different level of organism burden / disease.

They should also address the questions raised in the data and methodology section, above.

Clarity and context

The discussion does a very good job of presenting how their data aligns with previously published data and within our current understanding of *C.albicans* disease. However, there are multiple instances where they need to be more precise with their language. For example, there are statements such as “mean expression is displayed in heatmaps” when they are really displaying average number of cells; “oxidative” is written in a discussion about “nitrosative” damage, and “host derived cells” should be non-hematopoietic cells.

Authors:

This comment is valuable, as we missed these imprecisions. All points have been addressed and corrected in the main text.

Reviewer #4

1. In figure 2C, among all 7 celltypes shared by kidney and brain, quantification of cell numbers of five celltypes showed a similar pattern in WT and *Il1r1*^{-/-} mice, but an opposite trend was observed in brain and kidney for celltype cDC1s and cDC2s. Is there an assumption to explain the observation?

Authors:

At this point we do not have an explanation to the DC differences.

Due to the opposite correlations in the kidney and the brain between fungal burden and DC numbers, we hypothesize that DCs do not explain the susceptibility of *Il1r1*^{-/-} mice. Future investigation is needed to shed light on this phenotype.

2. For figure 1E, figure 2 and figure 3B, the authors evaluated the phenotypes at DPI=3. However, DPI=2 was chosen as the time point for figure 1F, 1G (GMS staining) and figure 3C (Ly6G and Cit H3 staining). Is there a specific reason to choose different DPI for different assays? According to figure 1D and 1E, an elevated fungal growth in brain has been observed in the first 24hrs and it lasted at least for 72hrs. It would be nice to check each individual at 24hrs, 48hrs and 72hrs post infection.

Authors:

At day 3 post infection, the *Il1r1*^{-/-} mice are extremely sick, and often we needed to euthanize them before, due to ethical reasons. At some point in the project, we opted to analyze mice at day 2 post infection, for the peak of infection and inflammation. At day 2, *Il1r1*^{-/-} mice have already a substantial fungal burden, that is significantly higher than in controls.

We added titer data for 24h post infection (Fig. 1D), 48h post infection (supplementary Fig. 1D), 8h and 16h (supplementary Fig. 7A).

"IL-1R-deficient mice are highly susceptible to C. albicans indicated by rapid loss of body weight and around 1000-fold elevated fungal titers in the kidney and the brain, but not in the spleen and liver, 3 days after infection with 10⁵ CFU C. albicans (Fig. 1B,C), when mice had to be euthanized due to animal ethics. Il1r1^{-/-} mice have significantly increased fungal titers in kidney and brain at 48h as well (supplementary Fig. 1D), and a low but significant difference to controls was observed as early as 24h p.i. (Fig. 1D)." (page 3, line 82)

3. Figure 1G showed the hyphae locations in a mouse brain at DPI=2, it seems there's a higher fungi density in the hypothalamus. It would be nice to show more staining results of various sagittal slices of the same individual from one side to another, or showing the same region of other individuals to indicate if there's a specific brain region/layer more sensitive to *C. albicans* infection.

Authors:

From our staining data, we could not identify specific areas of the brain where *C. albicans* "chose" to infect. Instead, we find *C. albicans* hyphae dispersed throughout the brain (see Fig. 1G, right panel, area A). More importantly, we do not wish to make claims regarding certain brain areas that are especially susceptible, as this would require a thorough analysis of various sections, that in our view, would be outside the scope of this paper.

Similarly, it would be nice to show more slices with NETs locations in figure 3C.

Authors:

Attached are three slices from three *Il1r1*^{-/-} mice, capturing different areas of the brain. The staining is of CitH3 (neutrophil NETs).

4. Suggestion, not required: For the first 16hrs post infection, the number of neutrophils in the brain of *Il1r1*^{-/-} mice are lower than WT mice. However, the number of neutrophils in the brain of *Il1r1*^{-/-} mice are much higher than WT mice 3 days post infection.

Is it possible to collect the infected brain samples at an early time point (8hrs or 16 hrs post infection) and a later time point (48hrs or 72hrs post infection) of *Il1r1*^{-/-} mice and perform the single cell sequencing? Maybe it can give a clue about how neutrophils and other celltypes reacted when infection progressed.

Authors:

This would indeed be an interesting experiment; however, we believe that it would go beyond the scope of this study, especially given the mouse and sequencing costs. Our findings regarding kidney metabolism and hypoxia are novel, and we chose to focus on these rather than on the brain. However, we do believe that there is still a lot that can be done to learn about the role of IL-1R in the brain during fungal infection.

5. For Figure 2B, CD45+ cells including brain specific cell type microglia were sorted out and taken for analysis. However, astrocytes which cannot be enriched by CD45+ selection are also considered as key immune effector cells in CNS and they can become activated in pathological states. I wonder if the authors have ever checked the cell number difference of astrocytes in WT and *Il1r1*^{-/-} post infection.

Authors:

For this revision, we reanalyzed the brain, this time preparing the brain to include astrocytes, and using a brain-specific antibody panel:

*“Flow cytometry analysis of the brain at 48h p.i., using a wider antibody panel, revealed significantly reduced numbers of both “activated” (Dectin⁺) and “homeostatic” microglia, as well as CD206⁺ border-associated macrophages and neutrophils in *Il1r1*^{-/-} mice, compared to *Il1r1*^{+/-} littermate controls (supplementary Fig. 2A-C). Interestingly, we observed significantly increased ACSA-2^{low} astrocytes and decreased ACSA-2^{high} astrocytes. At this time point (48h post-infection), there was already a considerably higher fungal load in the brains of *Il1r1*^{-/-} mice (supplementary Fig. 1D), which could indicate that microglia and border-associated macrophages might be especially vulnerable and die due to infection. The heightened inflammation may explain the difference in astrocyte numbers.”* (page 7, line 191)

Additionally, we used a similar analysis using out NFkB reporter mice, injected with IL-1b:

“In the brain, we found IL-1R-signaling mediated NF-κB activation in Border-associated macrophages (BAM) (GFP⁺CD45⁺CD206⁺), astrocytes (GFP⁺CD45⁺ACSA-2⁺) and in undefined CD45⁺CD31⁻EpCam⁻ cells, but not in endothelial and epithelial cells (Fig. 4D and supplementary Fig. 4B-C).” (page 9, line 251)

6. For figure 5C, it's a little bit confusing to understand according to the simple description of legend and the main text.

The Figure.5C legend says **“(C) Shared genesets between celltype specific DE genes for *Il1r1*^{-/-} infected (top) versus *Il1r1*^{+/-} infected (bottom) comparison”.**

The statement made about Figure 5C in the main text: **“Genes upregulated in the infected *Il1r1*^{-/-} mice were highly overlapping between different kidney cell types, suggesting a similar uniform response in the whole kidney.”**

Does it mean that the DEG sharing showed in figure.5C top panel was plotted based on up-regulated genes in *Il1r1*^{-/-} infected cells, while figure.5C bottom panel was plotted based on down-regulated in *Il1r1*^{-/-} infected cells (up-regulated genes in *Il1r1*^{+/-} infected cells) ?

Authors:

Yes, this is correct. The top panel in Fig.5C corresponds to the genes with adjusted p-value < 0.05 and log₂FC > 0 from *Il1r1*^{-/-} infected vs *Il1r1*^{+/-} infected comparison. Bottom panel is opposite (adj. p-value < 0.05 & log₂FC < 0), so corresponding to the genes upregulated in the *Il1r1*^{+/-} infected condition.

If that is the case, it's kind of weird to give the conclusion like “suggesting a similar uniform response in the whole kidney” only based on up-regulated genes.

Authors:

We rephrased it, limiting it to the upregulation of gene expression only,
*“Genes upregulated in the infected *Il1r1*^{-/-} mice highly overlapped between different kidney cell types, suggesting that in these mice, infection caused an upregulation of non-cell-specific gene programs. On the contrary, downregulated genes upon infection were rather cell-specific (Fig. 5C).”* (page 11, line 314)

The main point that we want to deliver is that upon infection in the *Il1r1*^{-/-} condition, similar gene programs are getting upregulated in different cell types, but the genes that are getting downregulated are more cell type-specific.

7. Minor question for figure.6A: As metascape uses all genes of the genome as the background gene set, the enrichment analysis might miss some terms which could be significant if the background list is limited to the genes expressed in the cells analysed in this study (for example, only consider the genes expressed in >5% of the cells and reads >= 2).

Authors:

We reran Metascape analysis with the 2541 background genes (expressed >5% of cells and have at least 2 reads per cell).

REVIEWER COMMENTS

Reviewer #1 (Remarks to the Author):

The authors have added significant amounts of new data including high dimensional flow and Seahorse metabolic data. I think the textual revisions made are satisfactory and the authors have sufficiently responded to each of my prior comments. Their responses are thorough and I have no further comments.

Reviewer #2 (Remarks to the Author):

I thank the authors for considering my comments and responding to them experimentally and by modifications to the text.

As the authors acknowledge, the link between the metabolic changes in the host dependent on IL-1 signalling and fungal phenotypes (hyphal growth, virulence) remains to be directly shown. Nevertheless, it is my opinion that the current study presents a new angle in our understanding of the interface between immune factors, host metabolism and fungal pathogenesis.

Reviewer #3 (Remarks to the Author):

The revised manuscript "IL-1 protects from fatal systemic candidiasis by inhibition of oxidative phosphorylation and hypoxia" by Horn, et al, is focused on identifying the mechanism(s) through which IL-1 receptor signaling acts to protect a "healthy" host from invasive candidiasis. The manuscript uses knock out mice and bone marrow chimera mice to dissect the immune responses to *C.albicans* infection. This manuscript addresses an important topic in the field of host-fungal interactions. In particular, the attempt to identify the mechanism of action through which IL-1 signaling impacts host defense rather than just identifying the pathways involved is very appealing. The biggest concern I have about the revised manuscript is that it is now overly full of potentially unconnected data, there are a huge number of interpretations of different findings, sometimes the authors link one finding to another that isn't especially relevant to it, and their conclusions are often overstating their findings. This is particularly prevalent given the many different comparisons that are happening across the different time points after infection. The main point(s) they are trying to make are now lost.

In an attempt to re-focus on the information the authors feel is most important, I went back to the abstract. Unfortunately, the abstract seems to be suffering the same problem and seems to have significant problems itself. Here are several statements from the abstract and specific concerns about them:

Authors: We thank the Reviewer for the opinion that we addressed an *important topic* and provided an *appealing attempt to identify mechanisms of defense*"

1. Lines 26/27: In the kidney, endothelial IL-1R contributes to fungal clearance independent of neutrophils...

They have not presented any data on what happens in the absence of neutrophils. There is an increase in fungal burden at 3 days after infection in *Pdgfb^{creERT2}Il1r1^{fl/fl}* mice compared to control mice but other data in the manuscript indicates that at 3 days after infection, there is substantial kidney damage, near overwhelming infection and a hyperinflammatory state so it's hard to know what the role of the endothelial cells was or when it was important. Furthermore, the authors are comparing the organism burden at day 3 with the neutrophil recruitment data from 8 hours after infection. They present data that indicates that the number of neutrophils in the kidney is tightly correlated to organism burden and the organism burden at 8 hours is the same in WT and *IL-1R1^{-/-}* mice. Thus, it's very difficult to untangle what might be happening and what can be attributed to the endothelial cells.

Authors:

We found that kidney endothelial IL-1R contributes to the defense against *C. albicans* in the kidney. However, we would not like to make any claims regarding the protective mechanism of the endothelial cells. The only claim we wish to make is that the role of endothelial IL-1R is independent of neutrophil *recruitment*. We added this important word in the abstract (line 27). Our statement regarding the neutrophil recruitment is based on the following observations:

1. We show that the fungal titer is higher in the kidneys of *pdgfb^{cre}Il1r1^{fl/fl}* mice, compared to littermate controls at day 3, but by far not as high as in the global *Il1r1^{-/-}* mice (Fig. 4E). Thus, we concluded that endothelial IL-1R in the kidney contributes to fungal clearance.
2. Unlike the *Il1r1^{-/-}* mice, who have slightly delayed neutrophil recruitment (seen at 8h but not at 16h) (Fig. 3E), *pdgfb^{cre}Il1r1^{fl/fl}* mice showed unimpaired neutrophil recruitment (Fig. 4F, G).
3. Neutrophil infiltration in response to IL-1b treatment (in naïve mice) was not-affected by the absence of IL-1R on endothelial cells:
"Interestingly, IL-1b treatment was sufficient to induce neutrophil infiltration to the kidney and brain of wild-type (*Il1r1^{fl/fl}*) mice, and a comparable recruitment was observed in *Pdgfb^{creERT2}Il1r1^{fl/fl}* mice (Fig. 4G), suggesting that the recruitment of myeloid cells is independent of endothelial expression of IL-1R." (lines 282-285)

Thus, we conclude that the role of endothelial IL-1R is independent of *early neutrophil recruitment*.

At 3 days p.i. the neutrophil numbers are elevated in both *Il1r1*^{-/-} mice and *pdgfb*^{cre}*Il1r1*^{fl/fl} mice, reflecting the higher fungal burden in both groups.
We edited the text so that it hopefully reflects our statement more precisely.

2. Lines 27-29: In the brain, IL-1R-signaling indirectly recruits neutrophils and monocytes by regulating chemokines and adhesion molecules...

There is data showing that CXCL1 levels in both brain and kidney are substantially lower at 8 hours after infection in IL-1R1-deficient mice and at the same time point, there are fewer neutrophils in both brain and kidney as well. Given the data in the paper and that presented by others (who previously demonstrated the same findings), it is certainly likely that it is the CXCL1 effect that's altering neutrophil recruitment. This isn't novel nor is it a completely explored mechanism.

More concerning is the statement that IL-1R signaling regulates adhesion molecules which then recruit neutrophils. The only times the word "adhesion" appears in the manuscript are line 29 in the abstract and in the gene ontology list within Figure 6A.

Authors:

In the text (lines 207-212), we describe RNA sequencing data from the brain, collected 8h post infection (supplementary figure 2E-H). The data show that in addition to reduced CXCL1, *Il1r1*^{-/-} mice had reduced transcription of the adhesion molecules *Icam*, *Sele* and *Vcam1*. However, we initially used the term "extravasation" instead of "adhesion" (now corrected – lines 207-209).

Additionally, we observed reduced levels of the chemokines *Ccl1* and *Csf1*.

These adhesion molecules and cytokines have been well-documented in trans-endothelial migration (extravasation or diapedesis) and monocyte differentiation, respectively. Our data provide additional insights into the role of IL-1R in early brain infection, extending it beyond the induction of *Cxcl1*.

3. Lines 29-30: Single-nucleus-RNA-sequencing revealed *Il1r1*-mediated NF-κB activation in distinct kidney cell types.

This is supported by Figure 4D but of unclear significance to the remainder of the investigation.

Authors:

One of the key questions in our study was: which cells of the kidney mediate the protective effect of IL-1R? To address this, we first needed to identify which cells directly respond to IL-1 signaling.

It is known that *Il1r1* transcripts are present at low copy numbers in many cell types, particularly in brain cell types, although IL-1R signaling is functionally essential (PMID: 30683620, Immunity 2019, and further references therein; PMID: 25698726, J Neurosci 2015). This, together with a low amount of RNA in the nucleus, resulted in a poor detection of *Il1r1* transcripts in our single-nucleus-RNA dataset, limiting our ability to identify the cells directly. Therefore, we used alternative methods to approximate the cellular targets of IL-1 cytokines. One approach was to look for reduced in NF-κB activity in the cells of the *Il1r1*^{-/-} mice: NF-κB activation early after infection may indicate IL-1R signaling, given that it is a key downstream target of IL-1R. In our dataset, we observed significantly reduced NF-κB activity in 11 out of 18 kidney cell types in *Il1r1*^{-/-} mice, suggesting that these cells typically exhibit IL-1R activity. These include endothelial cells, podocytes, tubule cells (proximal, intercalated and distal), medullar cells and principal cells – supporting our hypothesis that IL-1R provides protection against *C. albicans* by targeting most kidney cell types.

"A reduction in NF-κB pathway activity is consistent with the absence of IL-1R and is observed to a significant extent in 11 cell types out of 18, suggesting the importance of IL-1R in most kidney cell types." (lines 354-356)

Related to our results shown in Figure 4C, D, where we measured IL-1R-mediated NF-κB activation in single cells by flow cytometry using NF-κB reporter mice, please see our answer to point 12 below.

- 4. Lines 30-32: Our data reveal excessive ribosomal activity and oxidative phosphorylation across all cell types in the kidney of Il1r1-deficient mice within a few hours upon infection, causing localized hypoxia at infection foci...
There is no data on ribosomal activity in the manuscript.**

Authors:

We found increased expression of ribosomal genes, as shown in Supplementary Figure 7B,C. Unfortunately, we missed mentioning these data in the results and discussion sections of the manuscript, and now updated the text (lines 338-341). However, we agree with the reviewer that we did not directly assess ribosomal activity. For this reason and because we do not further address the implication of increased expression of ribosomal genes experimentally, we abstain from using this term in the abstract. We have updated the abstract by replacing the word “ribosomal” with “metabolic” to better reflect the focus of our findings.

- 5. There is upregulation of oxidative phosphorylation gene pathways in the kidney of IL-1R1-deficient mice at 8 hours after infection but no in vivo data on the process itself.**

Authors:

We attempted to measure oxidative phosphorylation ex vivo in infected mice; however, we encountered technical challenges with isolating non-immune cells from the kidney and accurately measuring their metabolism. As an alternative, we used kidney cell lines to study the effect of IL-1R signaling on oxidative phosphorylation.

While we acknowledge that in-vivo data would have been more exciting, this approach allowed conceptual insight and indicated a direct effect of IL-1 signaling on cellular metabolism.

- 6. Furthermore, there is no data to demonstrate that it is increased oxidative phosphorylation that is causing the hypoxia which is seen at 24 hours after infection.**

Authors:

This is correct, at this stage, the findings are correlated, and causation is yet to be proven. We updated the text to clarify that our data indicates the association of oxidative phosphorylation and hypoxia.

In our discussion of the data, we hypothesize that the increased oxidative phosphorylation, along with the pathogen, could together contribute to the exacerbation of hypoxia. We edited the text to reflect the state of our knowledge (lines 411-418, 513-516)

“In the absence of IL-1R, increased oxidative phosphorylation is associated with stronger hypoxia, suggesting the possibility that kidney cells consume more oxygen upon infection. IL-1 is critical to prevent hypoxia in the kidney microenvironment around C. albicans infection foci and, thereby, Candida invasiveness.” (lines 513-516)

- 7. In the revised manuscript, the authors have included information correlating levels of neutrophil recruitment with organism burden and added an interpretation that the number of neutrophils in the tissue “at the height of infection” is due to the fungal burden. They also added data on BUN and ACR that demonstrates that at 10 hours, there is no significant difference between WT and knockout mice in terms of kidney damage. This is very supportive of their conclusions that their 8 hour gene expression findings are related to IL-1R signaling and not to overwhelming infection. However – it is still not at all clear that the hypoxia they demonstrate occurring at 24 hours after infection (Fig 7D) is a direct effect of IL-1R signaling and not either organism burden or neutrophil activity.**

Authors:

Unfortunately, hardly any Pimonidazole-positive areas were detected at 8h post infection. Since the pimonidazole sensitivity is limited, we rely on the single nuclear RNA sequencing data, which indicates that the kidney cells of *Il1r1*^{-/-} mice experience hypoxia as early as 8h post infection, based on significant activation of HIF-1 α .

Importantly, the difference in hypoxia observed at 24h p.i. cannot be explained by differences in fungal burden, since there were no differences between WT and *Il1r1*^{-/-} mice at this time point (see point 8 below).

8. Figure 1D indicates that the organism burden in the kidney at 24 hours after infection is statistically “not different” from that seen in WT mice although the mean burden in IL-1R1-deficient mice is approx. 0.3 logs higher than WT and with the distribution of the data points, one wonders if that would be “significant” if more points had been included.

Authors:

While Figure 1D in the original version of the manuscript shows an insignificant elevation of fungal burden in the kidney after 24h, we saw no difference between WT and KO mice in other experiments.

In the revised manuscript, we replace the data from the original Figure 1D with the data below, which show measurements of fungal burden from the same experiments used for the hypoxia quantification with Pimonidazole (Fig 7) and therefore support that the differences in hypoxia differences are not due to differences in fungal burden. A significant difference in the fungal burden in the kidney manifests after 24h, while in the brain, the titer is already increased at 24h.

9. In figure 7, though, the areas with increased pimonidazole staining co-locate with areas that clearly have a much higher local organism burden. Based on other studies, those foci of infection also typically are strong foci of neutrophil recruitment. Given all the extra activity that is going on in those areas, isn't it possible that the hypoxia seen in the kidney is an indirect effect stemming from the host/pathogen interaction rather than from IL-1R1-deficiency?

Authors:

It is possible that host pathogen interactions contribute to the development of hypoxia at 24h. However, there is no difference in neutrophils numbers between *Il1r1*^{-/-} and control mice at that time point. Additionally, our single nucleus sequencing data indicate that hypoxia pathways are already active at an early timepoint (8h), likely reflecting the result of direct of IL-1R deficiency, rather than secondary affects. This was the most striking phenotype we observed, which is why we followed up on it. The protective role of IL-1R is likely relevant at very early stages of infection, and the later one looks, the more secondary effects become apparent.

10. The additional data goes a long way to argue that the changes in gene expression seen at 8 hours after infection in the kidney are a function of IL-1R1 signaling. This is a very interesting finding. The finding that it is a non-hematopoietic requirement for IL-1R1-deficiency that is driving the ability to control infection is also very interesting. If the authors could focus on these points, they could make this argument clearly and relatively cleanly. Without full mechanistic data, no one could “prove” this is a direct effect, but their data is interesting enough to warrant publication and further investigations. In order to do that, they need to substantially trim and organize the data and their discussion of it.

Authors:

We edited the discussion of the manuscript, making sure that the important observations are in focus.

- 11. I don't think that the inclusion of data on brain infection is substantially contributing to their points and it is certainly "muddying the waters." It should be moved to a different manuscript.**

Authors:

To address the concerns of the other three reviewers, we had to do more experiments and added additional data on the mechanism in the brain. While we agree that including data from both organs makes the manuscript rather dense, we believe that it is important to highlight that IL-1R provides protection against *C. albicans* in both organs through non-hematopoietic cells, albeit via distinctly different mechanisms. We find this distinction interesting and scientifically intriguing.

- 12. Likewise, I'm not sure that the data in Figures 2 and Figure 4C, D, and G contribute substantially to the manuscript.**

Authors:

1. The data in Figure 2 reveal the important finding that, despite IL-1's established role as a pro-inflammatory cytokine, the absence of IL-1R signaling does not lead to reduced inflammation.

"These data demonstrate that Il1r1^{-/-} mice experience a hyperinflammatory state at this stage of the infection, indicating that IL-1R signaling is not necessary for the activation of a pro-inflammatory immune response during systemic C. albicans infection." (lines 119-121)

2. In Figure 4C,D, we aim to identify the cells in the kidney and brain that may contribute to IL-1R-mediated protection. Our formulation in the paper was not clear, thus we improved it:

"IL-1R triggering induces a signaling cascade resulting in NF-κB activation³⁰. IL-1R1 cell surface expression is too low for detection by flow cytometry (data not shown); therefore, to identify the non-hematopoietic cell types in the kidney and brain that contribute to IL-1R-mediated defense against C. albicans, we measured NF-κB activation in response to IL-1b stimulation as a proxy for IL-1R expression." (lines 245-249)

Figure 4C demonstrates that multiple non-hematopoietic kidney cells can mediate the protective effects of IL-1, and Figure 4D highlights astrocytes as a key population that responds to IL-1.

3. In Figure 4G we make sure that IL-1-driven neutrophil infiltration is independent of endothelial IL-1R expression.

We edited the text to make it clearer:

"We next treated naive Il1r1^{fl/fl} and Pdgfb^{creERT2}Il1r1^{fl/fl} mice with IL-1b intravenously and quantified the myeloid cells in the kidney and the brain after 4h. Interestingly, IL-1b treatment was sufficient to induce neutrophil infiltration to the kidney and brain of wild-type (Il1r1^{fl/fl}) mice, and a comparable recruitment was observed in Pdgfb^{creERT2}Il1r1^{fl/fl} mice (Fig. 4G), suggesting that the recruitment of myeloid cells is independent of endothelial expression of IL-1R." (lines 280-285)

- 13. The data on hypoxia is relevant to the gene expression data and certainly interesting but still a bit underwhelming. This includes my concern above about the co-localization of the hypoxia with dense organism numbers and, one expects, high density of neutrophils.**

Authors:

The colocalization of hypoxic regions with the *Candida* indeed suggests that the pathogen itself drives the development of hypoxia by depleting the oxygen in its environment. However, this doesn't happen to the same extent in the control mice (Figure 7D, top panel). At 24h post infection there is not yet higher fungal burden or neutrophil numbers in the *Il1r1^{-/-}* mice. IL-1R signaling seems to protect the kidney from the development of hypoxia, and we already see indications for it at 8h post infection: *Il1r1^{-/-}* mice upregulate the HIF-1α pathway (indicating that already then, the cells experience hypoxic conditions).

This leads us to speculate that the hypoxia is a direct consequence of IL-1R-deficiency.

“Interestingly, we found co-localization of hypoxic regions and infection foci (Fig. 7D) more in the *Il1r1^{-/-}* mice than in the *Il1r1^{+/-}* controls. This observation suggests that *C. albicans* itself could contribute to the creation of hypoxia by overconsumption of oxygen. However, since IL-1R seems to prevent most of the hypoxia, as seen in *Il1r1^{+/-}* control mice, we believe that the kidney cells have a significant role in its exacerbation. (lines 411-415)

14. An additional concern is the seahorse data on a kidney podocyte cell line. If they feel that endothelial cells are the location at which presence or absence of IL-1R1 matters, why did they test seahorse data on podocytes?

Authors:

We could demonstrate that endothelial IL-1R plays a role; however, it is clear that IL-1R in other cell types is also important. Complete knock out of IL-1R results in 3 to 4-log higher fungal load in the kidney at 3 dpi, while deletion of IL-1R in endothelial cells only led to 0.5-1-log increase at the same time point. Additionally, our data suggest that most, if not all, non-hematopoietic kidney cells express IL-1R. Sequencing data from 8h post infection indicate that IL-1R deficiency affects almost every kidney cell type including podocytes and tubular cells (Figure 6A, right panel), as evidenced by significant upregulation of metabolic pathways, including oxidative phosphorylation. For this reason, we did not restrict our choice of cell lines for the Seahorse experiments.

To provide evidence that IL-1 β directly influences cellular metabolism and mitochondrial respiration, we tested two human kidney cell lines that were available to us: the human proximal tubule cell line (HK-2) and a differentiated podocyte cell line (AB8/13). Although the HK-2 cells responded to IL-1 β in a trend similar to the podocytes, their overall ATP production was extremely low (see ECAR and OCR levels below). Unfortunately, these results were not significant.

In contrast, the podocyte cell line exhibited higher overall ATP production, and a stronger response to stimulation. Therefore, we chose to present this data as an example of how IL-1R signaling can modulate kidney cell metabolism by reducing mitochondrial ATP production relatively to glycolytic ATP.

Figure: Seahorse ATP assay, performed on proximal tubule cells (HK-2 cells), treated overnight with either IL-1 β or Anakinra.

Figure: Seahorse ATP assay, performed on differentiated podocytes (AB8/13 cells), treated overnight with either IL-1 β or Anakinra.

15. Why isn't there any statistical analysis of this data?

Authors:

We performed statistical analysis, see figure 7C, right panel (statistical test is specified in the figure legend).

16. Is the amount of difference observed either statistically or biologically relevant? They include untreated cells and cells treated with IL-1B or anakinra. What was the anakinra doing in the cell cultures that should have had very low levels of IL-1B in them to start with? The anakinra argument would have been much more convincing if they had treated cells with both anakinra AND IL-1B.

Authors:

The biological significance lies in the fact that IL-1 β alone could influence the metabolism of kidney cells, though the strength of this effect depends on the specific cell line (see point 14). Additionally, this data supports the single-nucleus sequencing data: IL-1 β reduces the mitochondrial/oxidative source of ATP.

Regarding the function of Anakinra in the culture: when we performed the experiment, we were unsure if the cells produce any IL-1, and therefore whether Anakinra would have an effect. Fortunately, adding Anakinra alone had a significant effect, indicating that low IL-1 levels were probably present in the unstimulated condition. We did not combine IL-1 β and anakinra because both target the same receptor, and we would need to know the appropriate stoichiometry to calculate the right doses to achieve a negative net effect. Our primary interest was the net effect of the treatments: increased IL-1R signaling in the IL-1 β treated cells, and decreased IL-1R signaling in the Anakinra treated cells. This allowed us to clearly observe the metabolic effect of IL-1R signaling.

17. There is very interesting data in this manuscript but it is still hampered with a variety of over-conclusions and inaccurate statements.

There are still a large number of places in the results and discussion which involve results obtained from 48 hours, 2.5 days, and 3 days after infection – this is the time that the authors have shown to line up with significant kidney damage and very high organism burden and neutrophil density. How much weight should be given to those results?

Authors:

After 48h of infection, we demonstrated that the kidney suffers from severe damage, and both the kidney and the brain enter a hyper-inflammatory state.

We used the 2.5/3 day time points interchangeably because we had to euthanize mice in some experiments before day 3 due to severe weight loss and disease burden (defined by ethical authorities).

We believe that obtaining data from these time points has led to important observations:

- Regarding the data showing a hyperinflammatory state in *Il1r1*^{-/-} mice 2.5/3 dpi: IL-1 is generally considered a pro-inflammatory cytokine. It was unclear whether deletion of IL-1R would cause reduced inflammation and the organism to become 'blind' to the pathogen, leading to increased fungal burden and death.

Some literature on IL-1 family of cytokines suggests that their pro-inflammatory characteristics can be detrimental during infections, potentially leading to damaging immune pathology. In such cases, deletion of the receptor could result in reduced inflammation and tissue damage.

However, our data clearly show that this is not the case for IL-1 in systemic candidiasis: deletion of the IL-1R leads to an even more pronounced inflammatory response.

- Regarding the titer differences at 2-3 days post infection (figure 1, figure 4): these differences highlight the susceptibility of the mice to *C. albicans*. The impact of the IL-1R deficiency in the form of fungal burden is only evident after 24h.

18. There are still a variety of overstatements such as:

**Lines 508/509... tissue hypoxia in the microenvironment around *C. albicans* infection foci, which we and others showed to increase *C. albicans* pathogenicity
The authors only showed that hypoxia increases invasive growth, not pathogenicity.**

Authors:

Invasive growth and pathogenicity are often used interchangeably, but it is important to be more precise. We have revised the relevant statements in the manuscript.

“IL-1 is critical to prevent hypoxia in the kidney microenvironment around *C. albicans* infection foci and, thereby, *Candida* invasiveness. Indeed, in agreement with others^{48, 49}, we showed that a hypoxic environment of 1% oxygen promotes *Candida albicans* hyphae formation.” (lines 514-517)

**19. Lines 534/535: Taken together, our data reveal a non-conventional role for IL-1b, and to a minor extent IL-1a, during early stages of *C. albicans* invasion into the kidney.
There is no data on how IL-1a may be involved.**

Authors:

In Figure 1E we used *Il1b*^{-/-} and *Il1a*^{-/-} mice, showing that deletion of IL-1 α alone resulted in very mildly increased susceptibility to *C. albicans*, only in the brain. Consistently, Il-1b transcripts in the kidney are induced about 9 log upon infection, whereas there is only 1-2 log increase in Il1a transcripts. Of course, both IL-1a and IL-1b induce IL-1R signaling, and the mechanism would be the same. Despite > 3 decades of research, it is still unclear why cells produce Il1a and Il1b differentially.

However, considering the incremental contribution of Il1a to *Candida* protection, we removed the statement about IL-1a.

20. Other concerns:

-EVERY figure should include a label in that figure about what time point the data came from.

Authors:

We updated the figures accordingly.

-Data on how many replicates per experiment and how many repeats of each experiment should be included

Authors:

We added the information in the figure legends for all experiments performed ≥ 2 .

The following figures are based on a single experiment:

Fig 1A shows upregulation of Il1a and Il1b transcripts upon infection. Informative data, but not critical for the message of the manuscript. We can omit them, if this reviewer insists.

The differences in cytokine and chemokine protein levels at 2 dpi (Fig 2), were confirmed by differences in transcript levels determined by RNA seq from an independent experiment, which we did not show in the manuscript.

Fig 3D: histology of neutrophil NETs. Clear black and white data, that support that NETs are created in KO mice (attached sections from other *Il1r*^{-/-} mice). This data was also supported by in vitro NET formation assay (Fig. 3C)

Fig 3H: reduced CXCL1 at 8h, was supported by RNA data from an independent experiment (supplementary Fig. 2)

Fig 4F,G: no difference was observed between pdgfb-cre and control mice.

Reviewer #4 (Remarks to the Author):

The authors achieved a significant improvement compared to the last version. Please find some points need to be addressed as follows:

- 1. In Line191-199, “Flow cytometry analysis of the brain at 48h p.i., using a wider antibody panel, revealed significantly reduced numbers of both “activated” (Dectin+) and “homeostatic” microglia, as well as CD206+ border-associated macrophages and neutrophils in *Il1r1*^{-/-} mice, compared to *Il1r1*^{+/-} littermate controls (supplementary Fig. 2A-C). Interestingly, we observed significantly increased ACSA-2^{low} astrocytes and decreased ACSA-2^{high} astrocytes.....”**

Is there a cell number difference in astrocytes/BAM can be observed between *Il1r1*^{-/-} mice and *Il1r1*^{+/-} mice without infection? It was only mentioned that there’s no difference in the cell types shown in supplementary figure 1C between *Il1r1*^{-/-} and WT mice, but no similar statement for supplementary Fig. 2A-C.

Authors:

Thank you for this comment. We analyzed the brain of naïve *Il1r1*^{-/-} mice with the extended antibody panel used for supplementary figure 2 (supplementary Fig. 2D).

BAMs were comparable between the groups. Interestingly, we could observe decreased ACSA-2 astrocytes already in uninfected brains of *Il1r1*^{-/-} mice compared to controls, indicating that IL-1R is necessary for the development or maintenance of ACSA-2 high astrocytes. Further research is required to explore the role of IL-1R in astrocytes.

- 2. Line 267-269: “Like global *Il1r1*^{-/-} mice, at 3 days p.i., *PdgfbicreERT2 Il1r1fl/fl* mice showed elevated number of neutrophils and inflammatory monocytes in the kidney, relatively to their *Il1r1fl/fl* littermates (Supplementary Fig. 5C). ”**

No p-value/significance indicated in the figure S5C.

Authors:

We added labels in the plots where there is significance ($p < 0.05$). These are the neutrophils and the Ly6C⁺ monocytes.

- 3. In the response to reviewers, the authors suggested “Attached are three slices from three *Il1r1*^{-/-} mice, capturing different areas of the brain. The staining is of CitH3 (neutrophil NETs). ”**

Cannot find any attached slices in the response file or supplementary file in terms of CitH3 staining. Please specify.

Authors:

The file called “NETs examples.pdf” can be found under supplementary information

- 4. minor point: Line 102: *Il1r1*^{-/-} should be *Il1r1*^{-/-}**

Authors: corrected, thank you for pointing it out.

p-t-p February 2025

REVIEWERS' COMMENTS

Reviewer #3 (Remarks to the Author):

The authors revision of the manuscript appropriately addresses my previous concerns. I agree with the authors that their data comprises new and interesting findings that will prompt many future questions in the field of *C.albicans* host-pathogen interactions.

A small concern is that the majority of the figure legends, including all of the figure legends for the primary (non-supplemental figures) do not indicate what p value is used to designate a *, **, or *** in the figures.

Our answer:

We updated the figure legends to include the information regarding the number of stars and significance levels.

I recommend that the following very minor inconsistencies, which might be typographical errors, be addressed:

Lines 86-87: The text indicates that a “small but significant difference to controls was observed in the brain as early as 24 h p.i (Fig 1D) but the figure doesn’t have any markings to indicate significance and a p value is not mentioned in the figure legend.

Our answer:

Thank you, this is a mistake. We updated the text to represent the data correctly.

“however no difference to controls was observed at 24h p.i. (Fig. 1D).”

Lines 89-90: The text reports that “the lack of IL-1a did not impair the ability to fight the fungus...” but in Figure 1E, brain samples, there is an apparent difference between the values for WT and IL-1a^{-/-} mice and there is a * for the comparison between those samples.

Our answer:

Thank you. Also here, we corrected the text to better represent the data.

“Interestingly, the lack of IL-1a only slightly increased susceptibility to the fungus, and only in the brain;”

Lines 101-102: The text reports that the data in Fig 1G is from day 3 but the figure indicates 2 dpi

Our answer:

This was a typo, corrected to 2dpi.

Reviewer #4 (Remarks to the Author):

The authors made essential revisions to improve the manuscript and supplied elaborate responses to the reviewers. I have no further comments.